



# Multi-pollutants emissions from the burning of major agricultural residues in China and the related health-economic effect assessment

Chunlin Li[1], Yunjie Hu[1], Fei Zhang[1], Jianmin Chen[1,2], Zhen Ma[1], Xingnan Ye[1], Xin Yang[1,2], Lin Wang[1,2], Xingfu Tang[1], Renhe Zhang[2], Mu Mu[2], Guihua Wang[2], Haidong Kan[3], Xinming Wang[4], Abdelwahid Mellouki[5]

[1]Shanghai Key Laboratory of Atmospheric Particle Pollution and Prevention (LAP[3]), Department of Environmental Science & Engineering, Fudan University, Handan Road 220, Shanghai 200433, China

[2]Institute of Atmospheric Sciences, Fudan University, Handan Road 220, Shanghai 200433, China

[3]Public Health School, Fudan University, Dongan Road 120, Shanghai 200032, China

[4]State Key Lab of Organ Geochemistry, Guangzhou Institute of Geochemistry, Chinese Academy of Sciences, Kehuajie Road 511, Guangzhou 510640, China

[5]Institut de Combustion, Aérothermique, Réactivité et Environnement, CNRS, 45071 Orléans cedex 02, France

*Correspondence to*: J. M. Chen (jmchen@fudan.edu.cn)

**Abstract.** Multi-pollutants in smoke particulate matter (SPM) were identified and quantified for biomass burning of five major agricultural residues such as wheat, rice, corn, cotton, and soybean straws in China by aerosol chamber system combining with various measurement techniques. The primary emission factors (EFs) for $PM_{1.0}$ and $PM_{2.5}$ are 3.04-12.64 and 3.25-15.16 g $kg^{-1}$. Organic carbon (OC), elemental carbon (EC), char-EC, soot-EC, water-soluble inorganics (WSI), water-soluble organic acids (WSOA), water-soluble amine salts (WSA), trace mineral elements (THM), polycyclic aromatic hydrocarbons (PAHs), and phenols in smoke $PM_{1.0}/PM_{2.5}$ are, 1.34-6.04/1.54-7.42, 0.58-2.08/0.61-2.18, 0.51-1.67/0.56-1.76, 0.05-0.41/0.05-0.42, 0.51-3.52/0.52-3.81, 0.13-0.64/0.14-0.77, $(4.39-85.72/4.51-104.79)\times10^{-3}$, $(11.8-51.1/14.0-131.6)\times10^{-3}$, $(1.1-4.0/1.8-8.3)\times10^{-3}$, and $(7.7-23.5/9.7-41.5)\times10^{-3}$ g $kg^{-1}$, respectively. EC and soot-EC mainly exist in



$PM_{1.0}$, which are confirmed by morphology analysis. Heavy metal-bearing particles
favor to reside in the range of smoke $PM_{1.0-2.5}$.
With respect to five scenarios of burning activities or straw field burning rates, the
total emissions of SPM from agricultural open burning in China in 2012 were
estimated for $PM_{2.5}$, $PM_{1.0}$, OC, EC, char-EC, soot-EC, WSI, WSOA, WSA, THM,
PAHs, and phenols to be 0.74-1.24, 0.66-1.11, 0.32-0.53, 0.10-0.16, 0.08-0.14,
0.02-0.03, 0.18-0.30, 0.019-0.031, $4.23\text{-}7.19 \times 10^{-3}$, $6.36\text{-}10.64 \times 10^{-3}$, $0.35\text{-}0.59 \times 10^{-3}$,
and $2.02\text{-}3.40 \times 10^{-3}$ Tg, respectively. The emissions were further temporal-spatially
characterized using geographic information system (GIS) at different regions in
summer and autumn post-harvest periods. It is found less than 25 % of the total
emissions were released during summer harvest period that was mainly contributed
by the North Plain and the Central of China, especially Henan, Shandong, and Anhui,
leading the top three provinces of smoke particle emissions.
Flux concentrations of primarily emitted smoke $PM_{2.5}$ that were calculated using
box-model method based on five versions of emission inventories all exceed the
carcinogenic risk permissible exposure limits (PEL). The health impacts and
health-related economic losses from the smoke $PM_{2.5}$ short-term exposure were
assessed. The results show that China suffered from 7836 (95 % confidence interval
(CI): 3232, 12362) premature mortality and 7267237 (95 % CI: 2961487, 1130784)
chronic bronchitis in 2012, which led to 8822.4 (95 % CI: 3574.4, 13034.2) million
US\$, or 0.1 % of the total GDP losses. We suggest that percentage of open burnt
crop straws at post-harvest period should be cut down by over 97 % to ensure risk
aversion from carcinogenicity, especially the North Plain and the Northeast, where
the emissions should decease at least by 94% to meet the PEL. Under such emission
control, over 92 % of the mortality and morbidity attributed to agricultural fire
smoke $PM_{2.5}$ can be avoided in China.
**Key words:** agricultural straw burning, aerosol chamber, smoke particle, emission
factor, emission inventory, health effect, emission control policy



## 1 Introduction

Biomass burning (BB) is a significant source of particulate- and gaseous- pollutants (Andreae et al., 2001; Clarke et al., 2007; Ram et al., 2011; Saikawa et al., 2009; Tian et al., 2008). It was estimated that open burning of biomass contributed approximately 40% of the globally averaged annual submicron black carbon (BC) aerosol emissions and 65 % of primary OC emissions (Bond et al., 2013). China is the major contributor that bears over 24 % of global emissions of carbonaceous aerosols, especially from agricultural field burning, about 0.04~0.5 Tg EC and 0.4~2.1 Tg OC are released annually (Bond, 2004; Cao et al., 2006; Qin et al., 2012; Saikawa et al., 2009), resulting in great radiative forcing, air quality deterioration, visibility reduction, premature mortality, and economic loss regionally and globally (Bølling et al., 2009; Bond et al., 2013; Huang et al., 2014; Janssen et al., 2011; Rosenfeld, 2006; Saikawa et al., 2009; Shindell et al., 2012).

BB also represents one of the most uncertainties in the emission, climate effect, and public health assessments, which finally relies on the uncertainties in detailed chemical emission factors or related properties and burning activities like strength or percentage of biomass fuel burned (Tian et al., 2008; Andreae et al., 2001; Levin et al., 2010). For example, studies have focused on OC and EC emissions due to their specific optical properties (Bond et al., 2013; Cao et al., 2006; Qin et al., 2012; Ram et al., 2011). OC like sulfate and nitrate can cool the atmosphere by increasing the Earth's reflectivity, however, smoke OC on the other hands together with brown carbon have been found to be a significant source of light absorption (Chen et al., 2015; Ackerman, 2000; Chakrabarty et al., 2010; Christopher et al., 2000). The coated or internal mixed sulfate or nitrate can act as lens to enhance the light absorption activity of BC (Zhang et al., 2008b), probably also the activity of brown carbon (Chen et al., 2015). However, primary emissions for OC, EC, and alkali components are confused and have a wide range (Sen et al., 2014; Cao et al., 2006; Hayashi et al., 2014), and some study still took OC with negative forcing activity (Saikawa et al., 2009; Shindell et al., 2012). Besides, smoke EC is consisting of soot





and char, and soot-EC has a higher light-absorption potential compared to char-EC
(Arora et al., 2015; Reid et al., 2005a). Division and quantification of char- and
soot-EC emissions for biomass burning are understudied (Arora et al., 2015; Han et
al., 2009). However, other components like organic acids, amines, phenols, and
mineral elements that enable CCN activity or endow health hazard of smoke aerosol
are also deficient, variable, or outdated, which may hinder our overall understanding
of biomass burning contributions and also atmospheric process of smoke particles
(Li et al., 2015; Akagi et al., 2011; Chan et al., 2005; Dhammapala et al., 2007; Ge
et al., 2011; Reid et al., 2005b).
Studies using carbon mass-balance method (CMB) and pollutant concentration-
chamber volume quantification method are the two common methods to derive the
emission factors for biomass burning aerosols (Akagi et al., 2011; Li et al., 2007;
Zhang et al., 2008a). Carbonaceous and inorganics components of smoke particles
not only vary with biomass issues (fuel types, water content, or burning strength),
but also relate to burning condition and environment (flaming or smoldering, field
burning or laboratory simulation), extent of aging, sampling methods and
measurement technologies (Grieshop et al., 2009; Hayashi et al., 2014; Reid et al.,
2005b). Comparing to field observations that are closer to the actual burning (Li et
al., 2007; Akagi et al., 2011; Rose et al., 2011; Saffari et al., 2013), laboratory
studies have a definite advantage over field burning research in emission analysis
(Zhang et al., 2008a; Sun et al., 2016; Jayarathne et al., 2014). For example, the
environment, amount of fuel, and burning conditions can be precisely controlled, the
contamination from ambient atmosphere to the emissions can be excluded, and
chemical compositions at different aging extent can be quantified using aerosol
chamber system (Li et al., 2015, 2016; Aurell et al., 2015; Dhammapala et al., 2007).
The activity rates of biomass burning (burning rate of biomass fuels) are also
response to the great uncertainties in the emission estimates (Zhang et al., 2008a;
Sun et al., 2016). Seldom study ever focused on the burning rates, and the limited
data were treated as simplex constant or dynamic values in many studies of emission
estimation in a certain year or for annual variations with a long time scales, thus,



significant difference among the results were founded (Zhang et al., 2011; Qin et al.,
2012; Qin et al., 2011; Zhao et al., 2012). For instance, Cao et al. (2006; 2011)
estimated primary smoke carbonaceous materials emissions for 2000 and 2007 in
China with same field burning rates, the results were almost the same for the two
year with 103-104 Gg $yr^{-1}$ BC and 425.9-433.3 Gg $yr^{-1}$ OC emitted. He et al. (2011b)
found the declining trends in biomass burning emissions in the Pearl River Delta for
the period 2003-2007 based on constant activity data of burning rates. Lu et al. (2011)
developed primary carbonaceous aerosol emissions in China for 1996-2010 with
time-dependent activity rates extrapolated from 2008 to 2010 based on national
fast-track statistic, rapid increase of OC and EC emissions were reported, and OC
increased from 1.5 to 2.3 Tg $yr^{-1}$, BC increased from 418 to 619 Gg $yr^{-1}$. Qin et al.
(2012) estimated BC emission from crop straw open burning for 1980-2009 with
variable burning rates based on peasants' income development, the increasing trend
in BC emission was also confirmed, and BC emission increased from 4.3 to 116.6
Gg $yr^{-1}$.
As most anthropogenic pollutants are concentrated in submicron particulate
matters ($PM_{1.0}$) (Ripoll et al., 2015), more pronounced relationship of ambient $PM_{1.0}$
to haze formation and adverse health effect has been reported (Huang et al., 2003;
Roemer et al., 2001; Shi et al., 2014). Nevertheless, associated chemical
characterization of $PM_{1.0}$ is still undefined (Li et al., 2015; Safai et al., 2013; Cheng
et al., 2006). The study of source-specific $PM_{1.0}$ chemical compositions and
emissions are necessary to replenish database for contribution assessment and model
application in atmospheric chemistry, climate changes, and public health evaluation.
The emission inventories and forecasting in the emissions of atmospheric
pollutants have been widely studied, and the incurred mortality, climatic effect, and
economic loss have also been estimated (Ostro et al., 1998; Saikawa et al., 2009;
Shindell et al., 2012), based on which the emission control policies were proposed.
Shindell et al. (2012) considered ~400 control measures in tropospheric BC and $O_3$
emissions for the benefit of global or regional human health and food security, and
14 optimal measures targeting $CH_4$ and BC emissions were identified. Saikawa et al.





(2009) compared different scenarios of OC, EC, and sulfate emissions in China in
2030, concluding that maximum feasible reduction may avoid over 480000
premature deaths in China and decrease the radiative force from -97 to -15 mW m$^{-2}$
globally. Wang et al. (2008) reported field burning restriction may save about 5
billion dollars losses from biological resource and air pollution. However, the
generalized strategies in emission reduction were inadequate and lack actual
practicality (Streets, 2007; Lin et al., 2010).

In this study, burning experiments with five major agricultural straws were

conducted using a combustion stove in combination with an aerosol chamber system.
Accurate compositions and emission factors for SPM in PM$_{1.0}$ and PM$_{2.5}$ were
characterized and established. Afterwards, up-to-date emissions for agricultural open
burning aerosol in 2012 were developed, health and health-related economic impacts
from smoke PM$_{2.5}$ exposure were also assessed. Finally, emission reduction strategy
that was implemented in field burning rate control for the carcinogenic risk concern
was proposed, which should help establish the policy and provide an idea for the
emission control.
**2 Methodology**
An overview of the research procedures including emission factors acquirement and
emission inventory calculation is shown in Fig. 1. Tabulation of emission factors is
self-established in our laboratory using a combustion stove to simulate open burning
and an aerosol chamber to quantify the emissions. Then, we use a bottom-up
approach to calculate the emission inventory of agricultural field burning over China
mainland based on crop production data in 2012. Emissions for each species are
estimated as:
$E_{k,j} = \sum_i A_{k,i} \times EF_{i,j}$                              (1)
where $E_j$ is emission, $A_{k,i}$ is effective biofuel consumption, and $EF_{i,j}$ is emission
factor. k, i, and j indicates region, agricultural residue type, and particulate chemical
species.

State-of-the art chemical transport and box models were commonly applied to



reproduce or simulate the ambient aerosol concentrations (Ram et al., 2011; Reddy
et al., 2000; Saikawa et al., 2009). In this study, spatio-temporal dynamic box model
is used to calculate the emission flux concentration. Regional crop straws are
premised to be combusted proportionally only in the fire occurrence days.
Dismissing interaction of emitted pollutants in space and time, pollutants will
distribute uniformly in a space covering an area of specific region with mixing
height of 0.5 km (atmospheric boundary layer). The flux concentration of
agricultural burning smoke can be calculated by Eq. (2):
$$C_{k,j} = \frac{E_{k,j}}{S_k \times h \times T_k} \qquad (2)$$
in Eq. (2), $C_{k,j}$ is flux concentration of smoke aerosol, $S_k$ is regional area, h is
boundary layer height, $T_k$ is agricultural field fire duration time.

**2.1 Aerosol chamber work and emission factors**

**2.1.1 Crop straws**

Five kinds of representative crop residues were used for the burning experiments, i.e.,
wheat, rice, corn, cotton, and soybean straws. The straws were collected based on
regional features of agricultural planting, winter wheat straws were collected from
Anhui province, late rice straws from Shanghai, corn straws from Henan province,
cotton and soybean residues from Xinjiang. All straws were stored under dark, airy,
and cooling condition. Prior to the burning experiment, the dirt and weeds were
removed, then straws were dehydrated (at 100 $^{o}$C for 24 h) to minimize effect of the
water content on the burning and pollutant emissions, as study found pollutants
emissions and combustion efficiencies (CE) are response to water content, increased
moisture content enhances the emissions but also alter the chemical compositions of
smoke aerosols (Reid et al., 2005b; Aurell et al., 2015; Hayashi et al., 2014).
Although straws in the field are not well dried and moisture contents vary with
weather, ventilation, and storing times, for the convenience of practical application
and comparison of burnings and emissions, water contents of the straws were
controlled within 2 %. The dry straws were then cut to a length of approximately 10





cm and weighted 10.0 g per serving.
**2.1.2 Burning experiments**
The experiments were conducted using an aerosol chamber system (Fig. S1 in
supplement information, SI), which was loaded in a temperature-controlled room
(18-22 $^{o}$C, 40%-60% RH). A stainless combustion stove was self-deigned to simulate
typical field burning of crop straws, automatic ignition with LPG (Liquid petroleum
gas) in particular, albeit on a small scale (ignition time less than 0.1 s). 10.0 g
conditioned residues were sealed in the 0.227 m$^3$ combustion stove in advance, once
ignited, the force-ventilation and HEPA filtrated particle-free air were supplied (300
L min$^{-1}$). The emissions were immediately injected into a clean, evacuated aerosol
chamber. The burning last about 1 min and over 1 m$^3$ particle-free air flushed the
stove to ensure all the emissions were transferred into the chamber.
The chamber was costume-built to quantify the emissions and characterize the
physiochemical properties of smoke aerosols, detailed description of the chamber
can be found elsewhere (Zhang et al., 2008a; Zhang et al., 2011; Li et al., 2015,
2016). Briefly, the chamber has a volume of 4.5 m³ with 0.3 mm Teflon coating on
the inner side, a magnetic fan fixed on the bottom to stir the aerosol uniformly, and a
hygroclip monitor (Rotronic, Model IM-4) equipped inside to measure the
temperature and relative humidity. Before experiment, the chamber was flushed with
particle-free air for 6 h, oxidized by high concentration ozone (~3 ppm) for 12 h,
then flushed again, filled with pure dry air to 80 KPa, and connected to the stove
finally. The emissions from straw burning were aspirated into the chamber till room
pressure, afterwards, size measurement and chemical samplings were conducted
from the chamber. For each type of straw, four burning experiments were conducted.
The unburned residues were weighted and deducted from 10.0 g after each test.
Modified combustion efficiency (MCE) for each burning was monitored. A
gas-chromatograph (GC, model 930, Shanghai, Hai Xin Gas Chromatograph Co.,
LTD) equipped with a flame ionization detector, an Ni-H convertor, and a stainless
steel column (2 m long) packed with 15% DNP was used to measure CO and $CO_2$





concentrations in the chamber. And MCE were 0.89-0.96 for all the experiments,
indicating flaming combustion dominated, which were comparable to that in the
field burning (Li et al., 2003; Li et al., 2007).

### 2.1.3 Size and morphology of smoke aerosol

Size distribution (10 nm-10 µm) of smoke particles was measured using Wide-range
Particle Spectrometer (WPS, Model 1000XP, TSI, USA), which has been described
by Zhang et al (2011). Briefly, WPS integrates the function of scan mobility particle
sizer (SMPS) and laser particle sizer (LPS), 0.3 L min$^{-1}$ flow is introduced to SMPS
part to classify mobility size from 10 nm to 500 nm, and 0.7 L min$^{-1}$ flow is
introduced to LPS part to measure aerodynamic diameter from 350 nm to 10 µm.
Particle density and refractive index are set as 1.0 g cm$^{-3}$ and 1.45. A diffusion dryer
tube filled with descant-silica gel is set prior to the inlet of WPS. Before experiment,
WPS was calibrated with certified polystyrene latex spheres (PSL, 40, 80, and 220
nm, Duke Scientific).
SPM from the 5 types crop straws burning were sampled onto copper grids coated
with carbon film (carbon type-B, 300-mesh copper, Tianld Co., China) using a
single-stage cascade impactor with a 0.5 mm diameter jet nozzle at a flow rate of 1.0
L min$^{-1}$. The sampler has a collection efficiency of 100 % at 0.5 µm aerodynamic
diameter. More information can be found elsewhere (Fu et al., 2012; Hu et al., 2015).
Then, a JEOL-2010F field emission high-resolution transmission electron
microscope (FE-HRTEM) coupled with an oxford energy-dispersive X-ray spectrum
(EDX) was applied to investigate the morphology, composition, and mixing state of
individual particles.

### 2.1.4 Chemical sampling and analysis

PM$_{1.0}$ and PM$_{2.5}$ samples for each burning were collected on 90 mm quartz filter
(Tissuquartz, Pall Corp., USA) from the chamber using a high-volume Particle
Sampler (HY-100, Qingdao Hengyuan S.T. Development Co., Ltd) operating at 100
L min$^{-1}$. Each filter sampling duration time is 5 min, and total 44 samples (including





4 blank samples) were gathered for the experiments. The quartz microfiber filters
were prebaked for 8 h at 450 $^o$C to eliminate contamination. Before and after the
sampling, the filters were weighted using a balance (Sartorius BP211D) with an
accuracy of 10 μg, and the balance was treated in an electronic desiccator (40 % RH,
22 $^o$C) for 24 h before its use. After weighting, the loaded filters were stored at -20
$^o$C in a refrigerator for further analysis.
Water soluble species including general inorganic ions (ions: $F^-$, $Cl^-$, $NO_2^-$, $NO_3^-$,
$SO_4^{2-}$, $Na^+$, $NH_4^+$, $K^+$, $Ca^{2+}$, $Mg^{2+}$), organic acids ($CH_3COOH$, $HCOOH$, $C_2H_2O_4$,
$CH_3SO_3H$), and seven protonated amines ($MeOH^+$, $TeOH^+$, $MMAH^+$, $DMAH^+$,
$TMAH^+$, $MEAH^+$, and $DEAH^+$ for short, corresponding to monoethanolaminium,
triethanolaminium, monomethylaminium, dimethylaminium, triethylaminium,
monoethylaminium, and diethylaminium ) were measured from 1/4 of each filter
with ion chromatography (IC, Model 850 Professional IC, Metrohm, USA) consists
of a separation column (Metrosep A Supp 7 250/4.0 for anion and organic acids,
Metrosep C-4 150/4.0 for cation, and Metrosep C4-250/4.0 for water soluble
aminiums).
1/4 of each filter was acid dissolved to measure the selected elements (As, Pb, Cr,
Cd, Ni, V, Zn, Al), of which As, Zn, Pb, Cr, Cd, and Ni are USEPA priority
controlled pollutants (Wu et al., 2011). The smashed filters were digested at 170 ℃
for 4 h in high-pressure Teflon digestion vessel with 3 mL concentrated $HNO_3$, 1 mL
concentrated $HClO_4$, and 1 mL concentrated HF. Afterwards, the almost dry solution
was diluted and characterized using Inductively Coupled Plasma Optical Emission
Spectrometer (ICP-OES, Atom Scan 2000, JarroU-Ash, USA).
Another 1/4 of each filter was ultrasonically extracted with $CH_2Cl_2$. The extracts
were then condensed with rotary evaporator. 16 targeted PAHs (2-ring, naphthalene
(Nap); 3-ring, acenaphthylene (Ac), acenaphthene (Ace), fluorene (Fl), phenanthrene
(Phe), anthracene (Ant); 4-ring, fluoranthene (Flu), pyrene (Pyr), benzo[a]anthracene
(BaA), chrysene (Chr); 5-ring, benzo[b]fluoranthene (BbF), benzo[k]fluoranthene
(BkF), benzo[a]pyrene (BaP), dibenzo[a,h]anthracene (DBA); and 6-ring:
indeno[1,2,3-cd] pyrene (IP), benzo[ghi]perylene (BghiP)) and 5 selected phenols





(phenol, 2-methoxyphenol, 4-ethylphenol, 4-ethyl-2-methoxyphenol,
2,6-dimethoxyphenol) were measured from those concentrated extracts using gas
chromatography-mass spectrometer (GC-MS, Agilent 6890-5973N) equipped with
column DB-5ms (Agilent 123-5532).
Organic carbon (OC) and elemental carbon (EC) were measured with the rest
quartz filters using a carbon analyzer (Sunset laboratory Inc., Forest Grove, OR)
based on the thermal-optical transmittance (TOT) method with a modified
NIOSH-5040 (National Institute of Occupational Safety and Health) protocol. Four
organic fractions (OC1, OC2, OC3, and OC4 at 150, 250, 450, and 550 $^{o}$C,
respectively), PC fraction (a pyrolyzed carbonaceous component determined when
transmitted laser returned to its original intensity after the sample was exposed to
oxygen), and three EC fractions (EC1, EC2, and EC3 at 550, 700, and 800 $^{o}$C,
respectively) are produced. And OC is technically defined as OC1 + OC2 + OC3 +
OC4 + PC, while EC is defined as EC1 + EC2 + EC3 - PC (Seinfeld et al., 2012).
Han et al (2007; 2009) furtherly differentiated char-EC and soot-EC from EC
measurement as EC2 + EC3 equals to soot-EC, and the rest is char-EC.
The quality of the data above was guaranteed by standard materials calibration,
recovery rate, and operational blank correction.
**2.1.5 Calculation of emission factors**
The emission quantities derived from the experiment were converted into quantities
per unit weight of initial residues as emission factor (EF, unit: g kg-1), which can be
calculated from effective filter sampling weight, chamber volume, and amount of
crop straw consumed (Dhammapala et al., 2007; Zhang et al., 2008a). To be more
accurate, wall loss and makeup air dilution of smoke particles in the chamber during
sampling should be corrected, and details see in SI.
**2.2 Emission inventory calculation**
**2.2.1 Agricultural field fire survey**
Fire sites over China from 2011 to 2013 were statistically analyzed, and the data was



collected from the Ministry of Environmental Protection of China that obtained by
NASA's Terra and Aqua satellites remote sensing (http://www.mep.gov.cn/).
Agricultural fire sites were screened out from MODIS daily fire products (1 km × 1
km resolution level 3 hotspot) using a high resolution real time land use based on
geography information system (GIS). Spatial and temporal distributions of fire sites
were displayed in Fig. S2 (SI), over 5000 fire sites were allocated into two
prominent filed burning periods corresponding to summer (May to July) and autumn
(September to November) harvests, and filed burning lasts 54 days and 60 days on
statistical average during the two harvests. In the North of China, open burning
occurred primarily in autumn, while temporal-character of field fires was not
significant in the North Plain and the Center of China.
**2.2.2 Crop straw production**
Corp straw production was generally derived from annul or monthly crop production
by multiplying crop-specific ratios of production-to-residue (He et al., 2011b; Cao et
al., 2011; Zhao et al., 2012). In this study, crop productions were furtherly classified
into summer harvest and autumn harvest productions according to field fire sites
analysis and traditional seasonal planting and harvesting. The amount of straw
produced was calculated by Eq. (3):
$M_{t,k,i} = P_{t,k,i} \times r_i \times H_{t,k,i} \times D_i$ (3)
in which M is mass of crop straws produced; P is annual crop-specific amount of
crop production; r is the production-to-residue ratio; D is the dry matter
content; $H_{t,k,i}$ is production ratio of crop i at region k during summer or autumn
harvest period t.
Province-level crop production data of wheat, rice, corn, cotton, and soybean were
taken directly from the China Yearbook 2013 (National Bureau of Statistics of China,
NBSC, 2013). Crop-specific production-to-residue ratios were cited from Chinese
Association of Rural Energy Industry (CAREI, 2000; Wang et al., 2008; data
available at http://www.carei.org.cn/index.php, in Chinese). Dry matter contents of
crop straws were referred to He et al. (2011b) and Greenhouse Gas Inventory



Reference Manual (IPCC, 2007). The parameters of production-to-residue ratios and
dry matter contents were summarized in Table S1 (SI). The regional crop production
ratios in summer and autumn harvests were listed in Table S2 (SI).

### 350    2.2.3 Field burning rate

Uncertainty of emission estimations mostly relies on intangibility of straw open
burning rate (Zhao et al., 2012; He et al., 2011b). However, regional or national
percentage of straw open burned was seldom studied, and the limited data were
outdated and variable. The available studies indicate national filed burning rate of
crop straws range from 15.2% to 27.2% in China (Daize, 2000; Wei et al., 2004;
Zhang et al., 2008a), and more detailed studies indicate about 31.9% of the crop
burned in the Pearl River Delta from 2003 to 2007 (He et al., 2011b), while the
corresponding figures were almost 100% for the Huabei region in 2003 (Zhao et al.,
2012). Two versions of province-level field burning rates that commonly used were
reported by Cao et al. and Wang et al. Cao et al. (2006; 2011; 2005; Chen et al., 2001)
deduced the rates based on regional economic level, the proposal of the rates to be
proportional to peasants' income was confirmed later, and the rates was first used to
calculate the open burning emission in 2000. Wang and Zhang (2008) obtained
provincial percentage of residue open burnt via filed survey in 2006. Herein, the two
versions were both applied directly into the emission estimation of 2012 and named
as business-as-usual scenarios (BAU, BAU-I from Cao et al. and BAU-II from Wang
and Zhang in specific).
In fact, the burning rates should be dynamic parameters that been influenced by
industrial structure, government policy orientation, or public awareness. With crop
yields increasing and energy consumption structure changes in rural areas, more
straws will be discarded and burned in the field. Nonetheless, rigorous agricultural
fire policy may still suppress the condition worsen as it worked during 2008 for
Beijing Olympics and 2010 for Shanghai Expo (Huang et al., 2013; Cermak et al.,
2009; Wang et al., 2010). Qin et al. (2012; 2011) ever deduced year specific open
burning rates in different zone for the period of 1980-2009 according to their



respective peasant income changes in a certain year on the basis of peasant income
and filed burning rates in 2006. However, the simple linear relationship should be
doubted, as great increase in per capita income after 2006 will surely overestimate
the burning rates. We supposed that the values were inverse proportional to peasants'
agricultural income proportion (AIP), without considering the policy or potential
gain or loss related to agricultural residue treatment. Thus the burning rates
established in 2000 and 2006 from Cao et al. (2005) and Wang and Zhang (2008) can
be converted into that of 2012 based on economic data from equation below:
$$R_{k,2012} = \frac{I_{k,2012}}{AI_{k,2012}} \times \frac{AI_{k,y}}{I_{k,y}} \times R_{k,y} \qquad (4)$$

where R is agricultural straw filed burnt rate, $I_{k,y}$ is peasants' annual income, $AI_{k,y}$
is peasants' annual agricultural income. y indicates reference year (2000 for BAU-I,
and 2006 for BAU-II). $I_{k,y}$ and $AI_{k,y}$ can be found or calculated from China
Yearbook and China Rural Statistic Yearbook (NBSC, 2004-2013).
The versions of converted rates based on primary industry level were called
Economic Models I and II (EM-I and EM-II in short) corresponding to BAU-I and
BAU-II. Besides, in 2013, the National Development and Reform Commission of
China published the Chinese agricultural straw treatment report of 2012 (NDRC,
[2014] No.516, data available at http://www.sdpc.gov.cn/, in Chinese) for the first
time. The percentages of crop residues discarded in the report were applied in our
estimation, which was called NDRC version.
**2.2.4 Emission and flux concentration**
From above study, emission of SPM pollutants can be calculated by recount of Eq.
(1), as Eq. (5) showed below:
$$E_{t,k,j} = \sum_i M_{t,k,i} \times R_k \times f_i \times EF_{i,j} \qquad (5)$$

where $E_{t,k,j}$ is emission amount of chemical species j at region k during harvest
period t; $f_i$ is burning efficiency, the crop specific values were cited as 0.68 for
soybean residue and 0.93 for the rest four straws (Zhang et al., 2011; Wang and





Zhang, 2008; Zhang et al., 2008a; Koopmans et al., 1997). Thus, flux concentration
of corresponded pollutants can be also assessed from box model as mentioned in
front.

### 2.3 Estimating health impacts and health-related economic losses

### 2.3.1 Carcinogenic risk of Smoke Particulate Matter (CRSPM)

Apart from the enormous climatic effects due to optical properties of smoke particle
from IPCC, new epidemiological and toxicological evidence have also linked
carbonaceous aerosol to cardiovascular and respiratory health effects according to
the World Health Organization (Bruce et al., 1987; IPCC, 2007). Here, we present
the fuel-specific carcinogenic risk of SPM (CRSPM, unit: per $\mu g\ m^{-3}$) to assess
health hazard from agricultural straw burning particles and help source-specific air
quality control. The cancer risk attributed to inhalation exposures of smoke $PM_{2.5}$
from crop straw i burning was calculated as:
$$CR_i = \sum_j f_j \times UnitRisk_j \qquad (6)$$
where $f_j$ is mass fraction of individual species j in smoke $PM_{2.5}$, $UnitRisk_j$ is
corresponded unit carcinogenic risk value of species j extracted from database
provided by the Integrated Risk Information System (IRIS), California
Environmental Protection Agency (CEPA), and related documents (Bruce et al.,
1987; Burkart et al., 2013; Tsai et al., 2001; Wu et al., 2011; Wu et al., 2009).
$CR_i$ is estimated based on dose addition model of selected hazardous air
pollutants (HAPs) including USEPA priority pollutants of PAHs and heavy metals.
And UnitRisk values of the selected HAPs presented in Table S3 (SI). Synergistic
interactions among pollutants are dismissed, albeit possible. The cancer risk of
chromium is adjusted by multiplying a factor of 0.2, assuming that only 20% Cr
measured is in the toxic hexavalent form (Bell et al., 1997). Benzo[a]pyrene (BaP) is
used as an indicator compound of carcinogenicity, legally binding threshold of BaP
in most countries ranges from 0.7 to 1.3 ng $m^{-3}$, corresponded carcinogenic risk of
BaP is about $1.1 \times 10^{-6}$ per ng $m^{-3}$ (Bruce et al., 1987; Burkart et al., 2013). Thus, one



in million level of carcinogenic potential is frequently used to identify risks of
concern in public health and environmental decision making, and permissible
exposure limits (PEL, unit: µg m$^{-3}$) of crop straw burning particles can be estimated
as:
$PEL_i = \frac{10^{-6}}{CR_i}$                               (7)

### 2.3.2 Human exposure and health impacts

Robust relationship between surface $PM_{2.5}$ and health effects has been revealed and
confirmed by many studies (Pope et al., 2004; Wong et al., 2008). $PM_{2.5}$-related
health endpoints are composed of a range of elements from sub-clinical effects to the
onset of diseases and the final death (Davidson et al., 2005). In this study, incidence
of commonly studied endpoints like premature mortality, respiratory and
cardiovascular hospital admissions, and chronic bronchitis from primary emitted
smoke $PM_{2.5}$ short-term exposure were assessed using the Poisson regression model,
shown as below (Guttikunda et al., 2014):
$\Delta E = \Delta Pop \times IR \times (1 - \frac{1}{e^{\beta \times \Delta C}})$                       (8)
where $\Delta E$ represents the number of estimated cases of mortality and morbidity, $\Delta C$
is the incremental concentration of particulate matter or flux concentration; $\Delta Pop$ is
the population exposed to the incremental particulate concentration of $\Delta C$; IR is
short for incidence rate of the mortality and morbidity endpoints, and $\beta$ is the
coefficient of exposure-response function, defined as the change in number case per
unit change in concentration per capita.
Concentration-response function and incidence rate of each health endpoint are
important in health impacts evaluation and they have variation for different
population and regions (Yang et al., 2012; Wong et al., 2008). Here, the variance for
sex and ages were neglected. Region-specific exposure-response coefficients for
individual mortality were summarized from previous studies, as presented in Table
S4 (SI). The coefficients for individual respiratory and cardiovascular hospital
admission, and chronic bronchitis were cited as 1.2 %, 0.7 %, and 4.4 % (per 10 µg



m$^{-3}$, 95% CI) from Aunan and Pan's work (Aunan et al., 2004). This is the case
because seldom studies ever confirmed these topics in China. Region-specific
mortality and hospitalization IRs were taken from statistical reports authorized by
National Health and Family Planning Commission of the People's Republic of China
(NHFPC, 2013), and morbidity of chronic bronchitis were defined as 13.8 ‰ based
on the forth national health survey, which was released by the Chinese Ministry of
Health in 2008 (CMH, 2009).
**2.3.3 Economic valuation of the health impacts**
The economic losses of the health impacts associated with smoke PM$_{2.5}$ exposure in
2012 were further evaluated. The amended human capital (AHC) approach was
employed to calculate the unit economic cost of premature mortality. The commonly
applied AHC method uses per capita GDP to measure the value of a statistical year
of life (IBRD and SEPA, 2007) based on Eq. (9). It can be used as a social statement
of the value of avoiding premature mortality and estimates human capital (HC) from
the perspective of entire society, neglecting individual differences (Hou et al., 2012).
$$HC_k = \frac{GDP_k}{POP_k} \times \sum_{i=1}^{\tau} \frac{(1+\alpha)^i}{(1+\gamma)^i} \qquad (9)$$
GDP$_k$ and POP$_k$ are gross domestic production and population of target region k
that were reported in the statistical yearbook in 2012; $\alpha$ and $\gamma$ are economic
parameters referring to national GDP growth rate and social discount rate, which
were 7.7 % and 8.0 % in 2012 from National Bureau of Statistics of China (NBSC,
2013, data available at http://www.stats.gov.cn/tjsj/ndsj/, in Chinese). $\tau$ is the
life-expectancy lost due to aerosol pollution, and 18 year of life was widely applied
(Hou et al., 2012). The annual exchange rate of US dollar to RMB was 6.31 in 2012.
One can deduce the HC values of the provinces, municipalities, and autonomous
regions in the country, and the calculated regional HC values were listed in Table S5
(SI). In this paper, the cost of respiratory, cardiovascular hospital admissions, and
chronic bronchitis were 632.2, 1223.4, and 948.6 US$ per case in 2012, which were
derived from the national health statistical reports (NHFPC, 2013).
The regional and national health-related economic loss from smoke PM$_{2.5}$





exposure can be calculated based on the excess mortality and morbidity multiplied
by the corresponding unit economic values.

**3 Result**

**3.1 Particulate chemical compositions and emission factors**

**3.1.1 Organic carbon and elemental carbon**

An overview of particulate chemical compositions for smoke $PM_{2.5}$ and $PM_{1.0}$ is
pie-graphically profiled in Fig. 2, and the corresponded emission factors are given in
Table 1-4. Significant differences of chemical compositions in size range and fuel
types can be observed, implying the non-uniform mixing and distribution of
particulate pollutants from biomass burning. Emission factors of particulate species
from this study are comparable with that from literature as summarized in Table 5.
EFs of smoke $PM_{2.5}$ and $PM_{1.0}$ were $8.99 \pm 5.55$ and $7.91 \pm 4.67$ g kg$^{-1}$ for the five
kinds of crop straws, and over 70 wt.% of SPM was organic components (OM and
EC), with average of 73.4 wt.% in $PM_{2.5}$ and 71.3 wt.% in $PM_{1.0}$. Organic matter
(OM) was converted from OC by multiplying a factor of 1.3 to account for
noncarbon materials (Li et al., 2007). Due to the technical limitation and ambiguous
artificial boundary, carbon contents of biomass burning particles have vast variability
and uncertainty (Lavanchy et al., 1999; Levin et al., 2010). EFs of EC and OC from
this work agree well with previous study, average EFs of OC were 4.21 and 3.58 g
kg$^{-1}$ in smoke $PM_{2.5}$ and $PM_{1.0}$, and the values for EC were 1.09 and 1.01 g kg$^{-1}$.
Mass ratio of OC/EC is an important parameter to indicate the primary organic
aerosol (OA) emission and secondary organic aerosol (SOA) production. The ratio is
response to burning conditions, source, aging extent, and particle size (Engelhart et
al., 2012; Grieshop et al., 2009). Smoke emitted from smoldering fires is
OC-dominated while flaming combustion produces more EC, and the discrepancy of
OC/EC ratio can be an order of magnitude (Grieshop et al., 2009). SOA production
upon photo-oxidation will increase the OC/EC ratio, and positive relation between
oxidation level of OA and OC/EC ratio was reported (Grieshop et al., 2009). Here,



OC/EC ratio in primary emissions varied from 2.4 to 6.2 under flaming condition,
similar to previous studies (Lewis et al. 2009; Dhammapala et al. 2007; Hayashi et al.
2014; Arora et al. 2015). The ratios were larger in $PM_{2.5}$ with average value of 3.8,
while it was 3.6 in $PM_{1.0}$, indicating more EC resides in $PM_{1.0}$.
EC in smoke particle can be further classified as char-EC and soot-EC based on
the distinct different physiochemical properties and formation mechanisms of soot
and char (Arora et al., 2015; Lin et al., 2011; Reid et al., 2005a; Richter et al., 2000).
Both char- and soot-EC represent the major light-absorbing fraction of PM; however,
light-absorption potential of soot-EC exceeds char-EC (Arora et al., 2015). Char-EC
can be distinguished as brown carbon, as carbonaceous materials that are optically
between the strongly absorbing soot and non-absorbing organics are operationally
defined as brown carbon (Yang et al., 2009; Andreae et al., 2006). Char-EC is
formed from solid residues during relative low-temperature combustion, while
generation of soot-EC takes place under high-temperature conditions from
recondensation and dihydrogen-carbonization of gaseous materials (Han et al., 2009;
Han et al., 2007). Average EFs of char- and soot-EC in smoke $PM_{2.5}$ were 0.93 $\pm$
0.50 and 0.15 $\pm$ 0.15 g $kg^{-1}$ in this study. Mass ratio of char-EC/soot-EC is a more
effective indicator for source identification and apportionment than OC/EC (Han et
al., 2009; Han et al., 2007). The ratios of char-EC/soot-EC also varied with fuel
types and PM fraction. Similar to OC/EC, char-EC/soot-EC was larger in $PM_{2.5}$ with
average ratio of 7.28, and the ratio was 6.29 in $PM_{1.0}$, the result indicates that
char-EC dominates the EC fraction in SPM and char particle has a larger size than
soot, as soot particle is mainly within several hundred nanometers, while char is
reported primarily to be supermicron particle (Arora et al., 2015; China et al., 2014;
Lin et al., 2011; Wornat et al., 2007). Besides, correlation among the multi-pollutants
was analyzed by relevance matrix as shown in Table S6 (SI), the strong positive
linear relationship ($R^2>0.99$, $p<0.05$) between EC and char-EC also confirms the
reliable source of biomass burning to produce char-EC (Lin et al., 2011; Arora et al.,

2015).





### 3.1.2 Water soluble organic acids

Smoke particles comprise a considerable amount of water soluble organic acids (WSOA), it was 3.35 wt.% in $PM_{2.5}$ and 3.17 wt.% in $PM_{1.0}$ on average, which was in line with previous work that organic acids measured represented less than 5 wt.% of the total smoke aerosol mass load (Falkovich et al., 2005; Gao et al., 2003). Acetic acid followed by methysulfonic acid contributes the most of the measured low molecule weight acids. The sums of EFs of these organic acids ranged from 46.7 to 770.0 mg kg$^{-1}$, and the WSOA were highly correlated with emissions of OC and PM in Table S6 (SI). Study has shown organic acids contribute a significant fraction of both oxygenated volatile organic compounds (OVOCs) in gaseous phase and SOA in particulate phase, the direct emission of particulate organic acids from biomass burning also represents a significant source of precursors for SOA formation, as the low molecular organic acids will evaporate into gas phase or involve in the heterogeneous reaction directly (Takegawa et al., 2007; Veres et al., 2010; Yokelson et al., 2007; Carlton et al., 2006). Moreover, as the significant fraction of water soluble organic carbon, organic acids plays major response to CCN activity of smoke particles, and organic acids coating or mixing can amplify hygroscopic growth of inorganic salts by decreasing the deliquescence RH, enable the particle to be CCN at relative low degree of supersaturation (Falkovich et al., 2005; Ghorai et al., 2014). In the ambient environment, organic acids can enhance atmospheric new particle formation by impairing nucleation barrier (Zhang et al., 2004), besides, particulate organic acids can also mobilize the solubility of mineral species, like iron, altering the chemical process of particles (Cwiertny et al., 2008). And prominent optical properties of organic acids like humic/fulvic substance make them as potential contributors to the global warming (Yang et al., 2009; Andreae et al., 2006).

### 3.1.3 Water soluble aminiums

Interest has been focused on the vital role of amines in particle nucleation-growth process and acidity regulating due to their strong base (Tao et al., 2016; Bzdek et al., 2011; Bzdek et al., 2010). Though ultratrace gaseous amines and particulate





aminiums were on the order of pptv and ng m$^{-3}$, aminium salts exhibit potential
climatic and health effect due to their significant different properties in hygroscopic,
optical and also toxicological (Qiu et al., 2012; Qiu et al., 2011; Samy et al., 2013;
Zheng et al., 2015; Ho et al., 2015; Tao et al., 2016). It ever proposed that biomass
burning is an important emission source of gaseous amines, especially from
smoldering burning, and particulate alkyl amides can be served as biomarkers (Ge et
al., 2011; Ho et al., 2015; Lee et al., 2013; Lobert et al., 1990; Simoneit et al., 2003).
However, seldom study ever quantitatively explored the particulate water soluble
amine salts (WSA) in primary smoke particle emissions (Schade et al., 1995; Ge et
al., 2011). From this study, WSA contributed 4.81 wt.‰ of smoke PM$_{2.5}$ and 4.69 wt.‰
of PM$_{1.0}$ on average, implicating aminium favored to be abundant in fine-mode
smoke particles, especially in PM$_{2.5-1.0}$. DEAH$^+$, TMAH$^+$, TEOH$^+$ and DMAH$^+$ made
up over 80 wt. % of the measured WSA. Fuel-dependence of WSA distribution and
emission were obvious. EFs of WSA ranged from 4.5 to104.8 mg kg$^{-1}$ in smoke
PM$_{2.5}$, the least was from burning of soybean straw and the largest from cotton and
rice straws. We used mass ratio of WSA to NH$_4^+$ to denote the enrichment of
aminium in particulate phase. WSA/NH$_4^+$ in smoke PM$_{1.0}$ and in PM$_{2.5}$ was 0.16 $\pm$
0.03 and 0.18 $\pm$ 0.06.
**3.1.4 PAHs and Phenols**
Atmospheric PAHs are primarily the byproduct of incomplete combustion of
biomass and fossil fuels (Simcik et al., 1999; Galarneau, 2008). Due to their high
degree of bioaccumulation and carcinogenic or mutagenic effect, the sources and
environmental fate of the ubiquitous PAHs have been the subjects of extensive
studies (Santodonato, 1997; Kim et al., 2013). PAHs can involve in photochemical
reaction to form SOA, the process is influenced by gas-to-particle partition and
meteorological conditions. Oxidation may increase the toxicity of PAHs (Arey et al.,
2003; Wang et al., 2011). Biomass burning is one of the main sources of gaseous and
particulate PAHs, which even contributes to about half of total PAHs emissions in
the atmosphere in China (Xu et al., 2006; Zhang et al., 2011). Burning conditions





can significantly influence the emission of PAHs, under the flaming phase in this
study, PAHs contributed 0.46 wt.‰ of smoke $PM_{2.5}$ and 0.28 wt.‰ of $PM_{1.0}$, over 60%
of the total PAHs were associated to respiratory submicron particles. Emission
factors of 16 PAHs in smoke $PM_{2.5}$ ranged from 1.81 to 8.30 mg $kg^{-1}$, which were
consistent with the values from literature (Dhammapala et al., 2007; Lee et al., 2005;
Zhang et al., 2011). Dhammapala et al. also found laboratory simulated burnings
might overestimate the emission factors of PAHs compared with field burnings
(Dhammapala et al., 2007). The distribution of particulate PAHs emission factors
was presented in Fig. 3a. Of the particle bound PAHs, 3~4-rings components were
the primary ones, including Pyr, Ant, Ace, Flu, Phe, and Chr. Concentration ratios of
selected PAHs, namely diagnostic ratios, were usually used to trace the source and
make apportionment of specific pollutions (Yunker et al., 2002; Simcik et al., 1999).
In this work, average Ant/(Ant+Phe), Flu/(Flu+Pyr), BaA/(BaA+Chr), and
IP/(IP+BghiP) ratios of 5 types agricultural residue burning smokes were 0.72, 0.36,
0.47, and 0.58, respectively. There was no significant difference of the ratios in
$PM_{1.0}$ and $PM_{2.5}$. According to previous work, Ant/(Ant+Phe) above 0.1 and
BaA/(BaA+Chr) above 0.35 indicate the dominance of combustion and pyrolytic
sources, Flu/(Flu+Pyr) and IP/(IP+BghiP) ratios greater than 0.50 suggest coal or
biomass burnings dominate (Simcik et al., 1999; Yunker et al., 2002). However,
validation of source apportionment using specific diagnostic ratios should have its
constraints, because of variations in source strengths and atmospheric processing of
PAHs (Arey et al., 2003; Galarneau, 2008).
From Table S6 (SI), The PAHs in smoke particles were highly correlated with EC
and OC contents. PAHs primarily originate from pyrolysis of organic materials
during combustion, and formation mechanisms of PAHs and soot are closely
intertwined in flames. High-molecular-weight PAHs (>500 atomic mass unit) act as
precursors of soot particles (Lima et al., 2005; Richter et al., 2000). Thus, PAHs with
3, 4, and 5 rings accumulate and dominate in the emissions of biomass burning, as
larger molecular weight PAHs tend to incorporate into soot particles. PAHs
expulsion-accumulation in OC and EC fractions were analyzed by linear fitting of



PAHs mass fractions and EC mass fractions in carbonaceous materials (EC+OC) in
Fig. 3b. The partitions can be parameterized as Eq. (10):
$$f_{PAHs} = \frac{m_{PAHs}}{m_{OC}+m_{EC}} = \beta_{EC} \times \frac{m_{EC}}{m_{OC}+m_{EC}} + \beta_{OC} \times \frac{m_{OC}}{m_{OC}+m_{EC}} = \beta_{EC} \times f_{EC} + \beta_{OC} \times f_{OC} \quad (10)$$

where $f_{EC}$ and $f_{OC}$ are the mass fraction of OC and EC in carbonaceous materials
(EC+OC). $\beta_{EC}$ and $\beta_{OC}$ are expulsion-accumulation coefficients of PAHs in OC and
BC. The coefficient of $\beta_{EC}$ is $1.1 \times 10^{-3}$ in smoke $PM_{1.0}$ and $1.9 \times 10^{-3}$ in $PM_{2.5}$; the
corresponded $\beta_{OC}$ is $0.3 \times 10^{-3}$ and $0.5 \times 10^{-3}$.
Phenols are the most common SOA precursor/product and organic pollutants in
the atmosphere (Berndt et al., 2006; Schauer et al., 2001). Hydroxyl functional group
and aromatic benzene ring make phenols a paradigm in heterogeneous reaction upon
photo oxidation research and aqueous phase reaction research. Phenols are also ROS
precursors that present health hazard (Bruce et al., 1987). Phenol and substituted
phenols are thermal products of lignin pyrolysis during biomass burning
(Dhammapala et al., 2007), and the five measured phenols contributed 2.98 wt. ‰
and 2.47 wt. ‰ of $PM_{2.5}$ and $PM_{1.0}$. 2, 6-dimethoxyphenol was the major one of the
measured phenols. Mass fraction of phenols was about 7~9 time of PAHs in smoke
aerosols.
**3.1.5 Inorganic components**
From Fig. 2, smoke particles consisted of approximately 24 wt.% water soluble
inorganics (WSI), and the inorganic salts resided more in $PM_{1.0}$. $K^+$, $NH_4^+$, $Cl^-$, and
$SO_4^{2-}$ were the major inorganic ions in WSI. Particulate enriched $K^+$ together with
levoglucose are treated as tracer of pyrogenic source (Andreae et al., 1998). And
specific mass ratio of $K^+/OC$ or $K^+/EC$ will help make source apportionment of
particulate pollutants with PMF (Positive Matrix Factorization) and PFA (Principle
Balance Analysis) models (Lee et al., 2015). $K^+/OC$ in smoke particles ranged from
0.11 to 0.25 with average value of 0.17 in $PM_{1.0}$ and 0.14 in $PM_{2.5}$, which were
similar to those reported for the Savannah burning and agricultural waste burning
emissions in India and China (Echalar et al., 1995; Ram and Sarin, 2011). However,
OC represents large uncertainty arise from degree of oxidization and burning





condition, $K^+$/EC is more practical parameter to distinguish the pyrogenic pollutants
in ambient study. To smoke particle emitted from flaming fires, K+/EC was 0.58 $\pm$
0.24 in $PM_{1.0}$ and 0.53 $\pm$ 0.18 in $PM_{2.5}$. $Cl^-$ was the main anion to balance the charge
of WSI in smoke particles. Mean charge ratio of $Cl^-$ : $K^+$ was 1.46 and 1.49 in $PM_{1.0}$
and $PM_{2.5}$, with atmospheric aging, the ratio will decrease as chloride be replaced by
secondary sulfate and nitrate (Li et al., 2015; Li et al., 2003). Equivalent charge ratio
of primary cations ($NH_4^+$ + $K^+$) to primary anions ($SO_4^{2-}$ + $Cl^-$) was 1.05 in $PM_{1.0}$
and 1.01 in $PM_{2.5}$ on average, and charge ratios of total cations to anions ($R_{C/A}$) was
1.09 and 1.07 in $PM_{1.0}$ and $PM_{2.5}$. $R_{C/A}$ was used to indicate the neutralizing level of
particulate matters in many studies. $R_{C/A} \geq 1$ indicates most of the acids can be
neutralized, while $R_{C/A} < 1$ means atmospheric ammonia is deficient and the aerosol is
acidic (Adams et al., 1999; He et al., 2011a; Kong et al., 2014). In ambient
environment, acidic aerosol was prevailing urban pollutants in many cities from field
investigation (He et al., 2011a; Kong et al., 2014). Acidic aerosols can increase the
risks to human health and affect the atmospheric chemistry by activating hazardous
materials and promoting the solubility of particulate iron and phosphorus (Amdur et
al., 1989; Meskhidze, 2005). The emission and transport of biomass burning
particles may neutralize the acidity of ambient particles. However, only limited WSI
were brought into in the analytical system, considering the existence of massive
organic acids and ammniums, it is not really to tell the acidity or base of smoke
particles.
Trace mineral elements attracted great attention for the role as catalyst in
atmospheric heterogeneous reaction and health cares (Davidson et al., 2005;
Dentener et al., 1996). Wet/dry deposition of particles during long range transport
will affect the ecological balance by releasing mineral elements (Jickells et al., 2005).
Dust storm, weathering, and industrial process are the main sources of particulate
metals, and incineration can also produce a lot of mineral elements (Moreno et al.,
2013). However, the emissions of trace metals from biomass burning are highly
uncertain (Li et al., 2007), the great influence from local soil environment and soil
heavy metal pollution will certainly affect the metal content in biomass fuel and





smoke particle. In this study, THM resided more in $PM_{2.5}$ than in $PM_{1.0}$. Smoke
$PM_{2.5}$ consisted of 6.7 wt.‰ THM on average, $PM_{1.0}$ comprised 4.1 wt.‰ THM. EFs
of THM in $PM_{2.5}$ and $PM_{1.0}$ were 0.056 $g\,kg^{-1}$ and 0.028 $g\,kg^{-1}$ for all the five straws
burning aerosol. Al contributed over 90% of total THM, and As was the second most
element. Smoke particles from wheat, rice, and corn straws contained more mineral
elements than that from cotton and soybean residues combustion. Regardless the
difference in biomass fuels, the result can imply that soil heavy metal pollution is
heaver in the East China than that in Xinjiang in the West North of China (Wei et al.,

2010).

### 3.2 Size, morphology, and mixing state of smoke particles

Fresh smoke particles exhibited unimodal size distribution within 500 nm (Fig. S3,
SI), and previous chamber study has also confirmed that agricultural fire produces
large amount of ultrafine particles, implying more profound threat to human health
(Araujo et al., 2008; Delfino et al., 2005; Zhang et al., 2011). However, the role of
particles in the atmospheric process and health hazard depends not only size, but also
morphology and chemical mixing states (Dusek et al., 2006; Kennedy, 2007;
Mikhailov et al., 2006; Schlesinger, 1985). From TEM images in Fig. 4, agricultural
straw burning aerosols comprised a broad class of morphological and chemically
heterogeneous particles. Non-uniformly internal mixing of the agglomerates was
noticeable, including the major carbonaceous particles and a considerable amount of
inorganic salt particles, which was consistent with particulate chemical compositions
analysis. KCl particles containing minor sulfate or nitrate were the primary inorganic
particles, which presented crystal or amorphous state from X-ray diffraction analysis
(Fig. 4 a, b, c). And potassium-bearing particles have been used as a tracer of
ambient biomass burning pollutants. Fly ash particles were arresting due to visible
morphology difference and mineral chemical composition (Fig. 4 d, e, f). Fly ash
particles were more compact and rich in mineral elements like Ca, Si, Fe, Al, Mn,
and Cr. Besides, these particles had larger size, statistical average diameter of fly ash
particles obtained from bulk analysis was 2.2 ± 1.6 μm. The result also proved heavy



metals resided more in $PM_{2.5}$ than $PM_{1.0}$. Fly ashes are by products of incineration
process (Buha et al., 2014), including coagulation of fuel issue debris, condensation
of evaporated mineral metal from biomass fuels or adhered dirt at different burning
phase. These fly ashes coated by or agglomerated with carbonaceous materials were
like mash of mineral without clear lattice. Tar ball as a specific form of brown
carbon and soot were representative particles of biomass burning aerosol (Wilson et
al., 2013; Chakrabarty et al., 2010; Tóth et al., 2014). From Fig. 4 g, chain-like soot
particles were coagulated with tar ball. Soot particles were agglomerates of small
roughly spherical elementary carbonaceous particles, these chemical consistent
particles were within 20~30 nm, and high-resolution TEM showed the soot spheres
consisted of concentrically wrapped graphitic layers, while monomeric tar balls
possessed disordered microstructure. Tar balls and soot corresponded to different
stages in the aging of organic particles; tar balls abundant in fresh or slightly aged
biomass smoke are formed by gas-to-particle conversion of high-molecular weight
organic species or from aged primary tar droplets upon biomass burning. Soot
represents further aged carbon-bearing particles, formed from the pyrolysis of lignin,
cellulose, or tar balls (Pósfai, 2004; Tóth et al., 2014). The botryoid aggregates in
Fig. 4 g can be viewed as transformation of tar ball to soot. Tar ball and soot were
also internal mixed with inorganic salt including sulfate and nitrate (Fig. 4 g, h, i),
which made the physiochemical properties of BC even complicated, as study has
confirmed inorganic sulfate mixing will enhance light absorption and hygroscopicity
of BC (Zhang et al., 2008b). Dark-ring like shell of tar ball (Fig. 4 g, h) and spot-like
particles adhered to the surface of tar ball (Fig. 4 i) were K-rich materials. And size
of soot particles was mainly within 200 nm, while tar ball and other carbonaceous
particles can be over one micrometer.

## 3.3 Open burning emissions

### 3.3.1 Crop straw production

The agricultural straw productions were calculated and geographically displayed in
Fig. 5 a-c. Totally 647.3 Tg agricultural straws were produced in 2012 and dispersed



mainly in the North and Northeast of China. The distributions of the straws clearly
correspond to the distinct planting regions that are divided by Qinling
Mountain-Huaihe River line and the Yangtze River. Rice is primarily planted in the
south of Qinling Mountain-Huaihe River line, only 10 % rice (single cropping rice
dominate) is planted in Heilongjiang, Jilin, and Liaoning province, while wheat and
corn are grew mostly in the north of the Yangtze River. Over 90 % of the wheat
planted in China is winter wheat that gets ripe in summer, and more than 80 % rice
including middle and late rice grows mature in autumn. Summer harvest contributed
about 25 % of the agricultural straw production, which solely consists of rice and
wheat straws in this period and distributes in the central and east of China. 493.9 Tg
crop straws were produced mainly from corn and rice harvesting in autumn. Soybean
and cotton straws account for about 8.6 % of autumn straw production that were
primarily produced in Heilongjiang and Xinjiang province.
**3.3.2 Open burning rate**
The five scenarios of field burning rates and regional AIP ($\frac{I_{k,y}}{AI_{k,y}}$) in the year of
2000, 2006, and 2012 were listed in Table 6 and statistically analyzed in Fig. 6. A
significant difference of regional burning rates among the versions was observed,
and the rates from NDRC report were generally higher. For convenience, six zones
were classified by geographic divisions and economic areas in China, including the
North Plain (Anhui, Shandong, Hebei, Shanxi, Tianjin, Beijing), the Central of
China (Hunan, Henan, Hubei), the Yangtze River Delta (Zhejiang, Jiangsu,
Shanghai), the Northeast of China (Heilongjiang, Liaoning, Jilin), the Pan-Pearl
River Delta (Hainan, Guangdong, Fujian, Guangxi, Guizhou, Sichuan, Yunnan,
Jiangxi), the West of China (Shannxi, Chongqing, Xinjiang, Qinghai, Ningxia, Tibet,
Inner Mongolia, Gansu). And the bulk-weighted burning rates that averaged from
BAU, EM, and NDRC versions for the six zones were 22.3 % ± 3.1 %, 21.1 % ±
3.3 %, 28.4 % ± 6.2 %, 23.3 % ± 9.2 % 21.4 % ± 6.5 %, and 14.2 % ± 8.0 %,
respectively. It was obvious that condition of agricultural field burning was most
serious in the Yangtze River Delta, especially in the Zhejiang province. The



nationwide filed burning rate was 21.4 %, 16.3 %, 26.0 %, 14.9 %, and 26.8 % for
BAU-I, BAU-II, EM-I, EM-II, and NDRC, respectively, which were comparable
with the document values (Daize, 2000; Wei et al., 2004; Zhang et al., 2008a).
**3.3.3 Agricultural open burning emissions**
$PM_{2.5}$ emissions from agricultural field burnings based on BAU, EM, and NDRC
versions were calculated and geographically presented in Fig. 7 (emissions of
detailed chemical species in SI). A similar spatial character of regional emission
distribution was observed for BAU, EM, and NDRC versions, most emissions were
allocated in the North Plain and the Central of China where the primary agricultural
regions locate, echoing the agricultural fire sites in Fig. S2 (SI). Although filed
burning rates were higher in the Yangtze River Delta, the crop residue productions of
this zone were much less, which only contributed 4.3 % of the national straw
productions. Take NDRC as the basis, BAU and EM scenarios all underestimated the
emissions in the Northeast of China, especially in Heilongjiang.
The temporal distributions of field burning emissions also echoed the crop residue
productions and the agricultural fire sites in summer and autumn harvest. Apart from
Henan and Tibet where the main crop straws were produced in summer period, more
pollutants were emitted in autumn harvest period to the rest place, which has been
confirmed by many studies (He et al., 2011b; Wang and Zhang, 2008). And the large
scale filed burning emissions in autumn exhibited great influence on the haze
formation and visibility degradation in the North and East of China (Leng et al.,
2014; Shi et al., 2014). In summertime, filed burnings concentrated in the North
Plain, the Central, and the South regions. While in autumn, filed burning emissions
became more ubiquitous and serious in the Northeast of China.
Nationwide emission inventories and flux concentrations were graphically
displayed in Fig. 8 and tabular presented in Table 7. The total $PM_{2.5}$ emission from
agricultural field burnings was 0.74-1.24 Tg in 2012, of which $PM_{1.0}$ was 0.66-1.11
Tg, OC was 0.32-0.53 Tg, and EC was 0.09-0.16 Tg, and the results were
comparable with the precious studies (Cao et al., 2006; Cao et al., 2011; Wang et al.,





2012). Allocated the emissions into the six zones to get the contribution of the North
Plain (21 %) ≥ the Northeast (20 %) > Pan-Pearl River Delta (19 %) ≥ the Central
(19 %) > the West (16 %) >the Yangtze River Delta (5 %). Summertime field
burnings accounted for 20-25 % of national emissions. 24.60 Gg char-EC, 3.79 Gg
soot-EC, 6.82 Gg WSOA, 1.00 Gg WSA, 0.11 Gg PAHs, 0.86 Gg phenol and
substituted phenols, and 2.07 Gg THM on average were released in summer harvest
from agricultural field burning. The corresponded values for autumn harvest were
88.77, 17.21, 18.36, 4.82, 0.37, 1.86, and 6.62 Gg, respectively. Zhang et al. (2011)
estimated particulate PAHs emissions form three types of crop residues to be 0.46
Gg in 2003. Xu et al. (2006) counted PAHs from all straws without considering
burning rates to get 5-10 Gg emissions in 2003.
The nationwide flux concentration of smoke $PM_{2.5}$ was 0.7-1.0 µg m$^{-3}$ d$^{-1}$ in
summer harvest and 1.4-3.5 µg m$^{-3}$ d$^{-1}$ in autumn harvest, while average annual flux
concentrations for OC and EC were 0.80 and 0.25 µg m$^{-3}$ d$^{-1}$. Saikawa et al. (2009)
assessed the annual concentrations of OC and BC from biomass burning primary
emission in China using global models of chemical transport (MOZART-2) to be 1.8
and 0.35µg m$^{-3}$. The most polluted areas were Anhui, Henan, Shandong, Jiangsu,
Liaoning, and Hunan.
**3.3.4 Uncertainties of the emissions**
The fuzziness and uncertainties of major pollutants emissions from fuel combustion
in China came from the uncertainties in specific-source emission factors and
effective consumption of bio- or fossil fuel. Frey et al. analyzed uncertainties in
emission factors and emissions of air toxic pollutants and technology dependent
coal-fire power plants via bootstrap simulation method (Frey et al., 2004; Frey et al.,
2002). Zhao et al. estimated uncertainties in national anthropogenic pollutants
emissions based on Monte Carlo simulation, and they believed activity rates (e.g.
fuel consumption) are not the main source of emissions uncertainties at the national
level (Zhao et al., 2012; Zhao et al., 2011). The uncertainties can also be estimated
by comparing the specific emissions from different studies. With this method, the





uncertainties represent the bias among different copies of emission inventory.

In this study, the bias in smoke PM$_{2.5}$ emissions among BAU, EM, and NDRC

versions was investigated and presented in Table 8. The average national smoke
PM$_{2.5}$ emissions had 19% relative error. More variability of the emissions was in the
West of China (51.4 %), followed by the Northeast (39.8 %), Pan-Pearl River Delta
(25.9 %), and Yangtze River Delta (21.5 %). Although uncertainty was largest in the
West, the contribution of the emissions was much less.
**3.4 Health and health-related economic impacts**
**3.4.1 Carcinogenic risk**
Calculated $CR_{SPM}$ for smoke PM$_{2.5}$ from wheat, corn, rice, cotton, and soybean
straw burning were $5.3 \times 10^{-6}$, $3.8 \times 10^{-6}$, $2.6 \times 10^{-6}$, $0.7 \times 10^{-6}$, and $1.3 \times 10^{-6}$ per µg m$^{-3}$,
respectively. And the corresponded one in million PEL was 0.2, 0.3, 0.4, 1.4, and 0.8
µg m$^{-3}$. Wu et al. (2009) ever assessed unit risk of wood and fuel burning particles
using metals merely, the results were $3.2 \times 10^{-6}$ and $1.5 \times 10^{-6}$ per µg m$^{-3}$, which were
close to that in our study. In actual application, PEL of smoke particles should be
bulk mass concentration of mixed aerosols.

It was noticeable that apart from Tibet and Qinghai, the flux concentration of

smoke PM$_{2.5}$ among all the five emission versions in other regions far surpassed the
PEL, especially the North Plain and the Central of China, exhibiting great potential
inhalable cancer risk. For the health care, emission flux concentration should be
constrained within the PEL of crop straw burning aerosol. Thus the critical filed
burning rates can be derived to ensure risk aversion following Eq. (11):
$$R_k \leq \frac{10^{-6} \times S_k \times h \times T_k}{\sum_j \sum_i P_{t,k,i} \times r_i \times H_{t,k,i} \times D_i \times f_i \times EF_{i,j} \times CRF_i} \qquad (11)$$

The conservative values of regional field burning rates from Eq. (11) were named

as Carcinogenic Risk Control scenarios (CRC) and listed in Table S7 (SI), which
would be instructive in emission control. Under CRC, national crop straw field
burning rate was less than 3%, emissions of PM$_{2.5}$ were geographically presented in
Fig. S4 (SI), and 146.3 Gg yr$^{-1}$ smoke PM$_{2.5}$ should be released at largest in China,





the corresponded annual flux concentration of $PM_{2.5}$ was within 0.3 $\mu g\ m^{-3}\ d^{-1}$ (see in
SI).

### 3.4.2 Health impacts

Health impacts from acute exposure of agricultural residue burning aerosol were
assessed using daily flux concentrations of smoke $PM_{2.5}$, the result was tabulated in
Table S8 (SI). The impacts from smoke $PM_{2.5}$ exposure were severest in Jiangsu,
Shandong, and Henan province, where annual premature mortality was over one
thousand. On average, China suffered from 7836 (95% CI: 3232, 12362) premature
death, 31181 (95% CI: 21145, 40881) respiratory hospital admissions, 29520 (95%
CI: 12873, 45602) cardiovascular hospital admissions, and 7267237 (95% CI:
2961487, 1130784) chronic bronchitis related to agricultural fire smoke in 2012 from
Table 9. According to national health statistical reports (NHFPC, 2013), the hospital
admission due to respiratory and cardiovascular disease was 5071523 in China in
2012, and smoke $PM_{2.5}$ exposure might contribute ~1.2% of the hospital admissions
from this study. Saikawa et al. (2009) ever reported 70000 premature deaths in China
and an additional 30000 deaths globally due to OC, EC, and sulfate exposure that
were primarily emitted from biofuel combustion in China in 2000, however, the
results should be overestimated not only in the exaggerated pollutant emissions but
also in the iterative operations of respective species induced mortality, besides, the
exposure-response coefficient β and incidence rate he applied from Pope et al. (2002)
and WHO (2000) were higher than the practical values from local research (Cao et
al., 2012; Chen et al., 2011; Hou et al., 2012). From Table 9, under CRC version,
over 92 % mortality and morbidity can be avoided.

### 3.4.3 Health-related economic losses

Health-related total economic losses from straw open burning smoke $PM_{2.5}$ exposure
were assessed to be 8822.4 (95% CI: 3574.4, 13034.2) million US$ on average from
Table 10, accounting for 0.1 % of the total GDP in 2012, and detailed regional
economic losses were listed in Table S9. Economic losses from premature death





contributed about 17% of total losses, and loss from chronic bronchitis dominated.
Hou et al. (2012) ever estimated 106.5 billion US\$ lost due to ambient $PM_{10}$
exposure in China in 2009; even a severe haze episode ($PM_{2.5}$ be focused on) in
January 2013 may cause 690 premature death and 253.8 million US\$ loss in Beijing,
and source-specification analysis stressed the emission from biomass burning (Yang
et al., 2015; Gao et al., 2015). It was obvious that smoke $PM_{2.5}$ contributed a
noticeable damage to public health and social welfare. According to CRC version
estimation, the carcinogenic risk control policy can save over 92 % of the economic
loss.

**4 Conclusion**

Detailed chemical compositions of smoke aerosol from five major agricultural
straws burning were characterized using an aerosol chamber system. And
corresponded emission factors for particulate OC-EC, char-/soot-EC, WSI, WSOA,
WSA, PAHs, Phenols, and THM in smoke $PM_{2.5}$ and $PM_{1.0}$ were established.
Permissible exposure limits (PEL) of the smoke particles were assessed for
carcinogenic risk concern based on selected hazard pollutants including PAHs and
THM in smoke $PM_{2.5}$. Daily exposure concentration should be constrained within
0.2, 0.3, 0.4, 1.4, and 0.8 µg m$^{-3}$ for wheat, corn, rice, cotton, and soybean straw,
respectively.
Emission inventories of primary particulate pollutants from agricultural field
burning in 2012 were estimated based on BAU-I, BAU-II, EM-I, EM-II, and NDRC
scenarios, which were further allocated into different regions at summer and autumn
open burning periods. The estimated total emissions were 0.74-1.24 Tg $PM_{2.5}$,
0.66-1.11 Tg $PM_{1.0}$, 0.32-0.53 Tg OC, 0.09-0.16 Tg EC, 0.08-0.14 Tg char-EC,
0.02-0.03 Tg soot-EC, 18.77-30.82 Gg WSOA, 4.23-7.19 Gg WSA, 0.35-0.59 Gg
PAHs, 2.02-3.40 Gg Phenols, and 6.36-10.64 Gg THM, respectively. The spatial and
temporal distributions of the five versions have similar characters that echo to the
agricultural fires sites from satellite remote sensing. Less than 25 % of the emissions
were released from summer field burnings that were mainly contributed by the North



Plain and the Central of China. Large uncertainties of the emissions were found in
the West and the Northeast of China (59.4% and 39.8% relative error). Flux
concentrations of annual smoke $PM_{2.5}$ that were calculated using box-model method
based on five versions all exceed the PEL. From assessment of health impacts and
health-related economic losses due to smoke $PM_{2.5}$ short-term exposure, China
suffered from 7836 (95%CI: 3232, 12362) premature mortality and 7267237 (95%
CI: 2961487, 1130784) chronic bronchitis in 2012, which led to 8822.4 (95%CI:
3574.4, 13034.2) million US$, or 0.1 % of the total GDP losses.
Percentage of open burned crop straws at post-harvest period should cut down to
less than 3% to ensure risk aversion from carcinogenicity, especially the North Plain
and the Northeast, where the emissions should decease at least by 94% to meet the
PEL. And by applying such emission control policy, over 92% of the mortality and
morbidity attributed to agricultural fire smoke $PM_{2.5}$ can be avoided in China.
**Supplementary material related to this article is available online at:**
*Acknowledgment*. This work is supported by National Natural Science Foundation of
China (No. 21190053, 21177025), Cyrus Tang Foundation (No. CTF-FD2014001),
Shanghai Science and Technology Commission of Shanghai Municipality (No.
13XD1400700, 12DJ1400100), Priority fields for Ph.D. Programs Foundation of
Ministry of Education of China (No. 20110071130003) and Strategic Priority
Research Program of the Chinese Academy of Sciences (Grant No. XDB05010200).

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



**Tables and figure captions**

**Table 1.** Emission factors of particulate chemical species in smoke $PM_{2.5}$ from agricultural residue burning.

**Table 2.** Emission factors of particulate chemical species in smoke $PM_{1.0}$ from agricultural residue burning.

**Table 3.** Emission factors of particulate THM, PAHs, and Phenols in smoke $PM_{2.5}$ from agricultural residue burning.

**Table 4.** Emission factors of particulate THM, PAHs, and Phenols in smoke $PM_{1.0}$ from agricultural residue burning.

**Table 5.** Comparison of emission factors with literature (specific chemical materials in form of $PM_{2.5}$)

**Table 6.** Summary of field burning rates and economic data in China

**Table 7.** National agricultural field burning emissions of BAU, EM, NDRC, and CRC scenarios in China, 2012.

**Table 8.** Uncertainties in emission estimates.

**Table 9.** Estimated number of cases (95% CI) attributable to agricultural fire smoke $PM_{2.5}$ exposure in China, 2012

**Table 10**. Health-related economic loss (95% CI) from agricultural fire smoke $PM_{2.5}$ exposure in China, 2012

**Figure 1.** Schematic methodology for developing emission estimations

**Figure 2.** Chemical profiles of smoke $PM_{2.5}$ and $PM_{1.0}$ from 5 types agricultural residue burnings. OM (organic matter = 1.3×OC). OWSI, other water soluble ions including $F^-$, $NO_2^-$, $Na^+$, $Ca^{2+}$, $Mg^{2+}$. THM, trace heavy metals. WSA, water-soluble amine salts. WSOA, water-soluble organic acids.

**Figure 3.** a) Emission factors of 16 USEPA priority PAHs in smoke $PM_{2.5}$ and $PM_{1.0}$; b) expulsion-accumulation of PAHs in OC-EC of smoke $PM_{2.5}$ and $PM_{1.0}$

**Figure 4.** Transmission electron microscope (TEM) images and EDX analysis of fresh agricultural residue burning particles. (a)-(c) Crystal and amorphous KCl particles internally mixed with sulfate, nitrate, and carbonaceous materials. (d)-(f) Heavy metal-bearing fractal-like fly ash particles. (e)-(g) Chain-like soot particles and tar ball.

**Figure 5.** Annual agricultural residue production of five major crops and allocated





into two harvest (summer and autumn harvest) based on agricultural yield in China,

1502 2012.

**Figure 6.** Statistical analysis of field burning rates from BAU, EM, and NDRC
versions
**Figure 7.** Spatial and temporal distribution of smoke $PM_{2.5}$ emissions and flux
concentrations from agricultural field burning over China, 2012
**Figure 8.** Nationwide $PM_{2.5}$ emissions and flux concentrations based on different
burning versions. The inset pie-graphs are chemical compositions of integrated $PM_{2.5}$
from five major agricultural residue burning.



**Table 1.** Emission factors of particulate chemical species in smoke $PM_{2.5}$ from
agricultural residue burning.

| Chemical Species (g kg$^{-1}$) | wheat straw | corn straw | rice straw | cotton residue | soybean residue |
|---|---|---|---|---|---|
| $PM_{2.5}$ | 5.803 ±0.363 | 5.988 ±0.723 | 14.732 ±2.417 | 15.162 ±2.053 | 3.249 ±0.350 |
| OC | 2.813 ±0.147 | 2.393 ±0.351 | 6.882 ±0.689 | 7.415 ±0.547 | 1.539 ±0.253 |
| EC | 0.676 ±0.027 | 0.778 ±0.152 | 2.182 ±0.278 | 1.192 ±0.171 | 0.614 ±0.190 |
| Char-EC | 0.606 ±0.024 | 0.667 ±0.132 | 1.761 ±0.166 | 1.072 ±0.154 | 0.564 ±0.177 |
| Soot-EC | 0.069 ±0.007 | 0.110 ±0.043 | 0.421 ±0.061 | 0.120 ±0.034 | 0.051 ±0.031 |
| **Inorganic ions (g kg$^{-1}$)** | **1.273 ±0.072** | **1.810 ±0.030** | **3.086 ±0.266** | **3.810 ±0.246** | **0.523 ±0.149** |
| $SO_4^{2-}$ | 0.084 ±0.028 | 0.217 ±0.041 | 0.409 ±0.127 | 0.701 ±0.081 | 0.073 ±0.014 |
| $Cl^-$ | 0.576 ±0.038 | 0.709 ±0.034 | 1.158 ±0.232 | 1.351 ±0.114 | 0.178 ±0.030 |
| $F^-$ | 0.023 ±0.061 | 0.061 ±0.005 | 0.073 ±0.024 | 0.265 ±0.012 | 0.009 ±0.004 |
| $NO_3^-$ | 0.023 ±0.000 | 0.032 ±0.002 | 0.051 ±0.025 | 0.072 ±0.004 | 0.009 ±0.004 |
| $NO_2^-$ | 0.006 ±0.001 | 0.016 ±0.002 | 0.018 ±0.002 | 0.036 ±0.001 | 0.004 ±0.003 |
| $Ca^{2+}$ | 0.030 ±0.011 | 0.036 ±0.003 | 0.046 ±0.007 | 0.060 ±0.003 | 0.010 ±0.002 |
| $Na^+$ | 0.005 ±0.001 | 0.012 ±0.001 | 0.028 ±0.004 | 0.050 ±0.004 | 0.005 ±0.001 |
| $NH_4^+$ | 0.152 ±0.005 | 0.197 ±0.010 | 0.542 ±0.107 | 0.347 ±0.008 | 0.029 ±0.004 |
| $Mg^{2+}$ | 0.005 ±0.000 | 0.017 ±0.002 | 0.023 ±0.004 | 0.032 ±0.002 | 0.005 ±0.001 |
| $K^+$ | 0.368 ±0.041 | 0.514 ±0.009 | 0.739 ±0.049 | 0.947 ±0.070 | 0.200 ±0.023 |
| **Organic Acids (mg kg$^{-1}$)** | **156.680 ±81.830** | **46.670 ±9.000** | **557.130 ±269.380** | **769.990 ±317.550** | **143.310 ±39.770** |
| $CH_3COOH$ | 148.900 ±79.290 | 36.640 ±8.210 | 417.930 ±186.140 | 743.320 ±159.600 | 135.500 ±62.320 |
| MSA | 7.170 ±2.110 | 10.030 ±30.000 | 136.990 ±81.700 | 12.980 ±1.530 | 3.200 ±1.530 |
| $H_2C_2O_4$ | 2.610 ±0.430 | ND | 2.210 ±1.560 | 4.760 ±2.640 | 2.170 ±2.380 |
| HCOOH | ND | ND | ND | 8.930 ±2.630 | 2.440 ±1.450 |
| **Amine salts (mg kg$^{-1}$)** | **19.246 ±9.368** | **32.877 ±19.141** | **104.787 ±15.635** | **102.409 ±13.379** | **4.514 ±1.776** |
| $MeOH^+ + MMAH^+$ | 1.322 ±0.086 | 5.735 ±0.102 | 17.226 ±1.454 | 19.888 ±0.351 | 0.456 ±0.196 |
| $MEAH^+$ | 0.201 ±0.055 | 0.675 ±0.135 | 4.175 ±0.920 | 3.690 ±1.959 | ND |
| $TEOH^+$ | 2.562 ±0.962 | 4.118 ±0.741 | 25.129 ±0.343 | 14.376 ±8.688 | 0.672 ±0.558 |
| $DEAH^+ + TMAH^+$ | 13.728 ±7.512 | 18.973 ±0.466 | 46.148 ±12.185 | 28.568 ±5.321 | 2.012 ±0.878 |
| $DMAH^+$ | 1.434 ±0.925 | 3.376 ±0.674 | 12.110 ±6.166 | 35.887 ±2.940 | 1.374 ±0.144 |
| **Elemental Species (mg kg$^{-1}$)** | **53.813 ±18.860** | **53.546 ±9.070** | **131.612 ±5.920** | **27.577 ±3.700** | **14.003 ±8.710** |
| **Phenols (mg kg$^{-1}$)** | **26.785 ±8.582** | **16.390 ±2.652** | **27.238 ±4.861** | **41.481 ±5.517** | **9.673 ±2.272** |
| **PAHs (mg kg$^{-1}$)** | **1.814 ±0.348** | **2.706 ±0.798** | **7.267 ±1.722** | **8.302 ±2.856** | **1.832 ±0.353** |

*ND means not detected*



**Table 2.** Emission factors of particulate chemical species in smoke $PM_{1.0}$ from
agricultural residue burning.

| Chemical Species (g kg⁻¹) | wheat straw | corn straw | rice straw | cotton residue | soybean residue |
|---|---|---|---|---|---|
| $PM_{1.0}$ | 5.298 ±0.295 | 5.360 ±0.551 | 13.200 ±1.440 | 12.635 ±1.243 | 3.036 ±0.257 |
| OC | 2.419 ±0.126 | 2.063 ±0.340 | 6.024 ±0.602 | 6.036 ±0.360 | 1.338 ±0.128 |
| EC | 0.650 ±0.037 | 0.728 ±0.122 | 2.083 ±0.413 | 1.023 ±0.205 | 0.575 ±0.260 |
| Char-EC | 0.567 ±0.033 | 0.580 ±0.098 | 1.671 ±0.331 | 0.916 ±0.184 | 0.511 ±0.233 |
| Soot-EC | 0.083 ±0.014 | 0.148 ±0.057 | 0.411 ±0.073 | 0.107 ±0.048 | 0.063 ±0.057 |
| **Inorganic ions (g kg⁻¹)** | **1.215 ±0.040** | **1.768 ±0.010** | **2.940 ±0.249** | **3.516 ±0.145** | **0.510 ±0.156** |
| $SO_4^{2-}$ | 0.078 ±0.011 | 0.199 ±0.032 | 0.333 ±0.107 | 0.581 ±0.054 | 0.073 ±0.056 |
| $Cl^-$ | 0.544 ±0.033 | 0.712 ±0.027 | 1.145 ±0.118 | 1.243 ±0.067 | 0.175 ±0.031 |
| $F^-$ | 0.022 ±0.007 | 0.041 ±0.004 | 0.078 ±0.030 | 0.151 ±0.011 | 0.001 ±0.001 |
| $NO_3^-$ | 0.021 ±0.005 | 0.027 ±0.002 | 0.043 ±0.016 | 0.061 ±0.003 | 0.009 ±0.002 |
| $NO_2^-$ | 0.006 ±0.001 | 0.010 ±0.003 | 0.013 ±0.004 | 0.019 ±0.002 | 0.004 ±0.003 |
| $Ca^{2+}$ | 0.027 ±0.013 | 0.028 ±0.002 | 0.045 ±0.008 | 0.067 ±0.005 | 0.010 ±0.002 |
| $Na^+$ | 0.004 ±0.000 | 0.012 ±0.000 | 0.027 ±0.003 | 0.056 ±0.006 | 0.005 ±0.002 |
| $NH_4^+$ | 0.147 ±0.005 | 0.191 ±0.009 | 0.511 ±0.067 | 0.401 ±0.004 | 0.031 ±0.005 |
| $Mg^{2+}$ | 0.005 ±0.001 | 0.035 ±0.001 | 0.024 ±0.006 | 0.033 ±0.002 | 0.005 ±0.001 |
| $K^+$ | 0.359 ±0.040 | 0.513 ±0.015 | 0.721 ±0.073 | 0.994 ±0.067 | 0.197 ±0.035 |
| **Organic Acids (mg kg⁻¹)** | **124.310 ±25.170** | **47.830 ±10.610** | **427.400 ±221.270** | **639.820 ±244.960** | **130.760 ±59.310** |
| $CH_3COOH$ | 115.790 ±21.940 | 38.960 ±9.610 | 383.360 ±179.050 | 615.790 ±232.860 | 124.310 ±69.000 |
| MSA | 6.830 ±2.030 | 8.870 ±2.730 | 41.380 ±38.480 | 11.380 ±2.360 | 3.200 ±1.730 |
| $H_2C_2O_4$ | 1.690 ±1.200 | ND | 2.660 ±1.760 | 3.620 ±1.250 | 1.560 ±1.670 |
| HCOOH | ND | ND | ND | 9.030 ±7.710 | 1.690 ±1.390 |
| **Amine salts (mg kg⁻¹)** | **18.191 ±5.351** | **29.891 ±13.480** | **81.726 ±11.455** | **85.720 ±21.337** | **4.385 ±1.445** |
| $MeOH^+ + MMAH^+$ | 1.300 ±0.282 | 5.647 ±0.342 | 16.627 ±0.104 | 18.834 ±1.991 | 0.464 ±0.265 |
| $MEAH^+$ | 0.157 ±0.037 | 0.787 ±0.211 | 3.581 ±0.602 | 2.771 ±1.304 | ND |
| $TEOH^+$ | 1.719 ±0.283 | 5.115 ±0.732 | 17.575 ±0.844 | 11.441 ±3.229 | 0.529 ±0.304 |
| $DEAH^+ + TMAH^+$ | 13.716 ±9.047 | 15.921 ±1.620 | 33.565 ±6.795 | 29.057 ±3.793 | 2.278 ±0.533 |
| $DMAH^+$ | 1.300 ±0.702 | 2.420 ±0.575 | 10.377 ±4.521 | 23.617 ±20.086 | 1.115 ±0.343 |
| **Elemental Species (mg kg⁻¹)** | **31.586 ±10.630** | **29.265 ±4.240** | **51.062 ±5.920** | **16.738 ±3.480** | **11.817 ±6.650** |
| **Phenols (mg kg⁻¹)** | **20.774 ±4.972** | **13.193 ±2.181** | **20.480 ±1.403** | **23.521 ±8.521** | **7.689 ±1.356** |
| **PAHs (mg kg⁻¹)** | **1.257 ±0.398** | **1.420 ±0.232** | **3.967 ±0.970** | **4.359 ±1.373** | **1.123 ±0.205** |

*ND means not detected*



**Table 3.** Emission factors of particulate THM, PAHs, and Phenols in smoke $PM_{2.5}$
from agricultural residue burning.

| Chemical Species (mg kg$^{-1}$) | wheat straw | corn straw | rice straw | cotton residue | soybean residue |
|---|---|---|---|---|---|
| **Elemental Species** | **53.813 ±18.860** | **53.546 ±9.070** | **131.612 ±5.920** | **27.577 ±3.700** | **14.003 ±8.710** |
| As | 6.433 ±1.424 | 4.684 ±0.879 | 6.724 ±0.737 | 2.082 ±1.078 | 0.777 ±0.525 |
| Zn | 0.868 ±0.180 | 0.358 ±0.624 | 0.275 ±0.177 | 0.229 ±0.264 | 0.053 ±0.046 |
| Pb | ND | ND | 0.467 ±0.313 | 0.063 ±0.053 | 0.059 ±0.047 |
| Cd | ND | ND | 0.053 ±0.000 | ND | ND |
| Ni | 0.726 ±0.074 | 0.695 ±0.138 | 1.100 ±0.113 | 0.372 ±0.170 | 0.193 ±0.092 |
| Cr | 1.026 ±0.335 | 0.746 ±0.299 | 3.324 ±0.257 | 0.543 ±0.055 | 0.266 ±0.127 |
| V | 0.159 ±0.006 | 0.104 ±0.061 | 0.560 ±0.022 | 0.110 ±0.011 | 0.051 ±0.044 |
| Al | 44.602 ±5.269 | 46.957 ±10.471 | 119.108 ±4.636 | 24.178 ±2.331 | 12.603 ±6.709 |
| **PAHs** | **2.407 ±0.348** | **2.706 ±0.798** | **7.267 ±1.722** | **6.017 ±2.856** | **1.832 ±0.353** |
| naphthalene | 0.417 ±0.116 | 0.087 ±0.077 | 0.780 ±0.128 | 0.116 ±0.086 | 0.093 ±0.041 |
| acenaphthylene | 0.032 ±0.023 | 0.028 ±0.013 | 0.701 ±0.269 | 0.201 ±0.277 | 0.004 ±0.006 |
| acenaphthene | 0.107 ±0.034 | 0.285 ±0.143 | 1.713 ±0.542 | 0.502 ±0.667 | 0.073 ±0.173 |
| flourene | 0.021 ±0.010 | 0.003 ±0.002 | 0.069 ±0.005 | 0.017 ±0.024 | 0.001 ±0.001 |
| anthracene | 0.343 ±0.121 | 0.384 ±0.111 | 0.656 ±0.003 | 1.177 ±0.536 | 0.245 ±0.127 |
| phenathrene | 0.179 ±0.090 | 0.112 ±0.030 | 0.202 ±0.007 | 0.547 ±0.239 | 0.105 ±0.011 |
| flouranthene | 0.368 ±0.071 | 0.561 ±0.217 | 0.926 ±0.029 | 0.930 ±0.250 | 0.306 ±0.042 |
| pyrene | 0.628 ±0.107 | 0.853 ±0.240 | 1.460 ±0.039 | 1.818 ±0.598 | 0.586 ±0.178 |
| benz[a]anthracene | 0.057 ±0.019 | 0.056 ±0.023 | 0.118 ±0.016 | 0.158 ±0.056 | 0.058 ±0.026 |
| chrysene | 0.058 ±0.008 | 0.088 ±0.033 | 0.119 ±0.010 | 0.166 ±0.057 | 0.063 ±0.010 |
| benzo[a]pyrene | 0.148 ±0.025 | 0.113 ±0.044 | 0.398 ±0.083 | 0.148 ±0.076 | 0.131 ±0.072 |
| benzo[b]flouranthene | 0.017 ±0.012 | 0.051 ±0.049 | 0.026 ±0.008 | 0.086 ±0.011 | 0.047 ±0.007 |
| benzo[k]flouranthene | 0.021 ±0.008 | 0.014 ±0.011 | 0.022 ±0.009 | 0.036 ±0.006 | 0.020 ±0.013 |
| benzo[g,h,i]pyrene | 0.006 ±0.003 | 0.024 ±0.024 | 0.011 ±0.004 | 0.046 ±0.011 | 0.033 ±0.046 |
| indeno[1,2,3-cd]pyrene | 0.005 ±0.001 | 0.086 ±0.011 | ND | 0.022 ±0.012 | ND |
| dibenz[a,h]anthracene | 0.002 ±0.001 | 0.038 ±0.051 | 0.068 ±0.027 | 0.066 ±0.003 | 0.067 ±0.047 |
| **Phenols** | **26.785 ±8.582** | **16.390 ±2.652** | **27.238 ±4.861** | **41.481 ±5.517** | **9.673 ±2.272** |
| phenol | 2.357 ±0.797 | 3.974 ±0.759 | 10.737 ±6.373 | 3.992 ±0.128 | 2.834 ±2.944 |
| 2-methoxyphenol | 0.567 ±0.061 | 0.531 ±0.015 | 2.545 ±0.200 | 0.371 ±0.083 | 0.363 ±0.712 |
| 4-ethylphenol | 2.239 ±0.323 | 1.417 ±0.536 | 1.624 ±0.740 | 5.105 ±0.707 | 0.475 ±0.358 |
| 4-ethyl-2-methoxyphenol | 0.671 ±0.318 | 0.290 ±0.070 | 0.383 ±0.116 | 1.588 ±0.244 | 0.187 ±0.375 |
| 2,6-dimethoxyphenol | 20.952 ±8.677 | 10.178 ±2.334 | 11.949 ±0.456 | 30.424 ±4.662 | 5.815 ±2.117 |

*ND means not detected*



**Table 4.** Emission factors of particulate THM, PAHs, and Phenols in smoke $PM_{1.0}$
from agricultural residue burning.

| Chemical Species (mg kg$^{-1}$) | wheat straw | corn straw | rice straw | cotton residue | soybean residue |
|---|---|---|---|---|---|
| **Elemental Species** | **31.586 ±10.630** | **29.265 ±4.240** | **51.062 ±5.920** | **16.738 ±3.480** | **11.817 ±6.650** |
| As | 2.781 ±1.159 | 2.984 ±0.617 | 4.861 ±0.737 | 1.751 ±1.529 | 0.342 ±0.750 |
| Zn | 0.607 ±0.514 | 0.137 ±0.091 | 0.293 ±0.489 | 0.112 ±0.059 | 0.040 ±0.035 |
| Pb | ND | ND | ND | 0.007 ±0.004 | 0.013 ±0.006 |
| Cd | ND | ND | 0.043 ±0.000 | ND | ND |
| Ni | 0.435 ±0.057 | 0.365 ±0.042 | 0.654 ±0.113 | 0.218 ±0.033 | 0.171 ±0.098 |
| Cr | 0.556 ±0.024 | 0.487 ±0.000 | 0.923 ±0.257 | 0.292 ±0.030 | 0.233 ±0.092 |
| V | 0.101 ±0.005 | 0.118 ±0.044 | 0.188 ±0.022 | 0.065 ±0.010 | 0.049 ±0.023 |
| Al | 27.106 ±3.566 | 25.115 ±3.497 | 44.037 ±4.636 | 14.293 ±1.834 | 10.968 ±5.592 |
| **PAHs** | **1.257 ±0.398** | **1.420 ±0.232** | **3.967 ±0.970** | **3.159 ±1.373** | **1.123 ±0.205** |
| naphthalene | 0.118 ±0.031 | 0.112 ±0.131 | 0.360 ±0.106 | 0.043 ±0.011 | 0.082 ±0.130 |
| acenaphthylene | 0.023 ±0.018 | 0.028 ±0.021 | 0.339 ±0.333 | 0.074 ±0.102 | 0.008 ±0.008 |
| acenaphthene | 0.034 ±0.014 | 0.173 ±0.055 | 0.828 ±0.783 | 0.269 ±0.354 | 0.068 ±0.025 |
| flourene | 0.009 ±0.007 | 0.003 ±0.001 | 0.033 ±0.005 | 0.006 ±0.006 | 0.002 ±0.000 |
| anthracene | 0.210 ±0.107 | 0.209 ±0.052 | 0.178 ±0.166 | 0.600 ±0.251 | 0.197 ±0.051 |
| phenathrene | 0.097 ±0.030 | 0.084 ±0.016 | 0.055 ±0.045 | 0.259 ±0.048 | 0.077 ±0.149 |
| flouranthene | 0.212 ±0.086 | 0.219 ±0.077 | 0.636 ±0.048 | 0.475 ±0.116 | 0.178 ±0.026 |
| pyrene | 0.391 ±0.146 | 0.385 ±0.142 | 1.160 ±0.009 | 1.043 ±0.714 | 0.298 ±0.065 |
| benz[a]anthracene | 0.031 ±0.009 | 0.032 ±0.016 | 0.097 ±0.006 | 0.086 ±0.010 | 0.033 ±0.019 |
| chrysene | 0.034 ±0.004 | 0.056 ±0.022 | 0.091 ±0.011 | 0.096 ±0.009 | 0.037 ±0.018 |
| benzo[a]pyrene | 0.071 ±0.031 | 0.057 ±0.038 | 0.129 ±0.039 | 0.107 ±0.010 | 0.055 ±0.002 |
| benzo[b]flouranthene | 0.013 ±0.005 | 0.018 ±0.018 | 0.047 ±0.033 | 0.043 ±0.010 | 0.031 ±0.005 |
| benzo[k]flouranthene | 0.009 ±0.003 | 0.011 ±0.008 | 0.014 ±0.005 | 0.018 ±0.001 | 0.012 ±0.013 |
| benzo[g,h,i]pyrene | 0.005 ±0.005 | 0.007 ±0.005 | ND | 0.005 ±0.005 | 0.014 ±0.057 |
| indeno[1,2,3-cd]pyrene | ND | 0.011 ±0.006 | ND | 0.011 ±0.012 | ND |
| dibenz[a,h]anthracene | ND | 0.014 ±0.015 | ND | 0.025 ±0.029 | 0.031 ±0.001 |
| **Phenols** | **20.774 ±4.972** | **13.193 ±2.181** | **20.480 ±1.403** | **23.521 ±8.521** | **7.689 ±1.356** |
| phenol | 3.296 ±1.962 | 4.389 ±0.089 | 8.917 ±2.588 | 2.824 ±0.031 | 1.660 ±0.293 |
| 2-methoxyphenol | 0.604 ±0.003 | 0.682 ±0.357 | 1.711 ±0.155 | 0.353 ±0.088 | 0.195 ±0.034 |
| 4-ethylphenol | 1.387 ±0.408 | 0.490 ±0.246 | 1.171 ±0.233 | 2.965 ±0.441 | 0.495 ±0.087 |
| 4-ethyl-2-methoxyphenol | 0.438 ±0.193 | 0.231 ±0.004 | 0.222 ±0.039 | 0.834 ±0.180 | 0.137 ±0.024 |
| 2,6-dimethoxyphenol | 15.050 ±6.336 | 7.402 ±0.478 | 8.459 ±0.759 | 16.545 ±2.113 | 5.202 ±0.917 |

*ND means not detected*



**Table 5.** Comparison of emission factors with literature (specific chemical materials in form of $PM_{2.5}$).

| Species | Emission factors (g kg$^{-1}$) | | Reference |
|---|---|---|---|
| | **This work** | **Reference value** | **Reference** |
| $PM_{2.5}$ | 8.99 ± 5.55 | 7.6~11.7(AR), 6.26~15.3 (TL), ~3.0 (AR), 2.2~15.0 (AR) | Li et al., 2007; Akagi et al., 2011; Dhammapala et al., 2007; Hayashi et al 2014 |
| $PM_{1.0}$ | 7.91 ± 4.67 | 4.4.3~12.1 (TL) | May et al., 2014 |
| OC | 4.21 ± 2.73 | 2.7~3.9 (AR), 2.3~9.7(TL) , ~1.9(AR) , 1.0~9.3 (AR), 0.8~5.9 (TL) | Li et al., 2007; Akagi et al., 2011; Dhammapala et al., 2007; Hayashi et al., 2014; May et al.2014 |
| EC | 1.09 ± 0.65 | 0.35~0.49 (AR), 0.37~0.91(TL), ~0.4(AR), 0.21~0.81(AR), 1.13~1.73 (TL) | Li et al., 2007; Akagi et al., 2011; Dhammapala et al., 2007; Hayashi et al., 2014; May et al.2014 |
| WSOA | 0.33 ± 0.31 | 0.039~0.109 (TL) | Akagi et al., 2011 |
| WSA | 0.05 ± 0.05 | 0.08~0.13 (TL), ~0.55 (TL) | Akagi et al., 2011; Andreae et al., 2001 |
| WSI | 2.10 ± 1.34 | 1.84~4.9 (AR),0.8~1.31(TL), 0.43~1.63 (AR) | Li et al., 2007; Akagi et al., 2011; Hayashi et al., 2014 |
| THM | 0.06 ± 0.05 | 0.06~0.09 (AR) | Li et al., 2007 |
| PAHs (×10$^3$) | 4.38 ± 3.15 | ~17(AR), 0.72~1.64(AR), ~9.0 (W) | Dhammapala et al., 2007; Zhang et al., 2011; Lee et al.2005 |
| Phenols (×10$^3$) | 24.31 ± 12.11 | ~35(AR), ~5 (AR), ~13 (TL) | Dhammapala et al., 2007; Hays et al., 2005; Andreae et al.2001 |

AR: agricultural residue; TL: total, including forest fires and straw burning; W: wood





**Table 6.** Summary of field burning rates and economic data in China.

| Province | Burning rate from literature | | Agricultural income ratio [c] | | | Estimated burning rate | | NDRC report [d] | Average rate |
|---|---|---|---|---|---|---|---|---|---|
| | BAU-I [a] | BAU-II [b] | 2000 | 2006 | 2012 | EM-I | EM-II | NDRC | |
| Beijing | 0.00 | 0.17 | 0.08 | 0.06 | 0.06 | 0.00 | 0.19 | 0.13 | 0.10 ±0.08 |
| Tianjin | 0.00 | 0.17 | 0.10 | 0.14 | 0.12 | 0.00 | 0.20 | 0.30 | 0.13 ±0.12 |
| Hebei | 0.20 | 0.17 | 0.27 | 0.22 | 0.24 | 0.22 | 0.16 | 0.19 | 0.19 ±0.02 |
| Shanxi | 0.20 | 0.17 | 0.20 | 0.21 | 0.25 | 0.16 | 0.14 | 0.22 | 0.18 ±0.03 |
| Inner Mongolia | 0.00 | 0.12 | 0.44 | 0.49 | 0.66 | 0.00 | 0.09 | 0.27 | 0.10 ±0.10 |
| Liaoning | 0.20 | 0.12 | 0.30 | 0.29 | 0.39 | 0.16 | 0.09 | 0.34 | 0.18 ±0.09 |
| Jilin | 0.30 | 0.12 | 0.73 | 0.73 | 0.77 | 0.28 | 0.11 | 0.25 | 0.21 ±0.08 |
| Heilongjiang | 0.30 | 0.12 | 0.99 | 0.83 | 0.59 | 0.50 | 0.17 | 0.25 | 0.27 ±0.13 |
| Shanghai | 0.00 | 0.32 | 0.10 | 0.08 | 0.09 | 0.00 | 0.29 | 0.12 | 0.15 ±0.14 |
| Jiangsu | 0.30 | 0.32 | 0.32 | 0.22 | 0.30 | 0.32 | 0.23 | 0.19 | 0.27 ±0.05 |
| Zhejiang | 0.30 | 0.32 | 0.19 | 0.08 | 0.09 | 0.64 | 0.28 | 0.22 | 0.35 ±0.15 |
| Anhui | 0.20 | 0.32 | 0.44 | 0.39 | 0.43 | 0.21 | 0.29 | 0.43 | 0.29 ±0.08 |
| Fujian | 0.30 | 0.32 | 0.18 | 0.10 | 0.14 | 0.39 | 0.22 | 0.17 | 0.28 ±0.08 |
| Jiangxi | 0.20 | 0.11 | 0.45 | 0.31 | 0.44 | 0.20 | 0.08 | 0.25 | 0.17 ±0.06 |
| Shandong | 0.30 | 0.17 | 0.33 | 0.25 | 0.24 | 0.40 | 0.17 | 0.21 | 0.25 ±0.09 |
| Henan | 0.20 | 0.17 | 0.39 | 0.35 | 0.33 | 0.23 | 0.18 | 0.22 | 0.20 ±0.02 |
| Hubei | 0.20 | 0.11 | 0.42 | 0.30 | 0.41 | 0.21 | 0.08 | 0.30 | 0.18 ±0.08 |
| Hunan | 0.20 | 0.33 | 0.47 | 0.31 | 0.43 | 0.22 | 0.24 | 0.35 | 0.27 ±0.06 |
| Guangdong | 0.30 | 0.33 | 0.19 | 0.10 | 0.13 | 0.44 | 0.25 | 0.18 | 0.30 ±0.09 |
| Guangxi | 0.20 | 0.33 | 0.40 | 0.25 | 0.33 | 0.25 | 0.25 | 0.35 | 0.28 ±0.06 |
| Hainan | 0.30 | 0.33 | 0.35 | 0.16 | 0.21 | 0.51 | 0.25 | 0.56 | 0.39 ±0.12 |
| Chongqing | 0.20 | 0.11 | 0.35 | 0.23 | 0.30 | 0.24 | 0.08 | 0.45 | 0.22 ±0.13 |
| Sichuan | 0.20 | 0.11 | 0.37 | 0.22 | 0.28 | 0.26 | 0.09 | 0.30 | 0.19 ±0.08 |
| Guizhou | 0.20 | 0.11 | 0.38 | 0.23 | 0.25 | 0.31 | 0.10 | 0.43 | 0.23 ±0.13 |
| Yunnan | 0.20 | 0.11 | 0.36 | 0.26 | 0.31 | 0.24 | 0.09 | 0.28 | 0.18 ±0.07 |
| Tibet | 0.00 | 0.16 | 0.15 | 0.09 | 0.05 | 0.00 | 0.30 | 0.16 | 0.12 ±0.11 |
| Shannxi | 0.20 | 0.17 | 0.33 | 0.27 | 0.26 | 0.25 | 0.18 | 0.28 | 0.22 ±0.04 |
| Gansu | 0.10 | 0.16 | 0.25 | 0.20 | 0.28 | 0.09 | 0.11 | 0.33 | 0.16 ±0.09 |
| Qinghai | 0.00 | 0.16 | 0.23 | 0.10 | 0.08 | 0.00 | 0.20 | 0.28 | 0.13 ±0.11 |
| Ningxia | 0.10 | 0.16 | 0.42 | 0.38 | 0.45 | 0.09 | 0.13 | 0.16 | 0.13 ±0.03 |
| Xinjiang | 0.10 | 0.16 | 0.43 | 0.61 | 0.73 | 0.06 | 0.13 | 0.30 | 0.15 ±0.08 |
| **Nationwide** | **0.21** | **0.16** | **0.34** | **0.27** | **0.31** | **0.26** | **0.15** | **0.27** | **0.21 ±0.05** |

*a.  Zhao et al., 2012; Cao et al., 2006; Cao et al., 2011*

*b.  Wang and Zhang., 2008*

*c.  Calculated based on data from China Yearbook 2001~2013 (NBSC, 2001-2013), China Rural Statistic Yearbook 2001~2013, data available at http://www.grain.gov.cn/Grain/*

d.  *Data from the National Development and Reform Commission report ([2014]No.516) : http://www.sdpc.gov.cn/*

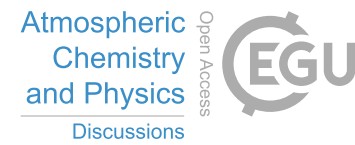

**Table 7.** National agricultural field burning emissions of BAU, EM, NDRC, and CRC scenarios in China in 2012.

| Unit: Gg | BAU-I | | | BAU-II | | | EM-1 | | | EM-2 | | | NDRC | | | Average | | |
|---|---|---|---|---|---|---|---|---|---|---|---|---|---|---|---|---|---|---|
| | Total | Summer | Autumn | Total | Summer | Autumn | Total | Summer | Autumn | Total | Summer | Autumn | Total | Summer | Autumn | Total | Summer | Autumn |
| $PM_{2.5}$ | 1001.05 | 218.99 | 782.06 | 835.42 | 209.29 | 626.13 | 1211.92 | 258.58 | 953.34 | 738.36 | 182.34 | 556.02 | 1241.69 | 258.24 | 983.46 | 1007.650 | 226.007 | 781.646 |
| $PM_{1.0}$ | 897.52 | 198.93 | 698.59 | 748.57 | 189.92 | 558.65 | 1087.05 | 234.85 | 852.20 | 661.81 | 165.61 | 496.20 | 1111.90 | 234.44 | 877.46 | 903.125 | 205.217 | 697.911 |
| OC | 429.51 | 102.87 | 326.64 | 360.99 | 97.67 | 263.32 | 519.26 | 121.33 | 397.94 | 318.84 | 85.55 | 233.29 | 533.19 | 120.86 | 412.33 | 433.184 | 105.885 | 327.300 |
| EC | 133.61 | 27.37 | 106.24 | 111.40 | 26.52 | 84.88 | 162.71 | 32.39 | 130.32 | 98.06 | 22.85 | 75.21 | 164.97 | 32.53 | 132.45 | 134.414 | 28.404 | 106.010 |
| char-EC | 112.75 | 23.76 | 88.99 | 93.82 | 22.88 | 70.94 | 137.15 | 28.09 | 109.06 | 82.79 | 19.81 | 62.98 | 139.21 | 28.14 | 111.07 | 113.366 | 24.596 | 88.770 |
| soot-EC | 20.80 | 3.59 | 17.21 | 17.54 | 3.62 | 13.91 | 25.50 | 4.28 | 21.22 | 15.23 | 3.02 | 12.21 | 25.70 | 4.36 | 21.33 | 20.992 | 3.787 | 17.205 |
| $SO_4^{2-}$ | 30.22 | 3.96 | 26.26 | 24.97 | 3.94 | 21.04 | 36.39 | 4.71 | 31.68 | 22.09 | 3.32 | 18.76 | 38.21 | 4.78 | 33.44 | 30.440 | 4.155 | 26.285 |
| $NO_3^-$ | 4.35 | 0.84 | 3.51 | 3.55 | 0.80 | 2.75 | 5.24 | 0.99 | 4.25 | 3.17 | 0.70 | 2.47 | 5.40 | 0.99 | 4.41 | 4.350 | 0.864 | 3.486 |
| $NH_4^+$ | 32.08 | 6.37 | 25.71 | 26.65 | 6.21 | 20.44 | 39.09 | 7.54 | 31.55 | 23.43 | 5.32 | 18.11 | 39.46 | 7.59 | 31.87 | 32.202 | 6.623 | 25.580 |
| $K^+$ | 67.49 | 13.12 | 54.38 | 54.75 | 12.38 | 42.37 | 81.40 | 15.45 | 65.95 | 49.10 | 10.90 | 38.20 | 83.62 | 15.36 | 68.26 | 67.412 | 13.469 | 53.943 |
| WSOA | 24.44 | 6.55 | 17.89 | 21.94 | 6.39 | 15.55 | 29.69 | 7.76 | 21.93 | 18.77 | 5.48 | 13.30 | 30.82 | 7.81 | 23.01 | 25.174 | 6.815 | 18.360 |
| WSA | 5.75 | 0.95 | 4.80 | 4.85 | 0.95 | 3.90 | 6.99 | 1.13 | 5.86 | 4.23 | 0.80 | 3.43 | 7.19 | 1.15 | 6.04 | 5.815 | 1.000 | 4.815 |
| PAHs | 0.48 | 0.11 | 0.37 | 0.40 | 0.10 | 0.30 | 0.58 | 0.12 | 0.45 | 0.35 | 0.09 | 0.26 | 0.59 | 0.13 | 0.47 | 0.480 | 0.109 | 0.371 |
| Phenols | 2.71 | 0.85 | 1.87 | 2.25 | 0.78 | 1.47 | 3.25 | 0.99 | 2.26 | 2.02 | 0.70 | 1.323 | 3.40 | 0.98 | 2.36 | 2.721 | 0.861 | 1.861 |
| THM | 8.68 | 2.01 | 6.67 | 7.19 | 1.92 | 5.27 | 10.56 | 2.37 | 8.19 | 6.36 | 1.67 | 4.69 | 10.64 | 2.37 | 8.27 | 8.702 | 2.073 | 6.628 |
| WSI | 249.96 | 47.46 | 202.50 | 204.46 | 45.24 | 159.22 | 301.75 | 56.01 | 245.74 | 182.31 | 39.50 | 142.82 | 310.31 | 55.88 | 254.43 | 250.269 | 48.927 | 201.342 |



**Table 8.** Uncertainties in emission estimates.

| Region | 2012 | Region | 2012 |
|---|---|---|---|
| Anhui | 28.9% | | |
| Shandong | 35.5% | | |
| Hebei | 11.4% | The North Plain | 12.7% |
| Beijing | 84.0% | | |
| Tianjin | 87.7% | | |
| Shanxi | 16.0% | | |
| Hubei | 43.5% | | |
| Hunan | 22.6% | The Center of China | 16.0% |
| Henan | 11.4% | | |
| Shanghai | 94.1% | | |
| Jiangsu | 19.4% | The Yangtze River Delta | 21.5% |
| Zhejiang | 42.0% | | |
| Liaoning | 48.0% | | |
| Jilin | 38.1% | The Northeast of China | 39.8% |
| Heilongjiang | 49.1% | | |
| Guangdong | 28.8% | | |
| Guangxi | 20.2% | | |
| Hainan | 31.3% | | |
| Fujian | 27.6% | | |
| Sichuan | 42.6% | The Pan-Pearl River Delta | 25.9% |
| Guizhou | 54.5% | | |
| Yunnan | 39.9% | | |
| Jiangxi | 37.5% | | |
| Inner Mongolia | 93.4% | | |
| Tibet | 91.5% | | |
| Shannxi | 19.6% | | |
| Gansu | 56.5% | | |
| Qinghai | 87.1% | The West of China | 51.4% |
| Ningxia | 22.9% | | |
| Xinjiang | 54.7% | | |
| Chongqing | 60.5% | | |
| **Nationwide** | **19.8%** | | |

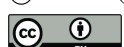



**Table 9.** Estimated number of cases (95% CI) attributable to agricultural fire smoke PM$_{2.5}$ exposure in China, 2012.

| Emission version | Mortality | Respiratory hospital admission | Cardiovascular hospital admission | Chronic bronchitis |
|---|---|---|---|---|
| BAU-I | 7864 (3154, 12489) | 31123 (21114, 40788) | 29454 (12849, 45481) | 7577067 (2952006, 11024705) |
| BAU-II | 7187 (3056, 11260) | 28711 (19443, 37693) | 27156 (11825, 42007) | 7132581 (2735111, 10523803) |
| EM-I | 9435 (3817, 14933) | 36950 (25151, 48269) | 35116 (15373, 54042) | 8712880 (3484325, 12430411) |
| EM-II | 6175 (2554, 9751) | 25166 (17004, 33112) | 23745 (10316, 36816) | 6383442 (2407643, 9526727) |
| NDRC | 8523 (3581, 13377) | 33957 (23015, 44542) | 32131 (14003, 49664) | 8332216 (3228351, 12148274) |
| Average | 7836 (3232, 12362) | 31181 (21145, 40881) | 29520 (12873, 45602) | 7267237 (2961487, 1130784) |
| CRC | 538 (227, 850) | 2191 (1462, 2920) | 2038 (874, 3199) | 636650 (214617, 1052153) |



**Table 10**. Health-related economic loss (95% CI) from agricultural fire smoke $PM_{2.5}$ exposure in China, 2012.

| Emission version | Economic cost (million US$) | | | | Total cost (million US$) | GDP ratio (‰) |
|---|---|---|---|---|---|---|
| | Mortality | Respiratory hospital admission | Cardiovascular hospital admission | Chronic bronchitis | | |
| BAU-1 | 1544.5 (730.7, 2430.0) | 19.6 (13.3, 25.7) | 36.0 (15.7, 55.6) | 7187.6 (2800.3, 10458.3) | 8787.8 (3560.0, 12969.4) | 1.0 (0.4, 1.5) |
| BAU-2 | 1453.9 (719, 2252.2) | 18.1 (12.2, 23.8) | 33.2 (14.4, 51.3) | 6766.0 (2594.5, 9982.9) | 8271.2 (3340.3, 12310.3) | 1.0 (0.4, 1.4) |
| EM-1 | 1855.2 (870.3, 2913.7) | 23.3 (15.9, 30.5) | 42.9 (18.8, 66.1) | 8265.0 (3305.2, 11791.5) | 10186.5 (4210.2, 14801.8) | 1.2 (0.5, 1.7) |
| EM-2 | 1228.1 (600.6, 1917.6) | 15.9 (10.7, 20.9) | 29.0 (12.6, 450) | 6055.3 (2283.9, 9037.1) | 7328.4 (2907.9, 11020.7) | 0.9 (0.3, 1.3) |
| NDRC | 1573.4 (759.3, 2456.2) | 21.4 (14.5, 28.1) | 39.3 (17.1, 60.7) | 7903.9 (3062.4, 11523.9) | 9538.2 (3853.4, 14069.0) | 1.1 (0.4, 1.6) |
| Average | 1531.0 (736.0, 2393.9) | 19.7 (13.3, 25.8) | 36.1 (15.7, 55.7) | 7235.6 (2809.3, 10558.7) | 8822.4 (3574.4, 13034.2) | 1.0 (0.4, 1.5) |
| CRC | 100.0 (48.0, 157.1) | 1.3 (0.9, 1.8) | 2.4 (1.0, 3.9) | 603.9 (203.6, 998.1) | 707.8 (253.6, 1160.9) | 0.1 (0.0, 0.1) |





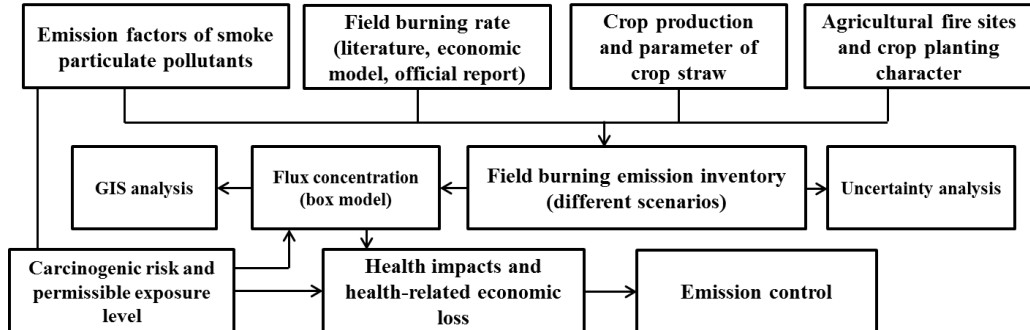

**Figure 1.** Schematic methodology for developing emission estimations.



**Figure 2.** Chemical profiles of smoke PM$_{2.5}$ and PM$_{1.0}$ from 5 types agricultural residue burnings. OM (organic matter = 1.3×OC). OWSI, other water soluble ions including F$^-$, NO$_2^-$, Na$^+$, Ca$^{2+}$, and Mg$^{2+}$.






**Figure 3.** (a) Emission factors of 16 USEPA priority PAHs in smoke PM$_{2.5}$ and PM$_{1.0}$; (b) expulsion-accumulation of PAHs in OC-EC of smoke PM$_{2.5}$ and PM$_{1.0}$.



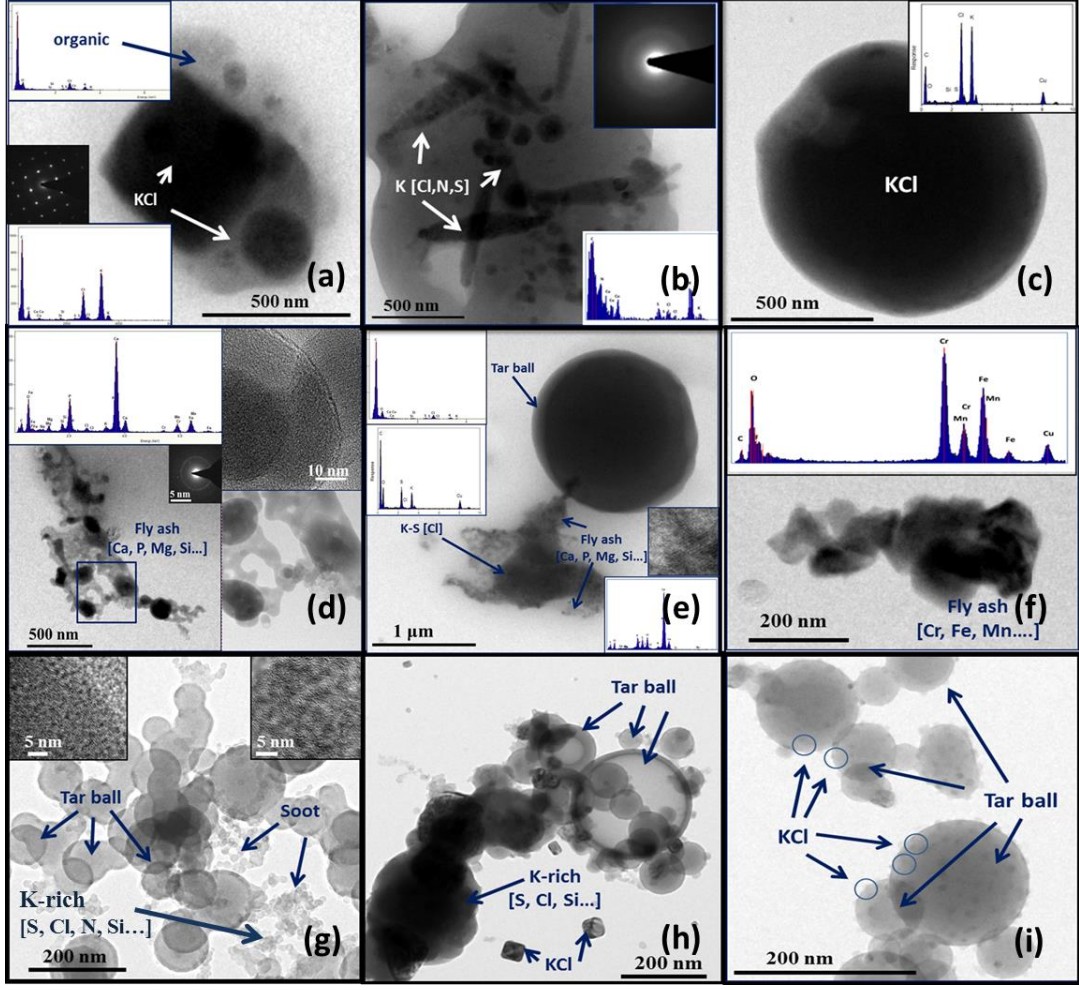

**Figure 4.** Transmission electron microscope (TEM) images and EDX analysis of fresh agricultural residue burning particles. (a)-(c) Crystal and amorphous KCl particles internally mixed with sulfate, nitrate, and carbonaceous materials. (d)-(f) Heavy metal-bearing fractal-like fly ash particles. (e)-(g) Chain-like soot particles and tar ball.





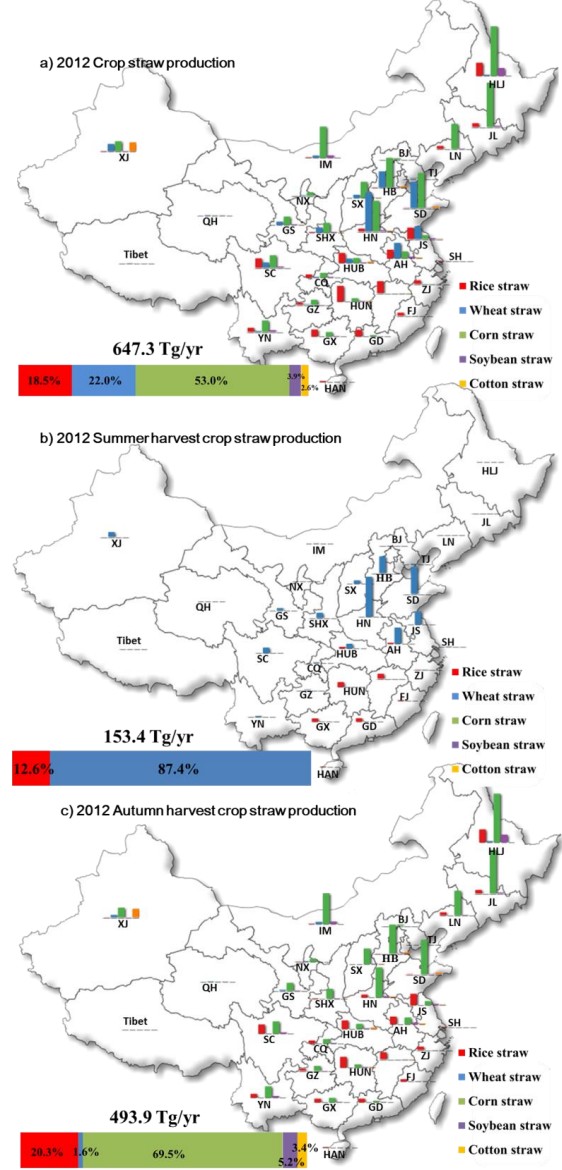

**Figure 5.** Annual agricultural residue production of five major crops and allocated into two harvest (summer and autumn harvest) based on agricultural yield in China, 2012. (Abbreviation, BJ: Beijing; TJ: Tianjin; HB: Hebei; SX: Shanxi; IM: Inner Mongolia; LN: Liaoning; JL: Jilin; HLJ: Heilongjiang; SH: Shanghai; JS: Jiangsu; ZJ: Zhejiang; AH: Anhui; FJ: Fujian; JX: Jiangxi; SD: Shandong; HN: Henan; HUB: Hubei; HUN: Hunan; GD: Guangdong; GX: Guangxi; HAN: Hainan; CQ: Chongqing; SC: Sichuan; GZ: Guizhou; YN: Yunnan; SHX: Shannxi; GS: Gansu; QH: Qinghai; NX: Ningxia; XJ: Xinjiang)





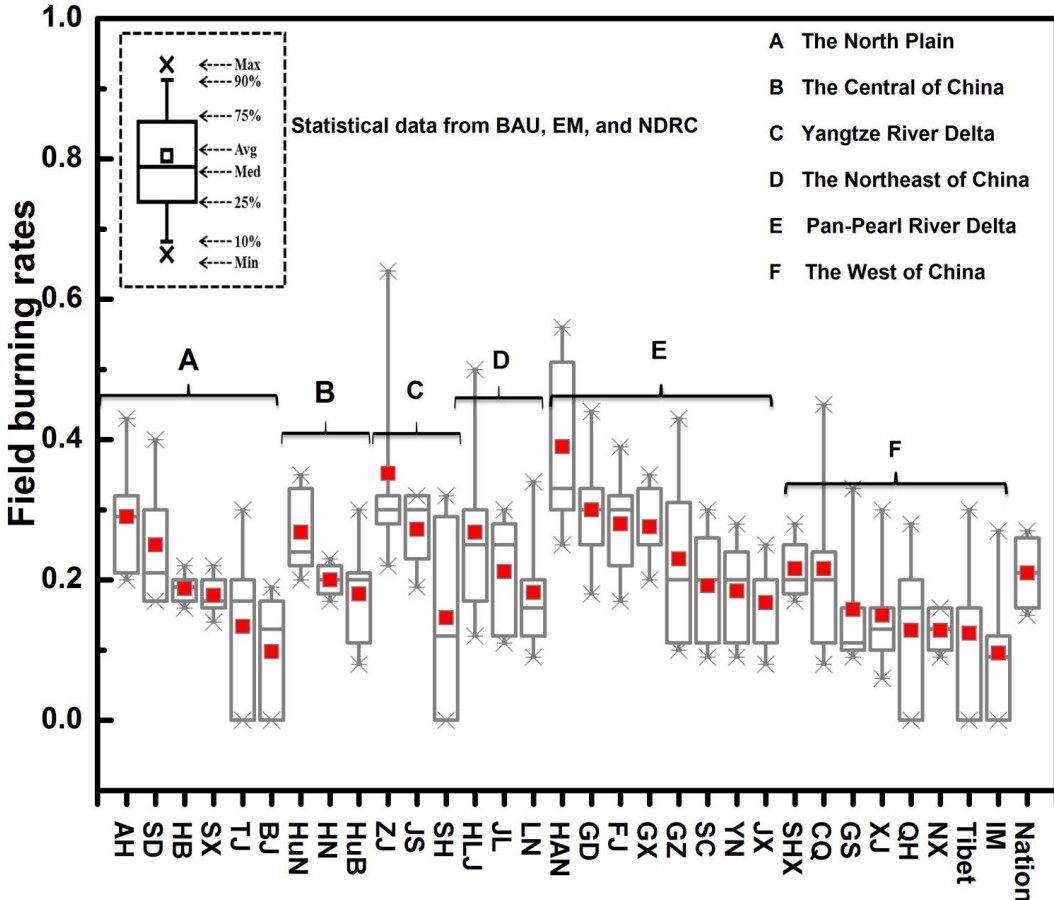

**Figure 6.** Statistical analysis of field burning rates from BAU, EM, and NDRC versions. The North Plain (Anhui, Shandong, Hebei, Shanxi, Tianjin, Beijing), the Central of China (Hunan, Henan, Hubei), the Yangtze River Delta (Zhejiang, Jiangsu, Shanghai), the Northeast of China (Heilongjiang, Liaoning, Jilin), the Pan-Pearl River Delta (Hainan, Guangdong, Fujian, Guangxi, Guizhou, Sichuan, Yunnan, Jiangxi), the West of China (Shannxi, Chongqing, Xinjiang, Qinghai, Ningxia, Tibet, Inner Mongolia, Gansu)





**Figure 7.** Spatial and temporal distribution of smoke $PM_{2.5}$ emissions and flux concentrations from agricultural field burning over China, 2012.



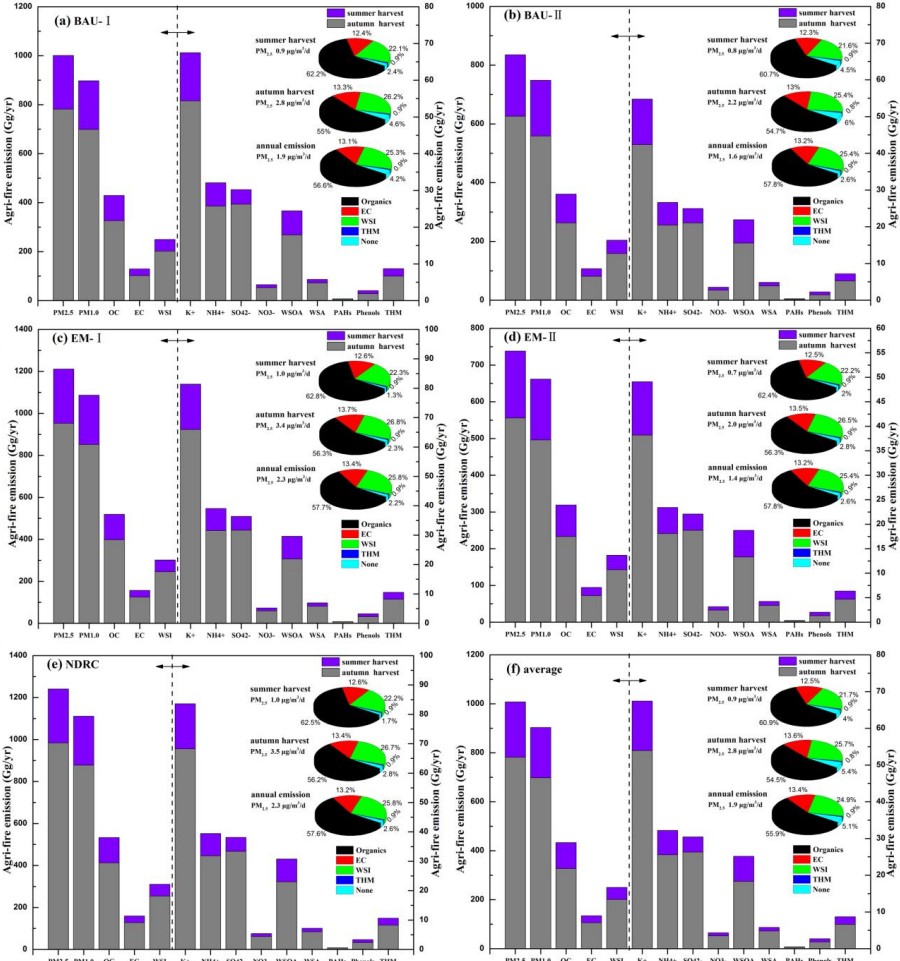

**Figure 8.** Nationwide $PM_{2.5}$ emissions and flux concentrations based on different burning versions. The inset pie-graphs are chemical compositions of integrated $PM_{2.5}$ from five major agricultural residue burning.