# Peer review of "Multi-pollutants emissions from the burning of major"

_Atmospheric Chemistry and Physics, 2016_

## Referee Comment (RC1) · Anonymous Referee #1 · 26 Sep 2016

**Review comments:** "Multi-pollutants emissions from the burning of major agricultural residues in China and the related health-economic effect assessment" by Li C. et al..

This paper describes results from experimental investigations on EFs of multi-pollutants from crop residues open burning in China, and try to estimate the health-economic effect under different scenarios. Considering the limited EFs for crop residues open burning, new emission data for various types of biomass are always welcome addition to the literature, and should be useful to air quality communities. *This paper is reasonably well written. But there are a number of revisions that should be addressed prior to publication.*

When comparing EFs in this study with literature data, I notice that only comparable data from literature is included (for example, Line 505 to 507 when comparing OC and EC EFs, Line 605 to 608 when comparing EFs of PAHs, etc ). However, it is already known and acknowledged that the EFs of crop residues burning could have a wide range due to different combustion condition, properties of biomass etc…(McMeeking et al., 2009; Reid et al., 2005). Both similarity and differences in EFs should be discussed. Moreover, due to crop residues used in this study were dehydrated at 100 degrees for 24h (thus had much lower moisture content compared with elsewhere), it tends to have a much lower EFs of incomplete combustion product, such as PM and OC from chamber studies (Chen et al., 2010; Hayashi et al., 2014). However, for biomass open burning combustion intensity may be higher than those from chamber studies, and thus this would result in a lower EF. When applying EFs from this study to estimate emission inventories, I would like to suggest that the uncertainties from the effect of moisture content and the burning style should be discussed in this paper.

China maps used in Figure 7 are incomplete, part of Xinjiang and Tibet is missing from maps in Figure 7, there should be a reason to explain this.

Line 228, the definition of MCE (Modified Combustion Efficiency) should be given. $MCE=\Delta CO_2/(\Delta CO_2+\Delta CO)$, where $\Delta CO_2$ and $\Delta CO$ are the excess molar mixing ratios of $CO_2$ and $CO$, and thus cannot be monitored directly, as stated on Line 228.

When stating there are "significant differences" between means, the statistical tests should be conducted and the results should be also given. Otherwise, there are no significant evidence that one mean differs from the other. The statistical test should be conducted in Line 495, 519, 617, 766…

Although several ways to estimate uncertainties of the emissions were mentioned in Section 3.3.4 (Line 827 to Line), it is not clear which method is used in this study. For the emission inventory in this study, a discussion of the overall inventory uncertainty is needed and this could be given by considering the uncertainties in each of the terms in the inventory (Eq 5).

There are also some errors and corrections needed where more care should also be taken, including but not limited to:

Line 215, "costume-built" should be "custom-built";

Citation formatting and styling errors should be corrected carefully. For example, Line 360, References should be cited with publication year. Chen et al. (2001) is cited under Cao's publication…

Line 374, Qin et al. (2012) is cited, but is missing from References list.

---

## Referee Comment (RC2) · Anonymous Referee #2 · 25 Nov 2016

This study investigates the emission factors of multi-pollutants from five major crop residues in China, and tries to estimate emission inventory and their corresponding health-economic effect. This paper is well organized and presents some interesting data. However, detailed explanations about the design should be given to ensure the data quality.

1) When the crop residues were dehydrated at 100 degree C for 24 hrs, what are the impacts to the emission factors and PM compositions?

2) There are huge variations on EFs of crop residues, and they depend on lots of factors such as, sources of crop resides, burning temperature, burning efficiency etc. What are the differences between chamber study and open burning? As the burning

last about 1 min only (in chamber study), can it represent the real open burning results? Moreover, what is the dilution ratio in the chamber study?

3) The detection limits (MDL) for all analysis should be provided.

4) It is interesting to determine char and soot, however, the temperature protocol is IMPROVE, but not NIOSH. Any calibrations have been performed with pure soot and char (standard)?

5) Please describe how to screen agricultural fire from MODIS daily fire products? What are the selecting criteria?

6) In this study, five crop residues were selected to determine their multi-pollutants emission factors, but there are other major crop resides not considered in this study, e.g. sugarcane, barley etc. There should be a reason to explain why such crop residues were not considered and how to determine the emission inventories in some provinces (with high sugarcane and barley production).

7) There is some typo errors found in the manuscript: Line 215, "costume-built" should be "custom-built" Line 330, "Corp straw" should be "Crop straw"

---

## Author Comment (AC1) · 29 Dec 2016

**1 Review comments: "Multi-pollutants emissions from the burning of major agricultural residues in China and the related health-economic effect assessment" by Li C. et al. This paper describes results from experimental investigations on EFs of multi-pollutants from crop residues open burning in China, and try to estimate the health-economic effect under different scenarios. Considering the limited EFs for crop residues open burning, new emission data for various types of biomass are always welcome addition to the literature, and should be useful to air quality communities. This paper is reasonably well written. But there are a number of revisions that should be addressed prior to publication. Reply: Thanks for your reviewing! Question 1: When**

comparing EFs in this study with literature data, I notice that only comparable data from literature is included (for example, Line 505 to 507 when comparing OC and EC EFs, Line 605 to 608 when comparing EFs of PAHs, etc). However, it is already known and acknowledged that the EFs of crop residues burning could have a wide range due to different combustion condition, properties of biomass etc. . .(McMeeking et al., 2009; Reid et al., 2005). Both similarity and differences in EFs should be discussed. Moreover, due to crop residues used in this study were dehydrated at 100 degrees for 24h (thus had much lower moisture content compared with elsewhere), it tends to have a much lower EFs of incomplete combustion product, such as PM and OC from chamber studies (Chen et al., 2010; Hayashi et al., 2014). However, for biomass open burning combustion intensity may be higher than those from chamber studies, and thus this would result in a lower EF. When applying EFs from this study to estimate emission inventories, I would like to suggest that the uncertainties from the effect of moisture content and the burning style should be discussed in this paper. Answer 1: Thanks so much for your suggestion. Chamber burn study has definite advantage over the field burning one, as combustion in the field will be affected by many influence factors including but not limited to meteorological condition, terrain, diffusion, air supply, contamination, fuel issue (fuel type, water content, weight), and burning intensity or fire characters etc. However, after phasing out all the influence variables, how to conduct more exercisable and comparable burning experiment in the lab, and how well the practical chamber burn study can represent field burning should be considered. In general, when combustion efficiency (CE) differences were taken into account, emission factors measured from filed will be reasonably agree with that from chamber burn (Dhammapala et al., 2007) . We added more comparison with other studies, and we put the activity data and parameters for the biomass fuel (dry matter fraction, burning efficiency) into the consideration in the final uncertainty assessment for the emission inventories (Line 1071~1105). Lin 561: add in the manuscript"which is consistent with the conclusion from Lee et al. (2015) and Giordano et al. (2015)." Line 564: add in the manuscript"since EFs in smoke PM1.0 were seldom reported, only smoke PM2.5

or total particulate matter emissions were collected, which were comparable with the results in this work" Line 567add in the manuscript "were in range of 3.25∼15.16 and 3.04∼13.20 g kg-1 for the five kinds of crop straws, a high ratio of PM1.0/PM2.5 was observed to be over 90 wt.%, which was in line with size distribution analysis of smoke particles given in Fig. S3 (SI)" Line 570: delete "8.99 ± 5.55 and 7.91 ± 4.67 g kg-1 for the five kinds of crop straws, and over 70 wt.% of SPM was organic components (OM and EC), with average of 73.4 wt.% in PM2.5 and 71.3 wt.% in PM1.0." Line 573: add in the manuscript "Li et al. (2007) measured the emissions from field burning of crop straws via CMB method, PM2.5 EFs for wheat and corn straw were estimated to be 7.6±4.1 and 11.7±1.0 g kg-1 (dry basis, MCE > 0.9), which were higher and presented more uncertainties than our result. As study ever found a positive relationship between particulate EFs and moisture content of agricultural residue (Hayashi et al., 2014), it was reasonable that combustion of the dehydrated crop straw produced less smoke aerosol in this work. Hayashi et al. (2014) measured particulate EFs to be 2.2 and 15.0 g kg-1 for rice and wheat straw of ∼10 wt.% moisture content, while corresponded EFs increased to 9.1 and 19.5 g kg-1 when water content of straw was ∼20 wt.%, and the linear equations between smoke EFs and straw moisture content were furtherly proposed. However, the simple linearity and its application scope should be doubted, as Hayashi et al. only considered two water content levels (10 wt.% vs 20 wt.%) and disregarded influence of combustion efficiency for the fires. PM2.5 EFs given by Dhammapala et al (2006, 2007a, b) were 4.7±0.4 g kg-1 for wheat straw and 12.1±1.4 g kg-1 for herbaceous fuel that were burnt using a chamber under flaming phase, and negative response for particulate EFs to combustion efficiency was observed. After all, smoke EFs vary with fires depend on fuel type and moisture, combustion phase, environmental conditions, and some other variables (Reid et al., 2005b)." Line 591: add in the manuscript "The carbonaceous materials (Organic matter and EC) are dominated in SPM, accounting for about 73.4 wt.% for PM2.5 and 71.3 wt.% for PM1.0 on average." Line 594: add in the manuscript "and Li et al. (2016) ever measured OM/OC ratio as ∼1.3 for fresh smoke particles via volatility analysis. EFs of EC and OC from this

work were consistent with most studies, average OC EFs were 4.21 and 3.58 g kg-1 in smoke PM2.5 and PM1.0, and the corresponded EC EFs were 1.09 and 1.01 g kg-1, respectively. These values fell within the ranges (0.9∼9.3 g kg-1 for OC and 0.2∼1.7 g kg-1 for EC) found in other similar sources (Dhammapala et al., 2007; Hayashi et al., 2014; Li et al., 2007; May et al., 2014)." Line 603: add in the manuscript "It was ever reported chamber burn study may overestimate EC EFs due to a misassigned OC-EC split for the heavily mass loaded filter samples (Dhammapala et al., 2007b). Moreover, carbon measurement based on TOT method with NIOSH protocol may overestimate OC fraction by sacrificing EC part compared with that of TOR (Thermal-Optical Reflectance) method with IMPROVE program (Han et al., 2016)." Line 608: delete "EFs of EC and OC from this work agree well with previous study, average EFs of OC were 4.21 and 3.58 g kg-1 in smoke PM2.5 and PM1.0, and the values for EC were 1.09 and 1.01 g kg-1" Line 637: add in the manuscript "To our knowledge, seldom study ever reported source specific EFs of char- and soot-EC for crop straw burnings. Here, particulate char- and soot-EC EFs in fine mode were estimated to be 0.56 ∼ 1.76 and 0.05 ∼ 0.42 g kg-1, while char- and soot-EC EFs in smoke PM1.0 were 0.51 ∼ 1.67 and 0.06 ∼ 0.41 g kg-1, respectively." Line 646: add in the manuscript "are also fuel types and PM size dependent.. Generally, char-EC/soot-EC is also controlled by combustion mode or even moisture content of biomass fuel, and biomass burning by smoldering at low temperatures results in high char-EC/soot-EC. Chuang et al. (2013) reported char-EC/soot-EC in smoke PM2.5 was 9.4±3.8 for biomass burning (BB), and Cao et al. (2005) proposed the ratio to be 11.6 for BB sources. These values were larger than the present study, as we estimated char-EC/soot-EC in PM2.5 to be 7.28±1.98 on average. It can be explained by different techniques for EC measurement, char-EC and soot-EC were mostly measured using TOR-IMPROVE method, while TOT-NIOSH method used in this study will overestimate PC fraction in OC-EC split, resulting in less char-EC fraction (EC1-PC) and lower char-EC/soot-EC ratio. Nonetheless, the results were still comparable for the two methods (Han et al., 2016). The char-EC/soot-EC ratio was 6.29 in PM1.0, which was smaller than that in smoke PM2.5, the result indicates

that SPM comprises a considerable amount of char-EC and char particle has a larger size than soot, in consistent with the conclusion that soot particles are mainly tens of nanometers in size and cluster together into loose aggregates of hundred nanometers, while char particles were reported to be larger with diameter in the range of 1~100 $\mu$m" Line 680: add in the manuscript "Oxalic acid is the dominated dicarboxylic acids measured in the ambient environment and biomass burning aerosol (Falkovich et al., 2005; Kundu et al., 2010), and oxalic acid EF was measured to be 2.2 ~ 4.8 and 1.6 ~ 3.6 mg kg-1 for smoke PM2.5 and PM1.0 in present work." Line 752: add in the manuscript "Statistical analysis showed WSA/NH4+ was 0.16 ± 0.03 and 0.18 ± 0.06 in smoke PM1.0 and PM2.5, respectively, which were almost one order of magnitude larger than that in the ambient aerosol (Liu and Bei, 2016; Tao et al., 2016). Tao et al. (2016) ever measured the ratio as a function of particle size during NPF days in Shanghai, and a noticeable enrichment of aminiums for ultrafine particles (<56 nm) was observed with WSA/NH4+ over 0.2, highlighting the competitive role for amines to ammonia in particle nucleation and initial growth of the nuclei, the ratio was then decreased with the increasing particle size, and the final increasing trend was found after ~ 1.0 $\mu$m, and average WSA/NH4+ for ambient bulk PM1.0 and PM2.5 were 3.2% and 3.5% , respectively." Line 752: add in the manuscript "Hays et al. (2005) estimated total EFs of 16 PAHs to be 3.3 mg Kg-1 in wheat straw burning PM2.5. Korenaga et al. (2001) measured PAHs EFs from rice straw burning to be 1.9 mg Kg-1 in particulate phase, while the value from Jenkins et al. (1996) was 16 mg Kg-1. Dhammapala et al. (2007b) found negative linear response for biomass burning source PAHs emissions to burning efficiency, and under flaming combustion, particulate total 16 PAHs EFs were 2 ~ 4 mg Kg-1. Zhang et al. (2011) simulated burning of rice, corn, and wheat straws, the corresponded PAHs EFs were measured as 1.6, 0.9, and 0.7 mg Kg-1 in fine smoke particles, respectively. Great uncertainties for PAHs EFs were evident that relied on burning phase, fuel types, moisture content, and also measurement techniques." Line 806: add in the manuscript "EFs for the sum phenols were 9.7 ~ 41.5 and 7.7 and 23.5 mg Kg-1 for smoke PM2.5 and PM1.0, respectively. Dhammapala et al. (2007a) estimated particulate methoxyphenols emissions to be 35 $\pm$ 24 mg Kg-1 for wheat straw burning, while Hays et al. (2005) measured the same compounds to be 6.8 mg Kg-1. Carbonaceous materials like PAHs and Phenols or aromatic and phenolic deviates are the main chromophores in the atmosphere, and the considerable fractions of PAHs and Phenols justify biomass burning as a significant source of brown carbon (Laskin et al., 2015), study has proved $\sim$ 50% of the light absorption in the solvent-extractable fraction of smoke aerosol can be attributed to these strong BrC chromophores (Lin et al., 2016). " Line 866: add in the manuscript "in line with result from domestic burning of wood and field investigation of crop straw burning (Li et al., 2007; Zhang et al., 2012)" Line 990: add in the manuscript "Qin and Xie (2011, 2012) developed national carbonaceous aerosol emission inventories from biomass open burning for multi-years with dynamic burning activity, they believed BC and OC emissions followed an exponential growth from 14.03 and 57.37 Gg in 1990 to 116.58 and 476.77 Gg in 2009. Cao et al. (2006, 2011) calculated smoke aerosol emissions from biomass burning in China for 2000 and 2007 using the same activity data from BAU-I scenarios, national OC and EC emissions were reported to be 425.9 and 103.0 Gg in 2000, however, no evident changes were found for the emissions in 2007, which were assessed to be 433.0 and 104.0 Gg. Huang et al. (2012b) estimated crop burning in the fields with unified EFs and burning rate ($\sim$6.6 %) for all kinds of crops across China in 2006, the estimated annual agricultural fire emissions were about 270, 100, and 30 Gg for PM2.5, OC, and BC, respectively. In present work, agricultural fire PM2.5 emissions in 2012 were allocated into six zones, average contribution in percentage for each zone was compared: NPC (23.1 %) $\geq$ NC (21.6 %) > PRD (18.4 %) $\geq$ CC (18.2 %) > WC (9.8 %) > YRD (8.8 %). Furtherly, contribution for summertime emissions was: NPC (35.5 %) > CC (28.8 %) $\geq$ PRD (21.1 %) > YRD (9.1 %)> WC (5.4 %) > NC (0.1 %), and for autumn harvest emissions: NC (27.8 %) > NPC (19.6 %) > PRD (17.6 %) > CC (15.1 %) > WC (11.1 %) > YRD (8.8 %)" Line 1011: add in the manuscript "It was obviously that the North Plain experienced extensive crop fire emissions during the whole harvest periods, where PM2.5, PM1.0, OC, and BC emissions in 2012 were 233.6, 209.8, 102.3, and 29.4 Gg

on average. Liu et al. (2015) developed emission inventories from agricultural fires in the North Plain based on MODIS fire radiative power, emission for PM2.5, OC, and BC in 2012 was reported to be 102.3, 37.4, and 13.0 Gg, respectively. However, EFs were also treated as unified values (e.g., Crop burning EFs for PM2.5, OC, and BC was 6.3, 2.3, and 0.8 g Kg-1) in the work of Liu et al. (2015) that was cited directly from Akagi et al. (2011) without considering fuel type dependence of EFs. Zhao et al. (2012) established comprehensive anthropogenic emission inventories for Huabei Region including the North Plain, Inner Mongolia, and Liaoning province, all crop straws were assumed to be burnt in the field, resulting in much more emissions of 446 Gg OC and 160 Gg BC in 2003. A specific temporal pattern for agricultural fire emissions was observed in the Northeast of China (Heilongjiang, Liaoning, and Jilin), where the open burning were mainly occurred in autumn harvest to produce great amount of pollutants (217.5 Gg PM2.5, 89.4 Gg OC, and 29.7 Gg EC), while emissions in the summertime can be neglected."

Question 2: China maps used in Figure 7 are incomplete, part of Xinjiang and Tibet is missing from maps in Figure 7, there should be a reason to explain this. Answer 2: Thanks for your comment. Figure 7 displays geographic distribution of pollutants which is drawn by ArcGIS software, the final graph was designed to contain the figures for all the five versions and also the average one, the map was clipped and zoomed in to show more detailed information of subgraph (the legend). Moreover, information of provincial emissions for Xinjiang, Tibet, and Heilongjiang was not lost. Question 3: Line 228, the definition of MCE (Modified Combustion Efficiency) should be given. MCE=$\Delta CO2/(\Delta CO2+\Delta CO)$, where $\Delta CO2$ and $\Delta CO$ are the excess molar mixing ratios of CO2 and CO, and thus cannot be monitored directly, as stated on Line 228. Answer 3: Thanks for your reminding, definition of MCE has been corrected and added in Line 229. Line 238: add in the manuscript"with CO and CO2 measuring to determine the burning phase and ensure the repeatability. MCE is defined as $\Delta CO2/(\Delta CO2+\Delta CO)$, where $\Delta CO2$ and $\Delta CO$ are the excess molar mixing ratios of CO2 and CO (Reid et al., 2005b)."

Question 4: When stating there are "significant differences" between means, the statistical tests should be conducted and the results should be also given. Otherwise, there are no significant evidence that one mean differs from the other. The statistical test should be conducted in Line 495, 519, 617, 766... Answer 4: Thanks for your comment, we have added the significance test for the corresponded statistical conclusions in the manuscript, e.g., from multivariate statistical analysis considering fuel type and size range effect on the chemical compositions for smoke PM2.5 and PM1.0, significant difference was found (P<0.05 at 95% CI) Table 1 Multivariate statistical analysis for chemical compositions of smoke PM2.5 and PM1.0 from five agricultural residues burning Species PM2.5 PM1.0 Emission factor Mass fraction Emission factor Mass fraction PM2.5 0.000 PM1.0 0.000 0.650 0.000 OC 0.000 0.275 0.000 0.170 EC 0.000 0.013 0.010 0.189 WSOA 0.004 0.040 0.003 0.049 WSA 0.000 0.011 0.000 0.015 WSI 0.001 0.000 0.000 0.000 SO42- 0.000 0.000 0.000 0.020 Cl- 0.000 0.000 0.000 0.000 NH4+ 0.000 0.000 0.000 0.000 K+ 0.000 0.000 0.000 0.000 THM 0.000 0.030 0.000 0.017 PAHs 0.001 0.008 0.001 0.037 Phenols 0.000 0.019 0.006 0.006 Note: SPSS analysis, P<0.05 means significant difference at 95% confidence interval (CI)

Question 5: Although several ways to estimate uncertainties of the emissions were mentioned in Section 3.3.4 (Line 827 to Line), it is not clear which method is used in this study. For the emission inventory in this study, a discussion of the overall inventory uncertainty is needed and this could be given by considering the uncertainties in each of the terms in the inventory (Eq 5). Answer 5: Thanks for your comment, in the previous manuscript, we only considered the uncertainties for the average emission inventory from the 5 versions using the uncertainty propagation calculation as: U_total=√(∑_Θ(U_i×x_i)ãĂŮˆ2 )/(∑_Θx_i)[1]U_total =√(∑_ΘU_Θ2)[2]Where U_i is uncertainty in percentage for variate i, x_i is the variate, and equation

Table 2 Uncertainties for the national smoke PM emissions in 2012 (pollutant emission in unit of Gg/yr, 95% CI in percentage) Species BAU-I BAU-II EM-I EM-II NDRC Average PM2.5 1001.1 (-52.3% , 73.5%) 835.4 (-48.7% , 68.8%) 1211.9 (-63.6% ,

84.3%) 738.4 (-55.9% , 74.3%) 1241.7 (-46.2% , 65.1%) 1005.7 (-24.6% , 33.7%)
PM1.0 897.5 (-51.6% , 73.0%) 748.6 (-48.4% , 68.6%) 1087.1 (-62.9% , 83.8%)
661.8 (-55.5% , 74.1%) 1111.9 (-45.7% , 64.7%) 901.4 (-24.4% , 33.5%) OC 429.5
(-50.5% , 71.5%) 361.0 (-48.9% , 69.2%) 519.3 (-61.4% , 81.8%) 318.8 (-55.6% ,
74.1%) 533.2 (-47.1% , 66.7%) 432.4 (-24.2% , 33.3%) EC 133.6 (-52.1% , 73.6%)
111.4 (-50.1% , 71.0%) 162.7 (-63.3% , 84.3%) 98.1 (-56.8% , 75.7%) 165.0 (-46.7%
, 66.0%) 134.2 (-24.8% , 34.0%) char-EC 112.8 (-51.1% , 73.3%) 93.8 (-49.4% ,
69.9%) 137.2 (-63.1% , 84.0%) 82.8 (-60.8% , 80.7%) 139.2 (-46.2% , 65.4%) 113.1
(-24.8% , 34.1%) soot-EC 20.8 (-53.7% , 74.7%) 17.5 (-55.3% , 77.6%) 25.5 (-65.9%
, 87.4%) 15.2 (-61.8% , 81.9%) 25.7 (-50.6% , 71.1%) 21.0 (-26.3% , 35.9%) WSOA
24.4 (-68.5% , 86.2%) 21.9 (-75.7% , 95.2%) 29.7 (-78.7% , 96.2%) 18.8 (-77.8% ,
95.4%) 30.8 (-67.5% , 85.1%) 25.1 (-33.3% , 41.4%) WSA 5.8 (-62.8% , 82.1%) 4.9
(-65.9% , 84.1%) 7.0 (-73.9% , 93.2%) 4.2 (-69.3% , 86.3%) 7.2 (-58.7% , 75.9%)
5.8 (-30.1% , 38.5%) WSI 250.0 (-54.4% , 77.2%) 204.5 (-47.5% , 67.4%) 301.8
(-66.9% , 89.3%) 182.3 (-56.1% , 74.8%) 310.3 (-46.9% , 66.4%) 249.8 (-25.4% ,
34.9%) THM 8.7 (-56.2% , 77.5%) 7.2 (-52.8% , 71.4%) 10.6 (-67.5% , 88.3%) 6.4
(-61.2% , 79.5%) 10.6 (-50.8% , 69.4%) 8.7 (-26.6% , 35.6%) PAHs 0.5 (-55.2% ,
75.7%) 0.4 (-52.4% , 72.2%) 0.6 (-66.5% , 86.8%) 0.4 (-58.8% , 76.9%) 0.6 (-49.3%
, 67.8%) 0.5 (-26.0% , 34.9%) Phenols 2.7 (-56.1% , 77.6%) 2.3 (-51.4% , 70.6%)
3.3 (-67.3% , 88.3%) 2.0 (-59.9% , 78.4%) 3.4 (-48.7% , 67.1%) 2.7 (-26.1% , 35.1%)
Line 1065: add in the manuscript "The uncertainties in emission inventory can also be
estimated by comparing different emission inventories for the same region and period
(Ma and Van Aardenne, 2004)" Line 1071: add in the manuscript"we investigated
the uncertainties of multi-pollutants emissions for agricultural residue open burning
using Monte Carlo Simulation. Detailed methodology was referred to Qin and Xie
(2011). We followed the assumption: a normal distribution with coefficient of variation
(CV) of 30% for the official statistics (e.g., crop production and GDP economic data
obtained from Statistic Yearbooks, field burning rates for agricultural straw derived
from NDRC report, etc.), a normal distribution with 50% CV for open burning rates

from literature (BAU-I and BAU-II), and a uniform distribution with $\pm$ 30% deviation for the rest activity data (crop-to-residue ratio, dry matter fraction, and burning efficiency). Regarding the emission factors, Bond et al. (2004) assumed that most particulate EFs followed lognormal distributions with CV of $\pm$ 50% for domestic EFs, and of $\pm$ 150% for EFs obtained from foreign studies. Here, we applied the CV of smoke EFs as we measured ones, which were chemical species and fuel type dependent. With randomly selected values within the respective probability density functions (PDFs) of EFs and activity data for each biomass type, Monte Carlo simulation was implemented for 10,000 times, and the uncertainties in national yearly multi-pollutants emissions at 95% CI were obtained for all the 5 versions. Afterwards, uncertainties for the average emission inventories were assessed using the propagation of uncertainty calculation that suggested by IPCC (1997) (method in SI), and all the emission uncertainties were presented in percentage in Table 6. Emissions for water soluble aminiums and organic acids had the vast uncertainties, due to their large deviation in EFs compared with other smoke species. Besides, emissions of BAU versions were more accurate than EM versions, because of more uncertainty addition in the burning rates conversion using economic data for EM versions. Otherwise, burning rates derived from NDRC report were assumed to have less uncertainty, resulting in the least uncertainties in smoke emission assessments. On average of all the 5 versions, mean, 2.5th percentile, and 97.5th percentile values for smoke PM2.5 emissions in 2012 were 1005.7, 758.3, and 1344.6 Gg, respectively. As to OC emissions, mean, 2.5th percentile, and 97.5th percentile values were 432.4, 327.8, and 576.4 Gg, the figure for EC was 134.2, 100.9, and 187.9 Gg. Therefore, the overall propagation of uncertainties for smoke PM2.5, OC, and EC at 95% CI was [-24.6%, 33.7%], [-24.4%, 33.5%], and [-24.2%, 33.3%], respectively. The uncertainties for OC and EC emissions were much less than the work of Qin and Xie (2011), in which emission and uncertainties were 266.7 Gg [-55.9%, 96.1%] for OC and 66. 9 Gg [-53.9%, 92.6%] for EC in 2005" Question 6: Line 215, "costume-built" should be "custom-built"; Citation formatting and styling errors should be corrected carefully. For example, Line 360, References should be cited with

publication year. Chen et al. (2001) is cited under Cao's publication... Line 374, Qin et al. (2012) is cited, but is missing from References list. Answer 6: "custom-built" has been corrected in Line 215, citation errors have been carefully checked and modified. Line 224: "custom-built" has been corrected Line 65: "Andreae and Merlet, 2001" has been corrected Line 71:"Qin and Xie, 2012"has been corrected and added in the reference list Line 79: "Andreae and Merlet, 2001" has been corrected Line 81:"Qin and Xie, 2012"has been corrected Line 94: "Arora and Jain, 2015" has been corrected Line 123: "Qin and Xie, 2011, 2012"has been corrected Line 148: "Ostro and Chestnut, 1998" has been corrected Line 182: "Reddy and Venkataraman, 2000" has been corrected Line 226: "Zhang et al., 2008a, 2011" has been corrected Line 404:"CAREI, 2000"deleted Line 435: "Cermak and Kuntti, 2009" has been corrected Line 490: "Bell and Hipfner, 1997" has been corrected Line 522: "Aunan and Pan, 2004" has been corrected Line 625: "Arora and Jain, 2015" has been corrected Line 633: "Andreae and Gelencsér, 2006" has been corrected Line 673: "Arora and Jain, 2015" has been corrected Line 702: "Andreae and Gelencsér, 2006" has been corrected Line 711: "Qiu and Zhang, 2012" has been corrected Line 715:"Lee and Wexler, 2013"has been corrected Line 718:"Schade and Crutzen, 1995"has been corrected Line 744: "Arey and Atkinson, 2003" has been corrected Line 798:"Berndt and Boge, 2006"has been corrected Line 849: "Amdur and Chen, 1989" has been corrected

Please also note the supplement to this comment:
http://www.atmos-chem-phys-discuss.net/acp-2016-651/acp-2016-651-AC1-supplement.pdf

**Supplement:**

**#1 Review comments:** "Multi-pollutants emissions from the burning of major agricultural residues in China and the related health-economic effect assessment" by Li C. et al.

This paper describes results from experimental investigations on EFs of multi-pollutants from crop residues open burning in China, and try to estimate the health-economic effect under different scenarios. Considering the limited EFs for crop residues open burning, new emission data for various types of biomass are always welcome addition to the literature, and should be useful to air quality communities. This paper is reasonably well written. But there are a number of revisions that should be addressed prior to publication.

Reply: Thanks for your reviewing!

**Question 1**: When comparing EFs in this study with literature data, I notice that only comparable data from literature is included (for example, Line 505 to 507 when comparing OC and EC EFs, Line 605 to 608 when comparing EFs of PAHs, etc). However, it is already known and acknowledged that the EFs of crop residues burning could have a wide range due to different combustion condition, properties of biomass etc…(McMeeking et al., 2009; Reid et al., 2005). Both similarity and differences in EFs should be discussed. Moreover, due to crop residues used in this study were dehydrated at 100 degrees for 24h (thus had much lower moisture content compared with elsewhere), it tends to have a much lower EFs of incomplete combustion product, such as PM and OC from chamber studies (Chen et al., 2010; Hayashi et al., 2014). However, for biomass open burning combustion intensity may be higher than those from chamber studies, and thus this would result in a lower EF. When applying EFs from this study to estimate emission inventories, I would like to suggest that the uncertainties from the effect of moisture content and the burning style should be discussed in this paper.

**Answer 1:** Thanks so much for your suggestion. Chamber burn study has definite advantage over the field burning one, as combustion in the field will be affected by many influence factors including but not limited to meteorological condition, terrain, diffusion, air supply, contamination, fuel issue (fuel type, water content, weight), and burning intensity or fire characters etc. However, after phasing out all the influence variables, how to conduct more exercisable and comparable burning experiment in the lab, and how well the practical chamber burn study can represent field burning should be considered. In general, when combustion efficiency (CE) differences were taken into account, emission factors measured from filed will be reasonably agree with that from chamber burn (Dhammapala et al., 2007) . We added more comparison with other studies, and we put the activity data and parameters for the biomass fuel (dry matter fraction, burning efficiency) into the consideration in the final uncertainty assessment for the emission inventories (Line 1071~1105).

Lin 561: add in the manuscript "which is consistent with the conclusion from Lee et al. (2015) and Giordano et al. (2015)."

Line 564: add in the manuscript "since EFs in smoke $PM_{1.0}$ were seldom reported, only smoke $PM_{2.5}$ or total particulate matter emissions were collected, which were comparable with the results in this work"

Line 567add in the manuscript "were in range of 3.25~15.16 and 3.04~13.20 g kg$^{-1}$ for the five kinds of crop straws, a high ratio of $PM_{1.0}/PM_{2.5}$ was observed to be over 90 wt.%, which was in line with size distribution

analysis of smoke particles given in Fig. S3 (SI)"

Line 570: delete "8.99 ± 5.55 and 7.91 ± 4.67 g kg$^{-1}$ for the five kinds of crop straws, and over 70 wt.% of SPM was organic components (OM and EC), with average of 73.4 wt.% in PM$_{2.5}$ and 71.3 wt.% in PM$_{1.0}$."

Line 573: add in the manuscript "Li et al. (2007) measured the emissions from field burning of crop straws via CMB method, PM$_{2.5}$ EFs for wheat and corn straw were estimated to be 7.6±4.1 and 11.7±1.0 g kg$^{-1}$ (dry basis, MCE > 0.9), which were higher and presented more uncertainties than our result. As study ever found a positive relationship between particulate EFs and moisture content of agricultural residue (Hayashi et al., 2014), it was reasonable that combustion of the dehydrated crop straw produced less smoke aerosol in this work. Hayashi et al. (2014) measured particulate EFs to be 2.2 and 15.0 g kg$^{-1}$ for rice and wheat straw of ~10 wt.% moisture content, while corresponded EFs increased to 9.1 and 19.5 g kg$^{-1}$ when water content of straw was ~20 wt.%, and the linear equations between smoke EFs and straw moisture content were furtherly proposed. However, the simple linearity and its application scope should be doubted, as Hayashi et al. only considered two water content levels (10 wt.% *vs* 20 wt.%) and disregarded influence of combustion efficiency for the fires. PM$_{2.5}$ EFs given by Dhammapala et al (2006, 2007a, b) were 4.7±0.4 g kg$^{-1}$ for wheat straw and 12.1±1.4 g kg$^{-1}$ for herbaceous fuel that were burnt using a chamber under flaming phase, and negative response for particulate EFs to combustion efficiency was observed. After all, smoke EFs vary with fires depend on fuel type and moisture, combustion phase, environmental conditions, and some other variables (Reid et al., 2005b)."

Line 591: add in the manuscript "The carbonaceous materials (Organic matter and EC) are dominated in SPM, accounting for about 73.4 wt.% for PM$_{2.5}$ and 71.3 wt.% for PM$_{1.0}$ on average."

Line 594: add in the manuscript "and Li et al. (2016) ever measured OM/OC ratio as ~1.3 for fresh smoke particles via volatility analysis. EFs of EC and OC from this work were consistent with most studies, average OC EFs were 4.21 and 3.58 g kg$^{-1}$ in smoke PM$_{2.5}$ and PM$_{1.0}$, and the corresponded EC EFs were 1.09 and 1.01 g kg$^{-1}$, respectively. These values fell within the ranges (0.9~9.3 g kg$^{-1}$ for OC and 0.2~1.7 g kg$^{-1}$ for EC) found in other similar sources (Dhammapala et al., 2007; Hayashi et al., 2014; Li et al., 2007; May et al., 2014)."

Line 603: add in the manuscript "It was ever reported chamber burn study may overestimate EC EFs due to a misassigned OC-EC split for the heavily mass loaded filter samples (Dhammapala et al., 2007b). Moreover, carbon measurement based on TOT method with NIOSH protocol may overestimate OC fraction by sacrificing EC part compared with that of TOR (Thermal-Optical Reflectance) method with IMPROVE program (Han et al., 2016)."

Line 608: delete "EFs of EC and OC from this work agree well with previous study, average EFs of OC were 4.21 and 3.58 g kg$^{-1}$ in smoke PM$_{2.5}$ and PM$_{1.0}$, and the values for EC were 1.09 and 1.01 g kg$^{-1}$"

Line 637: add in the manuscript "To our knowledge, seldom study ever reported source specific EFs of char- and soot-EC for crop straw burnings. Here, particulate char- and soot-EC EFs in fine mode were estimated to be 0.56 ~ 1.76 and 0.05 ~ 0.42 g kg$^{-1}$, while char- and soot-EC EFs in smoke PM$_{1.0}$ were 0.51 ~ 1.67 and 0.06 ~ 0.41 g kg$^{-1}$, respectively."

Line 646: add in the manuscript "are also fuel types and PM size dependent.. Generally, char-EC/soot-EC is

also controlled by combustion mode or even moisture content of biomass fuel, and biomass burning by smoldering at low temperatures results in high char-EC/soot-EC. Chuang et al. (2013) reported char-EC/soot-EC in smoke $PM_{2.5}$ was $9.4\pm3.8$ for biomass burning (BB), and Cao et al. (2005) proposed the ratio to be 11.6 for BB sources. These values were larger than the present study, as we estimated char-EC/soot-EC in $PM_{2.5}$ to be $7.28\pm1.98$ on average. It can be explained by different techniques for EC measurement, char-EC and soot-EC were mostly measured using TOR-IMPROVE method, while TOT-NIOSH method used in this study will overestimate PC fraction in OC-EC split, resulting in less char-EC fraction (EC1-PC) and lower char-EC/soot-EC ratio. Nonetheless, the results were still comparable for the two methods (Han et al., 2016). The char-EC/soot-EC ratio was 6.29 in $PM_{1.0}$, which was smaller than that in smoke $PM_{2.5}$, the result indicates that SPM comprises a considerable amount of char-EC and char particle has a larger size than soot, in consistent with the conclusion that soot particles are mainly tens of nanometers in size and cluster together into loose aggregates of hundred nanometers, while char particles were reported to be larger with diameter in the range of 1~100 μm"

Line 680: add in the manuscript "Oxalic acid is the dominated dicarboxylic acids measured in the ambient environment and biomass burning aerosol (Falkovich et al., 2005; Kundu et al., 2010), and oxalic acid EF was measured to be 2.2 ~ 4.8 and 1.6 ~ 3.6 mg $kg^{-1}$ for smoke $PM_{2.5}$ and $PM_{1.0}$ in present work."

Line 752: add in the manuscript "Statistical analysis showed WSA/$NH_4^+$ was 0.16 $\pm$0.03 and 0.18 $\pm$0.06 in smoke $PM_{1.0}$ and $PM_{2.5}$, respectively, which were almost one order of magnitude larger than that in the ambient aerosol (Liu and Bei, 2016; Tao et al., 2016). Tao et al. (2016) ever measured the ratio as a function of particle size during NPF days in Shanghai, and a noticeable enrichment of aminiums for ultrafine particles (<56 nm) was observed with WSA/$NH_4^+$ over 0.2, highlighting the competitive role for amines to ammonia in particle nucleation and initial growth of the nuclei, the ratio was then decreased with the increasing particle size, and the final increasing trend was found after ~ 1.0 μm, and average WSA/$NH_4^+$ for ambient bulk $PM_{1.0}$ and $PM_{2.5}$ were 3.2% and 3.5% , respectively."

Line 752: add in the manuscript "Hays et al. (2005) estimated total EFs of 16 PAHs to be 3.3 mg $Kg^{-1}$ in wheat straw burning $PM_{2.5}$. Korenaga et al. (2001) measured PAHs EFs from rice straw burning to be 1.9 mg $Kg^{-1}$ in particulate phase, while the value from Jenkins et al. (1996) was 16 mg $Kg^{-1}$. Dhammapala et al. (2007b) found negative linear response for biomass burning source PAHs emissions to burning efficiency, and under flaming combustion, particulate total 16 PAHs EFs were 2 ~ 4 mg $Kg^{-1}$. Zhang et al. (2011) simulated burning of rice, corn, and wheat straws, the corresponded PAHs EFs were measured as 1.6, 0.9, and 0.7 mg $Kg^{-1}$ in fine smoke particles, respectively. Great uncertainties for PAHs EFs were evident that relied on burning phase, fuel types, moisture content, and also measurement techniques."

Line 806: add in the manuscript "EFs for the sum phenols were 9.7 ~ 41.5 and 7.7 and 23.5 mg $Kg^{-1}$ for smoke $PM_{2.5}$ and $PM_{1.0}$, respectively. Dhammapala et al. (2007a) estimated particulate methoxyphenols emissions to be 35 $\pm$24 mg $Kg^{-1}$ for wheat straw burning, while Hays et al. (2005) measured the same compounds to be 6.8 mg $Kg^{-1}$. Carbonaceous materials like PAHs and Phenols or aromatic and phenolic deviates are the main chromophores in the atmosphere, and the considerable fractions of PAHs and Phenols

justify biomass burning as a significant source of brown carbon (Laskin et al., 2015), study has proved ~ 50% of the light absorption in the solvent-extractable fraction of smoke aerosol can be attributed to these strong BrC chromophores (Lin et al., 2016). "

Line 866: add in the manuscript "in line with result from domestic burning of wood and field investigation of crop straw burning (Li et al., 2007; Zhang et al., 2012)"

Line 990: add in the manuscript "Qin and Xie (2011, 2012) developed national carbonaceous aerosol emission inventories from biomass open burning for multi-years with dynamic burning activity, they believed BC and OC emissions followed an exponential growth from 14.03 and 57.37 Gg in 1990 to 116.58 and 476.77 Gg in 2009. Cao et al. (2006, 2011) calculated smoke aerosol emissions from biomass burning in China for 2000 and 2007 using the same activity data from BAU-I scenarios, national OC and EC emissions were reported to be 425.9 and 103.0 Gg in 2000, however, no evident changes were found for the emissions in 2007, which were assessed to be 433.0 and 104.0 Gg. Huang et al. (2012b) estimated crop burning in the fields with unified EFs and burning rate (~6.6 %) for all kinds of crops across China in 2006, the estimated annual agricultural fire emissions were about 270, 100, and 30 Gg for $PM_{2.5}$, OC, and BC, respectively. In present work, agricultural fire $PM_{2.5}$ emissions in 2012 were allocated into six zones, average contribution in percentage for each zone was compared: NPC (23.1 %) ≥ NC (21.6 %) > PRD (18.4 %) ≥ CC (18.2 %) > WC (9.8 %) > YRD (8.8 %). Furtherly, contribution for summertime emissions was: NPC (35.5 %) > CC (28.8 %) ≥ PRD (21.1 %) > YRD (9.1 %)> WC (5.4 %) > NC (0.1 %), and for autumn harvest emissions: NC (27.8 %) > NPC (19.6 %) > PRD (17.6 %) > CC (15.1 %) > WC (11.1 %) > YRD (8.8 %)"

Line 1011: add in the manuscript "It was obviously that the North Plain experienced extensive crop fire emissions during the whole harvest periods, where $PM_{2.5}$, $PM_{1.0}$, OC, and BC emissions in 2012 were 233.6, 209.8, 102.3, and 29.4 Gg on average. Liu et al. (2015) developed emission inventories from agricultural fires in the North Plain based on MODIS fire radiative power, emission for $PM_{2.5}$, OC, and BC in 2012 was reported to be 102.3, 37.4, and 13.0 Gg, respectively. However, EFs were also treated as unified values (e.g., Crop burning EFs for $PM_{2.5}$, OC, and BC was 6.3, 2.3, and 0.8 g $Kg^{-1}$) in the work of Liu et al. (2015) that was cited directly from Akagi et al. (2011) without considering fuel type dependence of EFs. Zhao et al. (2012) established comprehensive anthropogenic emission inventories for Huabei Region including the North Plain, Inner Mongolia, and Liaoning province, all crop straws were assumed to be burnt in the field, resulting in much more emissions of 446 Gg OC and 160 Gg BC in 2003. A specific temporal pattern for agricultural fire emissions was observed in the Northeast of China (Heilongjiang, Liaoning, and Jilin), where the open burning were mainly occurred in autumn harvest to produce great amount of pollutants (217.5 Gg $PM_{2.5}$, 89.4 Gg OC, and 29.7 Gg EC), while emissions in the summertime can be neglected."

**Question 2**: China maps used in Figure 7 are incomplete, part of Xinjiang and Tibet is missing from maps in Figure 7, there should be a reason to explain this.

**Answer 2:** Thanks for your comment. Figure 7 displays geographic distribution of pollutants which is drawn by ArcGIS software, the final graph was designed to contain the figures for all the five versions and also the average one, the map was clipped and zoomed in to show more detailed information of subgraph (the legend). Moreover, information of provincial emissions for Xinjiang, Tibet, and Heilongjiang was not lost.

**Question 3**: Line 228, the definition of MCE (Modified Combustion Efficiency) should be given. $MCE=\Delta CO_2/(\Delta CO_2+\Delta CO)$, where $\Delta CO_2$ and $\Delta CO$ are the excess molar mixing ratios of $CO_2$ and $CO$, and thus cannot be monitored directly, as stated on Line 228.

Answer 3: Thanks for your reminding, definition of MCE has been corrected and added in Line 229.

Line 238: add in the manuscript "with CO and $CO_2$ measuring to determine the burning phase and ensure the repeatability. MCE is defined as $\Delta CO_2/(\Delta CO_2+\Delta CO)$, where $\Delta CO_2$ and $\Delta CO$ are the excess molar mixing ratios of $CO_2$ and $CO$ (Reid et al., 2005b)."

**Question 4**: When stating there are "significant differences" between means, the statistical tests should be conducted and the results should be also given. Otherwise, there are no significant evidence that one mean differs from the other. The statistical test should be conducted in Line 495, 519, 617, 766…

**Answer 4**: Thanks for your comment, we have added the significance test for the corresponded statistical conclusions in the manuscript, e.g., from multivariate statistical analysis considering fuel type and size range effect on the chemical compositions for smoke $PM_{2.5}$ and $PM_{1.0}$, significant difference was found ($P<0.05$ at 95% CI)

Table 1 Multivariate statistical analysis for chemical compositions of smoke $PM_{2.5}$ and $PM_{1.0}$ from five agricultural residues burning

| Species | $PM_{2.5}$ | | $PM_{1.0}$ | |
| --- | --- | --- | --- | --- |
| | Emission factor | Mass fraction | Emission factor | Mass fraction |
| $PM_{2.5}$ | 0.000 | | | |
| $PM_{1.0}$ | 0.000 | **0.650** | 0.000 | |
| OC | 0.000 | **0.275** | 0.000 | **0.170** |
| EC | 0.000 | 0.013 | 0.010 | **0.189** |
| WSOA | 0.004 | 0.040 | 0.003 | 0.049 |
| WSA | 0.000 | 0.011 | 0.000 | 0.015 |
| WSI | 0.001 | 0.000 | 0.000 | 0.000 |
| $SO_4^{2-}$ | 0.000 | 0.000 | 0.000 | 0.020 |
| $Cl^-$ | 0.000 | 0.000 | 0.000 | 0.000 |
| $NH_4^+$ | 0.000 | 0.000 | 0.000 | 0.000 |
| $K^+$ | 0.000 | 0.000 | 0.000 | 0.000 |

| | | | | |
|---|---|---|---|---|
| THM | 0.000 | 0.030 | 0.000 | 0.017 |
| PAHs | 0.001 | 0.008 | 0.001 | 0.037 |
| Phenols | 0.000 | 0.019 | 0.006 | 0.006 |

Note: SPSS analysis, P<0.05 means significant difference at 95% confidence interval (CI)

**Question 5**: Although several ways to estimate uncertainties of the emissions were mentioned in Section 3.3.4 (Line 827 to Line), it is not clear which method is used in this study. For the emission inventory in this study, a discussion of the overall inventory uncertainty is needed and this could be given by considering the uncertainties in each of the terms in the inventory (Eq 5).

**Answer 5:** Thanks for your comment, in the previous manuscript, we only considered the uncertainties for the average emission inventory from the 5 versions using the uncertainty propagation calculation as:

$$U_{total} = \frac{\sqrt{\sum_i^n (U_i \times x_i)^2}}{\sum_i^n x_i} \qquad [1]$$

$$U_{total} = \sqrt{\sum_i^n U_i^2} \qquad [2]$$

Where $U_i$ is uncertainty in percentage for variate i, $x_i$ is the variate, and equation 1 for addition rule, equation 2 for multiplication rule.

We reassessed the uncertainties for each copy of the emission inventory via Monte Carlo simulation. We assumed: a normal distribution with coefficient variance (CV) of 30% for all the official statistics (crop production and economic data from Statistic Yearbook, filed burning ratio from NDRC report), a normal distribution with CV of 50% for open burning ratios collected from literature (BAU-I and BAU-II), and a normal distribution with CV of 30% for the rest activity data (crop-to-residue ratio, dry matter fraction, and burning efficiency). The uncertainties for smoke EFs are species-fuel type dependent, and we applied the uncertainties (95% CI) for smoke EF species as we measured ones. We ran 10000 Monte Carlo simulations to estimate the uncertainties for all the 5 versions of emission inventories, then we applied the uncertainty propagation calculation of equation 1 and 2 to assess the uncertainties for the average emissions, the uncertainties for national smoke PM emissions in 2012 were presented in the table below:

Table 2 Uncertainties for the national smoke PM emissions in 2012 (pollutant emission in unit of Gg/yr, 95% CI in percentage)

| Species | BAU-I | | BAU-II | | EM-I | | EM-II | | NDRC | | Average | |
|---|---|---|---|---|---|---|---|---|---|---|---|---|
| $PM_{2.5}$ | 1001.1 | (-52.3% , 73.5%) | 835.4 | (-48.7% , 68.8%) | 1211.9 | (-63.6% , 84.3%) | 738.4 | (-55.9% , 74.3%) | 1241.7 | (-46.2% , 65.1%) | 1005.7 | (-24.6% , 33.7%) |
| $PM_{1.0}$ | 897.5 | (-51.6% , 73.0%) | 748.6 | (-48.4% , 68.6%) | 1087.1 | (-62.9% , 83.8%) | 661.8 | (-55.5% , 74.1%) | 1111.9 | (-45.7% , 64.7%) | 901.4 | (-24.4% , 33.5%) |
| OC | 429.5 | (-50.5% , 71.5%) | 361.0 | (-48.9% , 69.2%) | 519.3 | (-61.4% , 81.8%) | 318.8 | (-55.6% , 74.1%) | 533.2 | (-47.1% , 66.7%) | 432.4 | (-24.2% , 33.3%) |
| EC | 133.6 | (-52.1% , 73.6%) | 111.4 | (-50.1% , 71.0%) | 162.7 | (-63.3% , 84.3%) | 98.1 | (-56.8% , 75.7%) | 165.0 | (-46.7% , 66.0%) | 134.2 | (-24.8% , 34.0%) |
| char-EC | 112.8 | (-51.1% , 73.3%) | 93.8 | (-49.4% , 69.9%) | 137.2 | (-63.1% , 84.0%) | 82.8 | (-60.8% , 80.7%) | 139.2 | (-46.2% , 65.4%) | 113.1 | (-24.8% , 34.1%) |
| soot-EC | 20.8 | (-53.7% , 74.7%) | 17.5 | (-55.3% , 77.6%) | 25.5 | (-65.9% , 87.4%) | 15.2 | (-61.8% , 81.9%) | 25.7 | (-50.6% , 71.1%) | 21.0 | (-26.3% , 35.9%) |
| WSOA | 24.4 | (-68.5% , 86.2%) | 21.9 | (-75.7% , 95.2%) | 29.7 | (-78.7% , 96.2%) | 18.8 | (-77.8% , 95.4%) | 30.8 | (-67.5% , 85.1%) | 25.1 | (-33.3% , 41.4%) |
| WSA | 5.8 | (-62.8% , 82.1%) | 4.9 | (-65.9% , 84.1%) | 7.0 | (-73.9% , 93.2%) | 4.2 | (-69.3% , 86.3%) | 7.2 | (-58.7% , 75.9%) | 5.8 | (-30.1% , 38.5%) |
| WSI | 250.0 | (-54.4% , 77.2%) | 204.5 | (-47.5% , 67.4%) | 301.8 | (-66.9% , 89.3%) | 182.3 | (-56.1% , 74.8%) | 310.3 | (-46.9% , 66.4%) | 249.8 | (-25.4% , 34.9%) |
| THM | 8.7 | (-56.2% , 77.5%) | 7.2 | (-52.8% , 71.4%) | 10.6 | (-67.5% , 88.3%) | 6.4 | (-61.2% , 79.5%) | 10.6 | (-50.8% , 69.4%) | 8.7 | (-26.6% , 35.6%) |
| PAHs | 0.5 | (-55.2% , 75.7%) | 0.4 | (-52.4% , 72.2%) | 0.6 | (-66.5% , 86.8%) | 0.4 | (-58.8% , 76.9%) | 0.6 | (-49.3% , 67.8%) | 0.5 | (-26.0% , 34.9%) |
| Phenols | 2.7 | (-56.1% , 77.6%) | 2.3 | (-51.4% , 70.6%) | 3.3 | (-67.3% , 88.3%) | 2.0 | (-59.9% , 78.4%) | 3.4 | (-48.7% , 67.1%) | 2.7 | (-26.1% , 35.1%) |

Line 1065: add in the manuscript "The uncertainties in emission inventory can also be estimated by comparing different emission inventories for the same region and period (Ma and Van Aardenne, 2004)"

Line 1071: add in the manuscript "we investigated the uncertainties of multi-pollutants emissions for agricultural residue open burning using Monte Carlo Simulation. Detailed methodology was referred to Qin and Xie (2011). We followed the assumption: a normal distribution with coefficient of variation (CV) of 30% for the official statistics (e.g., crop production and GDP economic data obtained from Statistic Yearbooks, field burning rates for agricultural straw derived from NDRC report, etc.), a normal distribution with 50% CV for open burning rates from literature (BAU-I and BAU-II), and a uniform distribution with $\pm 30\%$ deviation for the rest activity data (crop-to-residue ratio, dry matter fraction, and burning efficiency). Regarding the emission factors, Bond et al. (2004) assumed that most particulate EFs followed lognormal distributions with CV of $\pm 50\%$ for domestic EFs, and of $\pm 150\%$ for EFs obtained from foreign studies. Here, we applied the CV of smoke EFs as we measured ones, which were chemical species and fuel type dependent. With randomly selected values within the respective probability density functions (PDFs) of EFs and activity data for each biomass type, Monte Carlo simulation was implemented for 10,000 times, and the uncertainties in national yearly multi-pollutants emissions at 95% CI were obtained for all the 5 versions. Afterwards, uncertainties for the average emission inventories were assessed using the propagation of uncertainty calculation that suggested by IPCC (1997) (method in SI), and all the emission uncertainties were presented in percentage in Table 6. Emissions for water soluble aminiums and organic acids had the vast uncertainties, due to their large deviation in EFs compared with other smoke species. Besides, emissions of BAU versions were more accurate than EM versions, because of more uncertainty addition in the

burning rates conversion using economic data for EM versions. Otherwise, burning rates derived from NDRC report were assumed to have less uncertainty, resulting in the least uncertainties in smoke emission assessments. On average of all the 5 versions, mean, 2.5th percentile, and 97.5th percentile values for smoke $PM_{2.5}$ emissions in 2012 were 1005.7, 758.3, and 1344.6 Gg, respectively. As to OC emissions, mean, 2.5th percentile, and 97.5th percentile values were 432.4, 327.8, and 576.4 Gg, the figure for EC was 134.2, 100.9, and 187.9 Gg. Therefore, the overall propagation of uncertainties for smoke $PM_{2.5}$, OC, and EC at 95% CI was [-24.6%, 33.7%], [-24.4%, 33.5%], and [-24.2%, 33.3%], respectively. The uncertainties for OC and EC emissions were much less than the work of Qin and Xie (2011), in which emission and uncertainties were 266.7 Gg [-55.9%, 96.1%] for OC and 66. 9 Gg [-53.9%, 92.6%] for EC in 2005"

**Question 6**: Line 215, "costume-built" should be "custom-built"; Citation formatting and styling errors should be corrected carefully. For example, Line 360, References should be cited with publication year. Chen et al. (2001) is cited under Cao's publication…   Line 374, Qin et al. (2012) is cited, but is missing from References list.

**Answer 6**: "custom-built" has been corrected in Line 215, citation errors have been carefully checked and modified.

Line 224: "custom-built" has been corrected

Line 65: "Andreae and Merlet, 2001" has been corrected

Line 71: "Qin and Xie, 2012" has been corrected and added in the reference list

Line 79: "Andreae and Merlet, 2001" has been corrected

Line 81: "Qin and Xie, 2012" has been corrected

Line 94: "Arora and Jain, 2015" has been corrected

Line 123: "Qin and Xie, 2011, 2012" has been corrected

Line 148: "Ostro and Chestnut, 1998" has been corrected

Line 182: "Reddy and Venkataraman, 2000" has been corrected

Line 226: "Zhang et al., 2008a, 2011" has been corrected

Line 404: "CAREI, 2000" deleted

Line 435: "Cermak and Kuntti, 2009" has been corrected

Line 490: "Bell and Hipfner, 1997" has been corrected

Line 522: "Aunan and Pan, 2004" has been corrected

Line 625: "Arora and Jain, 2015" has been corrected

Line 633: "Andreae and Gelencsér, 2006" has been corrected

Line 673: "Arora and Jain, 2015" has been corrected

Line 702: "Andreae and Gelencsér, 2006" has been corrected

Line 711: "Qiu and Zhang, 2012" has been corrected

Line 715: "Lee and Wexler, 2013" has been corrected

Line 718: "Schade and Crutzen, 1995" has been corrected

Line 744: "Arey and Atkinson, 2003" has been corrected

Line 798: "Berndt and Boge, 2006" has been corrected

Line 849: "Amdur and Chen, 1989" has been corrected

**#2 Review comments:** "Multi-pollutants emissions from the burning of major agricultural residues in China and the related health-economic effect assessment" by Li C. et al..

This study investigates the emission factors of multi-pollutants from five major crop residues in China, and tries to estimate emission inventory and their corresponding health-economic effect. This paper is well organized and presents some interesting data. However, detailed explanations about the design should be given to ensure the data quality.

Answer: Thanks for your carefully review!

**Question 1**: When the crop residues were dehydrated at 100 degree C for 24 hrs, what are the impacts to the emission factors and PM compositions?

**Answer 1**: Pretreatment of biomass fuel in burning simulation is a practical and necessary procedure to ensure the result can be comparable with other studies under the defined conditions, like dehydration at 100 $^o$C for 24 hrs to ensure water content of the residue within 2 wt. %, which has been applied in many burning studies (Hayashi et al., 2014; Huo et al., 2016; Li et al., 2015; Oanh et al., 2011; Zhang et al., 2008; Zhang et al., 2011). Water content of residue is a variate response to the smoke particle emissions and burning efficiency of biofuel (Hayashi et al., 2014; Oanh et al., 2011), and residue moistness has been shown to be positively correlated with particle emissions in range of 5~35 wt.%. However, empirical emission inventory calculation has to simplify the water content of residues to get the final dry matter, thus we designed our combustion method by dehydration the biomass fuel in front, besides, the residues we collected from filed have water content of less than 5 wt.% on average (wheat: 3.7 wt.%, rice: 4.4 wt.%, corn: 6.3 wt.%, soybean: 5.1 wt.%, cotton: 4.6 wt.% ), the dehydration to get water content within 2 wt.% will have much weaker influence on the chemical emissions.

**Question 2**: There are huge variations on EFs of crop residues, and they depend on lots of factors such as, sources of crop resides, burning temperature, burning efficiency etc. What are the differences between chamber study and open burning? As the burning last about 1 min only (in chamber study), can it represent the real open burning results? Moreover, what is the dilution ratio in the chamber study?

**Answer 2**: Chamber simulation has defined advantages over the field burning study, but it is also the paradox that chamber work can hardy reproduce the practical burning that be impacted by many influences in the field. To the emission factor measurements, we have to trade-off and simplify some impacts reasonably, for example burning efficiency, water content, and meteorological parameter etc, however, we don't mean these impacts can be neglected. Previous work told result from lab simulation will be reasonably agree with that from field burning under fixed combustion efficiency, while some particulate compounds like EC and PAHs may be overestimated in chamber study due to high and concentrated mass loading of PM and their impact on the measurements (Dhammapala et al., 2007). We controlled the preparation time to be less than 5 min (<2 min for the burning, ~3 min for chamber stabilization), minimizing the aging and diffusion/deposition of the primary emissions, and under the fixed combustion efficiency, seldom studies ever considered burning time as an impact factor on emission factor estimation (Dhammapala et al., 2007; Zhang et al., 2008). The chamber has a volume of 4.5m $^3$, mass concentration for smoke $PM_{2.5}$ at initial time

in the chamber are 10~30 mg m$^{-3}$, during sampling and monitoring from the chamber, the dilution ratio for is 10:1~50:1 (details in supporting information).

**Question 3**: The detection limits (MDL) for all analysis should be provided.

**Answer 3**: The detection limits (DL) were added in the manuscript as below presented, water soluble species were measured using IC techniques, and DLs were about 0.5~3.5 ng mL$^{-1}$.

Table 3 Detection limit and recovery rate of water soluble species measured by IC

| Water soluble species | ng mL$^{-1}$ | recovery rate |
|---|---|---|
| Na$^+$ | 0.59 | 99.1% |
| NH$_4^+$ | 0.63 | 96.5% |
| K$^+$ | 1.65 | 98.8% |
| Ca$^{2+}$ | 3.33 | 103.0% |
| Mg$^{2+}$ | 2.07 | 101.5% |
| F$^-$ | 0.72 | 99.3% |
| Cl$^-$ | 0.47 | 99.6% |
| NO$_2^-$ | 1.11 | 92.5% |
| NO$_3^-$ | 0.93 | 101.0% |
| SO$_4^{2-}$ | 0.68 | 99.0% |
| MeOH$^+$ | 1.12 | 94.4% |
| MMAH$^+$ | 0.59 | 97.3% |
| MEAH$^+$ | 1.03 | 106.1% |
| TEOH$^+$ | 1.13 | 95.0% |
| DEAH$^+$ + TMAH$^+$ | 0.61 | 103.6% |
| DMAH$^+$ | 1.37 | 104.2% |

OC-EC was measured using Thermal/Optical Carbon Analyzer, as aerosol samples were deposited onto the quartz filter, detection limits for total OC and EC were 0.82 and 0.22 µg C cm$^{-2}$. Elements (As, Zn, Pb, Cd, Cr, Ni, V, and Al) were measured by ICP-OES, DLs were within 0.2~2.1 ng mL$^{-1}$. PAHs and Phenols were measured using GC-MS, we prepared standard solutions with 6 concentration gradients of 0.020, 0.04, 0.10, 0.15, 0.25, 0.40 ppm for 16 target mixed PAHs, and 0.05, 0.10, 0.25, 0.50, 1.50, 4.00 ppm for 5 target mixed Phenols, all reagents used were of highest grades, and water used were Mili-Q one. Before all the measurements, recovery tests for the chemical components were conducted.

Table 4 Detection limit and recovery rate of multi-pollutants

| Elements | Molecular weight | recovery rate | DLs (ng mL$^{-1}$) |
|---|---|---|---|
| As | 74.9 | 93.0% | 2.0 |
| Zn | 65.4 | 98.3% | 0.2 |
| Pb | 207.2 | 99.7% | 1.5 |
| Cd | 112.4 | 96.7% | 0.1 |
| Ni | 58.7 | 102.0% | 0.2 |
| Cr | 52.0 | 94.9% | 0.2 |
| V | 50.9 | 98.7% | 0.5 |
| Al | 27.0 | 95.7% | 0.9 |
| **PAHs** | **Molecular weight** | **recovery rate** | **DLs (ng mL$^{-1}$)** |
| naphthalene | 128 | 97.0% | 0.8 |

| | | | |
|---|---|---|---|
| acenaphthylene | 152 | 98.5% | 1.8 |
| acenaphthene | 154 | 96.2% | 1.1 |
| flourene | 166 | 88.3% | 1.0 |
| anthracene | 178 | 96.9% | 0.5 |
| phenathrene | 178 | 97.9% | 0.9 |
| flouranthene | 202 | 94.4% | 1.0 |
| pyrene | 202 | 96.1% | 0.5 |
| benz[a]anthracene | 228 | 96.7% | 0.1 |
| chrysene | 228 | 94.1% | 2.0 |
| benzo[a]pyrene | 252 | 95.3% | 1.0 |
| benzo[b]flouranthene | 252 | 88.7% | 0.9 |
| benzo[k]flouranthene | 252 | 84.9% | 0.9 |
| benzo[g,h,i]pyrene | 276 | 84.9% | 0.6 |
| indeno[1,2,3-cd]pyrene | 278 | 90.9% | 0.6 |
| dibenz[a,h]anthracene | 276 | 84.9% | 0.8 |
| **Phenols** | **Molecular weight** | **recovery rate** | **DLs (ng mL$^{-1}$)** |
| phenol | 94 | 93.0% | 0.5 |
| 2-methoxyphenol | 124 | 82.0% | 2.0 |
| 4-ethylphenol | 122 | 84.5% | 4.0 |
| 4-ethyl-2-methoxyphenol | 152 | 86.1% | ~5.0 |
| 2,6-dimethoxyphenol | 154 | 87.5% | ~5.0 |

Line 290: add in the manuscript "Sampled filters were ultrasonically extracted with 15.0 mL deionized water (Mili-Q water, 18.2 MΩ cm), extracted solutions were filtrated using 0.2 μm filters before injected into IC for measurement. Detection limits (DLs) for the ions and aminiums were within 0.5~3.5 ng mL$^{-1}$, the correlation coefficients for all calibration curves were better than 0.99, and recovery rates for aminiums were in the range of 93%~106% (see in SI, Table S1). Details for the aminium measurements can be found in the work of Tao et al. (2016).

Line 303: add in the manuscript "The following wavelength lines of the ICP-OES analysis were used: As 189.042, Pb 220.353, Cd 228.802, Cr 205552, Ni 231.604, V 311.071, Zn 206.191, and Al 394.401. All reagents used were of highest grades, and recovery tests were conducted with standard additions, recoveries of each element were in the range of 93%~102% (see in SI, Table S1)."

Line 317: add in the manuscript "using an Agilent 6890 Series gas chromatography system coupled with a HP 5973 Mass Selective Detector (GC-MS, Agilent Technologies, Wilmington DE) . A DB-5ms (30 m × 0.32 mm ×0.25 mm, Agilent 123-5532) column was installed. The temperature programs were presented as follows: initially at 40 $^o$C, hold for 4 min, to 150 $^o$C at 20 $^o$C min$^{-1}$, then to 280 $^o$C at 5 $^o$C min$^{-1}$, hold for 10 min. The interface temperature was kept at 280 $^o$C, the MS was operated in electron impact mode with an ion source temperature of 230 $^o$C, and the high-purity helium (99.999%) carrier gas was maintained at a constant pressure of 16.2 psi with a flow of 2.0 mL min$^{-1}$. The calibration curves were optimized to be better than 99.9%. Prior to the measurements, PAHs and Phenols recovery studies were undertaken, and recoveries

were acceptable with rates of 82%~99% (see in SI, Table S1). In addition, Phenanthrene-d10 (Phe-d10) as internal standard surrogate was added into the PAHs mixture, recovery rate of which was 94%."

Line 342: add in the manuscript "The instrument detection limits for total OC and EC that deposit on the filter are 0.25 and 0.12 µg C cm$^{-2}$. Moreover, environmental EC in aerosol is a mixture of compounds from slightly charred, biodegradable materials to highly condensed and refractory soot, different EC materials have distinct different thermodynamic properties, study found char-EC decomposes much rapidly than soot when exposed to chemical and thermal oxidation, e.g., EC decomposition temperatures in air increased from ~520 °C for char to ~620 °C for soot, and exceeded 850 °C for graphite, thus, regarding to different oxidation temperatures,"

**Question 4**: It is interesting to determine char and soot, however, the temperature protocol is IMPROVE, but not NIOSH. Any calibrations have been performed with pure soot and char (standard)?

**Answer 4:** Yes, as you mentioned, most of the studies discriminated Char-/soot-EC from carbon analyzer based on TOR (Thermal-optical reflectance) with IMPROVE protocol, it is empirical function to define char-EC as EC1-PC and soot-EC as EC2+EC3, we did not calibrated the performance of the carbon analyzer used in soot/char classification, but we ever characterized diesel soot particles (diesel engine exhaust) and wood-char (600 °C power plant) using the OC-EC analyzer (TOT-NIOSH) combined with TG-MS (Thermalgravimetric-MS analysis) and chemical analysis, the result was given below:

[Figure]

Figure 1 characterization of soot particles (size distribution, morphology, and chemical profiles)

[Figure]

Figure 2 Chemical profiles and TG-MS-Carbon analysis of wood-char

It was obviously soot deposited into EC2 and EC3 fractions ((EC2+EC3)/EC~1.0), while char responded to EC1 ((EC1-PC)/EC~0.94), it seems that NIOSH protocol method is also possible to measure char-/soot-EC of aerosol. Han et al (2016) compared the OC-EC measurements between TOT and TOR methods with different protocols (IMPROVE, IMPROVE-A, EUSAAR-2 or modified NIOSH), good correlations among carbon results measured with the various methods were observed, but TOT-NIOSH method may overestimate PC fraction sacrificing EC part compared with TOR-IMPROVE method, thus different methods have impact on OC-EC split. However, char-/soot-EC were classified by different oxidation temperature, carbon analyzer based on TOT-NIOSH method in this study was set to operate at the same temperature gradient as Han et al (2007; 2009; 2016) ever performed, that means neglecting the impact of the methods on OC-EC split, calculation of char-EC and soot-EC can be also feasible in this study, but this method may underestimate char-EC, leading to lower char-EC/soot-EC ratio. More precise experiments will be conducted in the future to investigate the applicability of TOT-NIOSH method in char-/soot-EC measurements.

**Question 5:** Please describe how to screen agricultural fire from MODIS daily fire products? What are the selecting criteria?

**Answer 5:** Data of mainland agricultural fire sites was derived from the daily report of the Ministry of Environmental Protection of China (MEPC) (website: http://hjj.mep.gov.cn/jgjs/). MEPC selected MODIS (Moderate Resolution Imaging Spectroradiometer) Thermal Anomalies/Fire products based on space observations of NASA's Terra and Aqua satellites. Fire detection algorithm was used MODIS Thermal Anomalies/Fire daily products (MOD 14/MYD14) through brightness temperature derived from the MODIS 3.95 and 11.0 μm channels, of which 3.95 μm channel to detect fire sites via infrared radiation, and 11.0 μm

channel to derive cloud and land background temperature. The product is tile-based, with each product file spanning one of 460 MODIS tiles, of which 326 contain land pixels, and in 1 Km gridded cell over each daily (24 h) compositing period. Two observations per day are possible with the Terra overpass at 10:30 local time and the Aqua overpass at 13:30 local time. Version 4 of MODIS fire detection data was used combining with 1 Km land cover dataset (Global land cover-China), active fire detection that occurred on the land cover classes defined as "farm" and "mosaic of cropping" was identified as crop residue burning in fields.

**Question 6:** In this study, five crop residues were selected to determine their multi-pollutants emission factors, but there are other major crop resides not considered in this study, e.g. sugarcane, barley etc. There should be a reason to explain why such crop residues were not considered and how to determine the emission inventories in some provinces (with high sugarcane and barley production).

Answer 6: Thanks for your comment, the article presents pollutant emissions from major agircultural residues burning in China, and wheat, rice, corn, soybean, and cotton are surely the domianted agiruclutral productions in China, which contribute over 90 wt.% of national residue yields from China Statistic Yearbook (NBSC, 2013; Qin and Xie, 2011). Filed burning of wheat, rice, and corn staws burning was the most common agricultrual open burning and drawn much attention. Some crops like sugarcane and barley are regional cash crops that are mainly planted in Guangdong, Guangxi, and Hainan, where the total residue productions make less than 8 wt.% of national ones, even in the specific province themselves, sugarcane and barley residues contribute less than 30 wt.% of the straws on average (NBSC, 2013). Besides, this study focused on giving the updated and comprehensive emission factors of the most filed burnt agricultural straws via chamber simulation method, we are quite sorry that we cannot take all the residues into consideration.

**Question 7:** There is some typo errors found in the manuscript: Line 215, "costume-built" should be "custom-built" Line 330, "Corp straw" should be "Crop straw"
**Answer 7:** Thanks for your comment, we have fully checked and modified the manuscript.
Line 224: "custom-built" has been corrected
Line 390: "Crop straw" has been corrected

**References:**

C. Li *et al.*, Evolution of biomass burning smoke particles in the dark, *ATMOS ENVIRON* **120**(2015), pp. 244-252.

H. Zhang *et al.*, A laboratory study of agricultural crop residue combustion in China: Emission factors and emission inventory, *ATMOS ENVIRON* **42**(2008), pp. 8432-8441.

H. Zhang *et al.*, Particle Size Distribution and Polycyclic Aromatic Hydrocarbons Emissions from Agricultural Crop Residue Burning, *ENVIRON SCI TECHNOL* **45**(2011), pp. 5477-5482.

J. Huo *et al.*, Online single particle analysis of chemical composition and mixing state of crop straw burning particles: from laboratory study to field measurement, *FRONT ENV SCI ENG* **10**(2016), pp. 244-252.

K. Hayashi *et al.*, Trace gas and particle emissions from open burning of three cereal crop residues: Increase in residue moistness enhances emissions of carbon monoxide, methane, and particulate organic carbon, *ATMOS ENVIRON* **95**(2014), pp. 36-44.

N.T.K. Oanh *et al.*, Characterization of particulate matter emission from open burning of rice straw, *ATMOS ENVIRON* **45**(2011), pp. 493-502.

NBSC, China Statistical Yearbook 2013., China Statistics Press Beijing, China (2013).

R. Dhammapala, C. Claiborn, C. Simpson and J. Jimenez, Emission factors from wheat and Kentucky bluegrass stubble burning: Comparison of field and simulated burn experiments, *ATMOS ENVIRON* **41**(2007), pp. 1512-1520.

Y. Han *et al.*, Evaluation of the thermal/optical reflectance method for discrimination between char- and soot-EC, *CHEMOSPHERE* **69**(2007), pp. 569-574.

Y. Qin and S.D. Xie, Historical estimation of carbonaceous aerosol emissions from biomass open burning in China for the period 1990–2005, *ENVIRON POLLUT* **159**(2011), pp. 3316-3323.

Y.M. Han *et al.*, Carbonaceous aerosols in megacity Xi'an, China: Implications of thermal/optical protocols comparison, *ATMOS ENVIRON* **132**(2016), pp. 58-68.

Y.M. Han, S.C. Lee, J.J. Cao, K.F. Ho and Z.S. An, Spatial distribution and seasonal variation of char-EC and soot-EC in the atmosphere over China, *ATMOS ENVIRON* **43**(2009), pp. 6066-6073.

---

## Author Response (AR2)

Reviewer Comments on acp-2016-651 of "Multi-pollutants emissions from the burning of major agricultural residues in China and the related health-economic effect assessment" by Li C. et al:

This study investigates the emission factors of multi-pollutants from five major crop residues in china, and tries to estimate emission inventory and their corresponding health-economic effect. This revised version replied most of the technical questions. However, detailed explanations about the experimental design should be provided and the uncertainty of the results should be addressed also.

Answer: Thanks for your review and comments, we will try to modify the manuscript following your suggestion.

1) Line 201-205: I agree with the authors that pretreatment of biomass fuel in burning simulation is a practical and necessary procedure to ensure the result can be comparable with other studies under the defined conditions. But when the crop residues were dehydrated at 100 degree C for 24 hrs, very low moisture content were obtained compared with other studies, and it tends to have a much lower EFs as mentioned in the manuscript. Moreover, most crop residues are not well dried in a field except in the dry season. So what is the uncertainty when calculating emission factor and health economic effect if 2% water contents were used in this study?

Answer 1: Thanks for your comment, and that is a practical and complicated question for emission factor research. Studies have found variance of smoke emissions from biomass burning with respect to moisture content, but profound relationship between EFs and water content of biomass was still undefined, it was hard to tell exact response of EFs to moisture content for various straws and chemical species, and EFs from many documents that were measured under different water content of fuel issues are comparable. Hayashi et al. (2014) conducted moisture effect study concerning three kinds of straws with only two water contents (~10 wt.% $vs$ ~20 wt.%), they found positive relationship for PM and moisture, but no effect of moisture on EC and most inorganic species emissions, and the regressed functions between EFs and moisture content were only applicable for moisture content range from 10~20 wt.%. Emanuela et al. (2011) simulated forest biomass burning regarding to various moisture (7~50 wt.%) and mass of fuel, they found no significant EFs changes for aromatic hydrocarbons but the EF of PM above 17 wt.% water content (no significant changes for PM EF with moisture content from 7~17 wt.%), however they did not phase out MCE effect on EFs for PM, thus, PM EFs in line with water content was not precise in their work. Most studies commonly applied $\pm 50\,\% \sim \pm 150\,\%$ variation for the EFs they cited for usage in the emission inventory calculation, and dry matter fraction for the residues were taken as 87~94 wt.% (that means moisture contents were assumed as 6~13 wt.%) (Bond et al., 2004; Qin et al., 2011, 2012), in this work, we applied dry matter fractions for field straw to be 85~94 wt.% (moisture content as 5~15 wt.%), and applied $\pm 30\,\%$ variation for dry matter and other burning activities, but to EFs, we conservatively applied coefficient of variation as the measured one. Deviation of EFs in this study can explain about 35%~47% of uncertainties in health economic assessment. And it was not possible to assess exact deviation of EFs under 2 wt.% in this study and EFs for practical burnings in the field, but your suggestion will help our future experiment design, and we will conduct field burning and lab simulation experiments to investigate smoke emissions for different agricultural residues under different conditions.

2) Line 233-242: Were CO and $CO_2$ measured by GC-FID continuously during the experiment? If yes, what is the time resolution of the repeated measurement? If no, when CO and $CO_2$ were collected during the experiment? Moreover, please show the results of combustion efficiency in supplement information.

Answer 2: CO and $CO_2$ were measured for each test using GC-FID, but not continuously covering the entire combustion process. In each test, all the emissions from agricultural burning were introduced into the chamber, then CO and $CO_2$ in the chamber were measured and blank corrected to represent the burning conditions, which can be viewed as integrated burning conditions.
The MCE for the burnings were collected in the supporting information as Table S1:

| Type of agricultural residue | MCE |
| --- | --- |
| Wheat straw | 0.91 ± 0.03 |
| Corn straw | 0.89 ± 0.07 |
| Rice straw | 0.96 ± 0.03 |
| Cotton residue | 0.93 ± 0.05 |
| Soybean residue | 0.91 ± 0.05 |

Method for CO and $CO_2$ standard curves preparation and MCE measurements were also added in the SI:
CO and $CO_2$ were injected into a gas chromatograph (Model 930, Shanghai Hai Xin Gas Chromatograph Co., LTD) equipped with a flame ionization detector, an Ni-H convertor, and a stainless steel column (2 m long) packed with 15% DNP. Before calibration, 2 mL CO was diluted with $N_2$ to a final volume of 500 $cm^3$ in a clean Teflon bag. The amounts of $CO_2$ used for calibration were 40, 60, 80, 100, 200, 500, 800, and 1000 μL, and the amounts of diluted CO were 20, 40, 80, 100, and 200 μL, respectively. The gas volume and the corresponding peak of each sample injected into the GC were recorded to calculate regression function, with which the CO and $CO_2$ concentrations in the aerosol chamber can be determined. Background CO and $CO_2$ concentrations were deducted. The working curves for both CO and $CO_2$ were linear with $R^2 > 0.99$.

3) Line 244-254: What is the scam time of WPS? The startup transients may be missed due to the long scan time limitation. High time-resolution scans are recommended for future studies.

Answer 3: Thanks for your suggestion, the scan time of WPS in this study was set to be 3 min/loop, the startup measurements were fixed at 5 min later after stabilization
of aerosol chamber. And for each test, particle size distribution was averaged for 4
loops of WPS measurements, which will surely capture the size character of straw
burning aerosol. Smoke particle ages rapidly in size once emitted and introduced into
the chamber (as below figure presented size evolution of smoke aerosol in chamber,
referring to Li et al., 2015), that's why we controlled each burning and fixed the
startup measurement time to ensure size distribution for fresh smoke aerosol being
reliable. In the future, we will use high-time resolution scans to measure the size of
aerosol from emission to aging.

[Figure]

Figure 1 Time profiles of smoke particle number size distribution

4) Line 336-344: As no calibration were carried out for soot/char classification, and
TOT-NIOSH method may underestimate char-EC, leading to lower char-EC/soot-EC
ratio. I don't prefer to report char and soot EC results until more precise experiments
will be conducted in the future.

Answer 4: Thanks for your suggestion, we are sorry that no calibration was
conducted for char- and soot-EC splitting, we have deleted related data and
discussions in the manuscript. In the future we will conduct more precisely
experiment on char- and soot-EC measurements.

5) Line 433-434: What is the theory behind the assumption that the burning rates
were inverse proportional to peasants' agricultural income proportion (AIP)?

Answer 5: The theory that the burning rates were inverse proportional to peasants'
agricultural income proportion (AIP) in this study was modified from assumption of
Cao et al. (2006), Wang et al. (2008), and Qin et al.(2011, 2012). The relationship
between spatial filed burning rates and peasants' income in China was confirmed
(spatial filed burning rate *vs* peasants' income) as higher percentage of straw filed
burning occurred at economic advanced regions, then, Cao et al. and Qin et al.
assumed temporal filed burning rates linearly changes with peasants' income increase (temporal filed burning rate $\propto$ peasants' income), however, their assumptions were unreasonable, as filed burning rate of specific region can hardly change in line with regional peasants' income increase, from 2000 to 2012, peasants' income primarily increased by 3~5 times in China (National Statistic Bureau of China), filed burning rates for most provinces would be 100 % or over under the assumption and in the study of Cao et al. (2006, 2011) and Qin et al (2011, 2012). It was reported that regions where the primary industry is developed or agricultural economy dominated (higher peasants' agricultural income proportion) have less filed burning rate, and these regions will also decrease the crop straw field disposal and make more usage of straw as domestic fuel or into other biomass waste treatment, thus we proposed the assumption based on previous studies that the burning rates were inverse proportional to peasants' agricultural income proportion.

6) Line 762-770: What is the objective to determine expulsion-accumulation coefficients of PAHs in OC in this study? Please explain the obtained results.

Answer 6: The objective to determine expulsion-accumulation coefficients of PAHs in OC and EC is try to provide an empirical method to distinguish source of PAHs, which can be used to estimate PAHs mass fraction from OC and EC measurement for crop straw burnings, and the function may also help assess PAHs partition in carbonaceous aerosol during aerosol transportation and aging process.

[revised manuscript text omitted]